# The Surprising Simplicity of the Early-Time Learning Dynamics of Neural Networks

Wei Hu[*]        Lechao Xiao[†]        Ben Adlam[‡]        Jeffrey Pennington[§]

## Abstract

Modern neural networks are often regarded as complex black-box functions whose behavior is difficult to understand owing to their nonlinear dependence on the data and the nonconvexity in their loss landscapes. In this work, we show that these common perceptions can be completely false in the early phase of learning. In particular, we formally prove that, for a class of well-behaved input distributions, the early-time learning dynamics of a two-layer fully-connected neural network can be mimicked by training a simple linear model on the inputs. We additionally argue that this surprising simplicity can persist in networks with more layers and with convolutional architecture, which we verify empirically. Key to our analysis is to bound the spectral norm of the difference between the Neural Tangent Kernel (NTK) at initialization and an affine transform of the data kernel; however, unlike many previous results utilizing the NTK, we do not require the network to have disproportionately large width, and the network is allowed to escape the kernel regime later in training.

## 1 Introduction

Modern deep learning models are enormously complex function approximators, with many state-of-the-art architectures employing millions or even billions of trainable parameters [Radford et al., 2019, Adiwardana et al., 2020]. While the raw parameter count provides only a crude approximation of a model's capacity, more sophisticated metrics such as those based on PAC-Bayes [McAllester, 1999, Dziugaite and Roy, 2017, Neyshabur et al., 2017b], VC dimension [Vapnik and Chervonenkis, 1971], and parameter norms [Bartlett et al., 2017, Neyshabur et al., 2017a] also suggest that modern architectures have very large capacity. Moreover, from the empirical perspective, practical models are flexible enough to perfectly fit the training data, even if the labels are pure noise [Zhang et al., 2017]. Surprisingly, these same high-capacity models generalize well when trained on real data, even without any explicit control of capacity.

These observations are in conflict with classical generalization theory, which contends that models of intermediate complexity should generalize best, striking a balance between the bias and the variance of their predictive functions. To reconcile theory with observation, it has been suggested that deep neural networks may enjoy some form of implicit regularization induced by gradient-based training algorithms that biases the trained models towards simpler functions. However, the exact notion of simplicity and the mechanism by which it might be achieved remain poorly understood except in certain simplistic settings.

---

[*]Princeton University. Work partly performed at Google. Email: `huwei@cs.princeton.edu`

[†]Google Research, Brain Team. Email: `xlc@google.com`

[‡]Google Research, Brain Team. Work done as a member of the Google AI Residency program (http://g.co/brainresidency). Email: `adlam@google.com`

[§]Google Research, Brain Team. Email: `jpennin@google.com`

One concrete mechanism by which such induced simplicity can emerge is the hypothesis that neural networks learn simple functions early in training, and increasingly build up their complexity in later time. In particular, recent empirical work Nakkiran et al. [2019] found that, intriguingly, in some natural settings the simple function being learned in the early phase may just be a linear function of the data.

In this work, we provide a novel theoretical result to support this hypothesis. Specifically, we formally prove that, for a class of well-behaved input distributions, the early-time learning dynamics of gradient descent on a two-layer fully-connected neural network with any common activation can be mimicked by training a simple model of the inputs. When training the first layer only, this simple model is a linear function of the input features; when training the second layer or both layers, it is a linear function of the features and their $\ell_2$ norm. This result implies that neural networks do not fully exercise their nonlinear capacity until late in training.

Key to our technical analysis is a bound on the spectral norm of the difference between the Neural Tangent Kernel (NTK) [Jacot et al., 2018] of the neural network at initialization and that of the linear model; indeed, a weaker result, like a bound on the Frobenius norm, would be insufficient to establish our result. Although the NTK is usually associated with the study of ultra-wide networks, our result only has a mild requirement on the width and allows the network to leave the kernel regime later in training. While our formal result focuses on two-layer fully-connected networks and data with benign concentration properties (specified in Assumption 3.1), we argue with theory and provide empirical evidence that the same linear learning phenomenon persists for more complex architectures and real-world datasets.

**Related work.** The early phase of neural network training has been the focus of considerable recent research. Frankle and Carbin [2019] found that sparse, trainable subnetworks – "lottery tickets" – emerge early in training. Achille et al. [2017] showed the importance of early learning from the perspective of creating strong connections that are robust to corruption. Gur-Ari et al. [2018] observed that after a short period of training, subsequent gradient updates span a low-dimensional subspace. Li et al. [2019a], Lewkowycz et al. [2020] showed that an initial large learning rate can benefit late-time generalization performance.

Implicit regularization of (stochastic) gradient descent has also been studied in various settings, suggesting a bias towards large-margin, low-norm, or low-rank solutions [Gunasekar et al., 2017, 2018, Soudry et al., 2018, Li et al., 2018, Ji and Telgarsky, 2019a,b, Arora et al., 2019a, Lyu and Li, 2019, Chizat and Bach, 2020, Razin and Cohen, 2020]. These results mostly aim to characterize the final solutions at convergence, while our focus is on the early-time learning dynamics. Another line of work has identified that deep linear networks gradually increase the rank during training [Arora et al., 2019a, Saxe et al., 2014, Lampinen and Ganguli, 2018, Gidel et al., 2019].

A line of work adopted the Fourier perspective and demonstrated that low-frequency functions are often learned first [Rahaman et al., 2018, Xu, 2018, Xu et al., 2019a,b]. Based on the NTK theory, Arora et al. [2019c] showed that for very wide networks, components lying in the top eigenspace of the NTK are learned faster than others. Using this principle, Su and Yang [2019], Cao et al. [2019] analyzed the spectrum of the infinite-width NTK. However, in order to obtain precise characterization of the spectrum these papers require special data distributions such as uniform distribution on the sphere.

Most relevant to our work is the finding of Nakkiran et al. [2019] that a neural network learned in the early phase of training can be almost fully explained by a linear function of the data. They supported this claim empirically by examining an information theoretic measure between the predictions of the neural network and the linear model. Our result formally proves that neural network and a corresponding linear model make similar predictions in early time, thus providing a theoretical explanation of their empirical finding.

**Paper organization.** In Section 2, we introduce notation and briefly recap the Neural Tangent Kernel. In Section 3, we present our main theoretical results on two-layer neural networks as well as empirical verification. In Section 4, we discuss extensions to more complicated architecture from both theoretical and empirical aspects. We conclude in Section 5, and defer additional experimental results and all the proofs to the appendices.

## 2 Preliminaries

**Notation.** We use bold lowercases $\boldsymbol{a}, \boldsymbol{b}, \boldsymbol{\alpha}, \boldsymbol{\beta}, \ldots$ to represent vectors, bold uppercases $\boldsymbol{A}, \boldsymbol{B}, \ldots$ to represent matrices, and unbold letters $a, b, \alpha, \beta, \ldots$ to represent scalars. We use $[\boldsymbol{A}]_{i,j}$ or $[\boldsymbol{a}]_i$ to index the entries in matrices or vectors. We denote by $\|\cdot\|$ the spectral norm (largest singular value) of a matrix or the $\ell_2$ norm of a vector, and denote by $\|\cdot\|_F$ the Frobenius norm of a matrix. We use $\langle \cdot, \cdot \rangle$ to represent the standard Euclidean inner product between vectors or matrices, and use $\odot$ to denote the Hadamard (entry-wise) product between matrices. For a positive semidefinite (psd) matrix $\boldsymbol{A}$, let $\boldsymbol{A}^{1/2}$ be the psd matrix such that $(\boldsymbol{A}^{1/2})^2 = \boldsymbol{A}$; let $\lambda_{\max}(\boldsymbol{A})$ and $\lambda_{\min}(\boldsymbol{A})$ be the maximum and minimum eigenvalues of $\boldsymbol{A}$.

Let $[n] := \{1, 2, \ldots, n\}$. For $a, b \in \mathbb{R}$ ($b > 0$), we use $a \pm b$ to represent any number in the interval $[a - b, a + b]$. Let $\boldsymbol{I}_d$ be the $d \times d$ identity matrix, $\boldsymbol{0}_d$ be the all-zero vector in $\mathbb{R}^d$, and $\boldsymbol{1}_d$ be the all-one vector in $\mathbb{R}^d$; we write $\boldsymbol{I}, \boldsymbol{0}, \boldsymbol{1}$ when their dimensions are clear from context. We denote by $\mathsf{Unif}(A)$ the uniform distribution over a set $A$, and by $\mathcal{N}(\mu, \sigma^2)$ or $\mathcal{N}(\boldsymbol{\mu}, \boldsymbol{\Sigma})$ the univariate/multivariate Gaussian distribution. Throughout the paper we let $g$ be a random variable with the standard normal distribution $\mathcal{N}(0, 1)$.

We use the standard $O(\cdot), \Omega(\cdot)$ and $\Theta(\cdot)$ notation to only hide universal constant factors. For $a, b \geq 0$, we also use $a \lesssim b$ or $b \gtrsim a$ to mean $a = O(b)$, and use $a \ll b$ or $b \gg a$ to mean $b \geq Ca$ for a sufficiently large universal constant $C > 0$. Throughout the paper, "high probability" means a large constant probability arbitrarily close to 1 (such as 0.99).

**Recap of Neural Tangent Kernel (NTK) [Jacot et al., 2018].** Consider a single-output neural network $f(\boldsymbol{x}; \boldsymbol{\theta})$ where $\boldsymbol{x}$ is the input and $\boldsymbol{\theta}$ is the collection of parameters in the network. Around a reference network with parameters $\bar{\boldsymbol{\theta}}$, we can do a local first-order approximation:

$$f(\boldsymbol{x}; \boldsymbol{\theta}) \approx f(\boldsymbol{x}; \bar{\boldsymbol{\theta}}) + \langle \nabla_{\boldsymbol{\theta}} f(\boldsymbol{x}; \bar{\boldsymbol{\theta}}), \boldsymbol{\theta} - \bar{\boldsymbol{\theta}} \rangle.$$

Thus when $\boldsymbol{\theta}$ is close to $\bar{\boldsymbol{\theta}}$, for a given input $\boldsymbol{x}$ the network can be viewed as linear in $\nabla_{\boldsymbol{\theta}} f(\boldsymbol{x}; \bar{\boldsymbol{\theta}})$. This gradient feature map $\boldsymbol{x} \mapsto \nabla_{\boldsymbol{\theta}} f(\boldsymbol{x}; \bar{\boldsymbol{\theta}})$ induces a kernel $K_{\bar{\boldsymbol{\theta}}}(\boldsymbol{x}, \boldsymbol{x}') := \langle \nabla_{\boldsymbol{\theta}} f(\boldsymbol{x}; \bar{\boldsymbol{\theta}}), \nabla_{\boldsymbol{\theta}} f(\boldsymbol{x}'; \bar{\boldsymbol{\theta}}) \rangle$ which is called the NTK at $\bar{\boldsymbol{\theta}}$. Gradient descent training of the neural network can be viewed as kernel gradient descent on the function space with respect to the NTK. We use *NTK matrix* to refer to an $n \times n$ matrix that is the NTK evaluated on $n$ datapoints.

While in general the NTK is random at initialization and can vary significantly during training, it was shown that, for a suitable network parameterization (known as the "NTK parameterization"), when the width goes to infinity or is sufficiently large, the NTK converges to a deterministic limit at initialization and barely changes during training [Jacot et al., 2018, Lee et al., 2019, Arora et al., 2019b, Yang, 2019], so that the neural network trained by gradient descent is equivalent to a kernel method with respect to a fixed kernel. However, for networks with practical widths, the NTK does usually stray far from its initialization.

## 3 Two-Layer Neural Networks

We consider a two-layer fully-connected neural network with $m$ hidden neurons defined as:

$$f(\boldsymbol{x}; \boldsymbol{W}, \boldsymbol{v}) := \frac{1}{\sqrt{m}} \sum_{r=1}^{m} v_r \phi\left(\boldsymbol{w}_r^\top \boldsymbol{x} / \sqrt{d}\right) = \frac{1}{\sqrt{m}} \boldsymbol{v}^\top \phi\left(\boldsymbol{W} \boldsymbol{x} / \sqrt{d}\right), \tag{1}$$

where $\boldsymbol{x} \in \mathbb{R}^d$ is the input, $\boldsymbol{W} = [\boldsymbol{w}_1, \ldots, \boldsymbol{w}_m]^\top \in \mathbb{R}^{m \times d}$ is the weight matrix in the first layer, and $\boldsymbol{v} = [v_1, \ldots, v_m]^\top \in \mathbb{R}^m$ is the weight vector in the second layer.[5] Here $\phi : \mathbb{R} \to \mathbb{R}$ is an activation function that acts entry-wise on vectors or matrices.

Let $\{(\boldsymbol{x}_i, y_i)\}_{i=1}^n \subset \mathbb{R}^d \times \mathbb{R}$ be $n$ training samples where $\boldsymbol{x}_i$'s are the inputs and $y_i$'s are their associated labels. Denote by $\boldsymbol{X} = [\boldsymbol{x}_1, \ldots, \boldsymbol{x}_n]^\top \in \mathbb{R}^{n \times d}$ the data matrix and by $\boldsymbol{y} = [y_1, \ldots, y_n]^\top \in \mathbb{R}^n$ the label vector. We assume $|y_i| \leq 1$ for all $i \in [n]$.

We consider the following $\ell_2$ training loss:

$$L(\boldsymbol{W}, \boldsymbol{v}) := \frac{1}{2n} \sum_{i=1}^{n} \left(f(\boldsymbol{x}_i; \boldsymbol{W}, \boldsymbol{v}) - y_i\right)^2, \tag{2}$$

and run vanilla gradient descent (GD) on the objective (2) starting from random initialization. Specifically, we use the following *symmetric initialization* for the weights $(\boldsymbol{W}, \boldsymbol{v})$:

$$\boldsymbol{w}_1, \ldots, \boldsymbol{w}_{m/2} \overset{\text{i.i.d.}}{\sim} \mathcal{N}(\boldsymbol{0}_d, \boldsymbol{I}_d), \quad \boldsymbol{w}_{i+m/2} = \boldsymbol{w}_i \, (\forall i \in [m/2]),$$

$$v_1, \ldots, v_{m/2} \overset{\text{i.i.d.}}{\sim} \mathsf{Unif}(\{1, -1\}),\, [6] \quad v_{i+m/2} = -v_i \, (\forall i \in [m/2]). \tag{3}$$

The above initialization scheme was used by Chizat et al. [2019], Zhang et al. [2019], Hu et al. [2020], Bai and Lee [2020], etc. It initializes the network to be the difference between two identical (random) networks, which has the benefit of ensuring zero output: $f(\boldsymbol{x}; \boldsymbol{W}, \boldsymbol{v}) = 0 \, (\forall \boldsymbol{x} \in \mathbb{R}^d)$, without altering the NTK at initialization. An alternative way to achieve the same effect is to subtract the function output at initialization [Chizat et al., 2019].

Let $(\boldsymbol{W}(0), \boldsymbol{v}(0))$ be a set of initial weights drawn from the symmetric initialization (3). Then the weights are updated according to GD:

$$\boldsymbol{W}(t+1) = \boldsymbol{W}(t) - \eta_1 \nabla_{\boldsymbol{W}} L\left(\boldsymbol{W}(t), \boldsymbol{v}(t)\right), \quad \boldsymbol{v}(t+1) = \boldsymbol{v}(t) - \eta_2 \nabla_{\boldsymbol{v}} L\left(\boldsymbol{W}(t), \boldsymbol{v}(t)\right), \tag{4}$$

where $\eta_1$ and $\eta_2$ are the learning rates. Here we allow potentially different learning rates for flexibility.

Now we state the assumption on the input distribution used in our theoretical results.

**Assumption 3.1** (input distribution). *The datapoints $\boldsymbol{x}_1, \ldots, \boldsymbol{x}_n$ are i.i.d. samples from a distribution $\mathcal{D}$ over $\mathbb{R}^d$ with mean $\boldsymbol{0}$ and covariance $\boldsymbol{\Sigma}$ such that $\mathrm{Tr}[\boldsymbol{\Sigma}] = d$ and $\|\boldsymbol{\Sigma}\| = O(1)$. Moreover, $\boldsymbol{x} \sim \mathcal{D}$ can be written as $\boldsymbol{x} = \boldsymbol{\Sigma}^{1/2} \bar{\boldsymbol{x}}$ where $\bar{\boldsymbol{x}} \in \mathbb{R}^d$ satisfies $\mathbb{E}[\bar{\boldsymbol{x}}] = \boldsymbol{0}_d$, $\mathbb{E}[\bar{\boldsymbol{x}}\bar{\boldsymbol{x}}^\top] = \boldsymbol{I}_d$, and $\bar{\boldsymbol{x}}$'s entries are independent and are all $O(1)$-subgaussian.[7]*

Note that a special case that satisfies Assumption 3.1 is the Gaussian distribution $\mathcal{N}(\boldsymbol{0}, \boldsymbol{\Sigma})$, but we allow a much larger class of distributions here. The subgaussian assumption is made due to the probabilistic tail bounds used in the analysis, and it can be replaced with a weaker bounded moment condition. The independence between $\bar{\boldsymbol{x}}$'s entries may also be dropped if its density is strongly log-concave. We choose to use Assumption 3.1 as the most convenient way to present our results.

We allow $\phi$ to be any of the commonly used activation functions, including ReLU, Leaky ReLU, Erf, Tanh, Sigmoid, Softplus, etc. Formally, our requirement on $\phi$ is the following:

**Assumption 3.2** (activation function). *The activation function $\phi(\cdot)$ satisfies either of the followings:*

*(i)* *smooth activation: $\phi$ has bounded first and second derivatives: $|\phi'(z)| = O(1)$ and $|\phi''(z)| = O(1) \, (\forall z \in \mathbb{R})$, or*

*(ii)* *piece-wise linear activation: $\phi(z) = \begin{cases} z & (z \geq 0) \\ az & (z < 0) \end{cases}$ for some $a \in \mathbb{R}, |a| = O(1)$.[8]*

We will consider the regime where the data dimension $d$ is sufficiently large (i.e., larger than any constant) and the number of datapoints $n$ is at most some polynomial in $d$ (i.e., $n \leq d^{O(1)}$). These imply $\log n = O(\log d) < d^c$ for any constant $c > 0$.

Under Assumption 3.1, the datapoints satisfy the following concentration properties:

**Claim 3.1.** *Suppose $n \gg d$. Then under Assumption 3.1, with high probability we have $\frac{\|\boldsymbol{x}_i\|^2}{d} = 1 \pm O\left(\sqrt{\frac{\log n}{d}}\right) (\forall i \in [n]), \frac{|\langle \boldsymbol{x}_i, \boldsymbol{x}_j \rangle|}{d} = O\left(\sqrt{\frac{\log n}{d}}\right) (\forall i, j \in [n], i \neq j), and \|\boldsymbol{X}\boldsymbol{X}^\top\| = \Theta(n).$*

The main result in this section is to formally prove that the neural network trained by GD is approximately a linear function in the early phase of training. As we will see, there are distinct contributions coming from the two layers. Therefore, it is helpful to divide the discussion into the cases of training the first layer only, the second layer only, and both layers together. All the omitted proofs in this section are given in Appendix D.

## 3.1 Training the First Layer

Now we consider only training the first layer weights $\boldsymbol{W}$, which corresponds to setting $\eta_2 = 0$ in (4). Denote by $f_t^1 : \mathbb{R}^d \to \mathbb{R}$ the network at iteration $t$ in this case, namely $f_t^1(\boldsymbol{x}) := f(\boldsymbol{x}; \boldsymbol{W}(t), \boldsymbol{v}(t)) = f(\boldsymbol{x}; \boldsymbol{W}(t), \boldsymbol{v}(0))$ (note that $\boldsymbol{v}(t) = \boldsymbol{v}(0)$).

The linear model which will be proved to approximate the neural network $f_t^1$ in the early phase of training is $f^{\text{lin1}}(\boldsymbol{x}; \boldsymbol{\beta}) := \boldsymbol{\beta}^\top \boldsymbol{\psi}_1(\boldsymbol{x})$, where

$$\boldsymbol{\psi}_1(\boldsymbol{x}) := \frac{1}{\sqrt{d}} \begin{bmatrix} \zeta \boldsymbol{x} \\ \nu \end{bmatrix}, \qquad \text{with } \zeta = \mathbb{E}[\phi'(g)] \text{ and } \nu = \mathbb{E}[g \phi'(g)] \cdot \sqrt{\text{Tr}[\boldsymbol{\Sigma}^2]/d}. \tag{5}$$

Here recall that $g \sim \mathcal{N}(0, 1)$. We also consider training this linear model via GD on the $\ell_2$ loss, this time starting from zero:

$$\boldsymbol{\beta}(0) = \mathbf{0}_{d+1}, \quad \boldsymbol{\beta}(t+1) = \boldsymbol{\beta}(t) - \eta_1 \nabla_{\boldsymbol{\beta}} \frac{1}{2n} \sum_{i=1}^n \left( f^{\text{lin1}}(\boldsymbol{x}_i; \boldsymbol{\beta}(t)) - y_i \right)^2. \tag{6}$$

We let $f_t^{\text{lin1}}$ be the model learned at iteration $t$, i.e., $f_t^{\text{lin1}}(\boldsymbol{x}) := f^{\text{lin1}}(\boldsymbol{x}; \boldsymbol{\beta}(t))$.

We emphasize that (4) and (6) have the same learning rate $\eta_1$. Our theorem below shows that $f_t^1$ and $f_t^{\text{lin1}}$ are close to each other in the early phase of training:

**Theorem 3.2** (main theorem for training the first layer). *Let $\alpha \in (0, \frac{1}{4})$ be a fixed constant. Suppose the number of training samples $n$ and the network width $m$ satisfy $n \gtrsim d^{1+\alpha}$ and $m \gtrsim d^{1+\alpha}$. Suppose $\eta_1 \ll d$ and $\eta_2 = 0$. Then there exists a universal constant $c > 0$ such that with high probability, for all $0 \le t \le T = c \cdot \frac{d \log d}{\eta_1}$ simultaneously, the learned neural network $f_t^1$ and the linear model $f_t^{\text{lin1}}$ at iteration $t$ are close on average on the training data:*

$$\frac{1}{n} \sum_{i=1}^n \left( f_t^1(\boldsymbol{x}_i) - f_t^{\text{lin1}}(\boldsymbol{x}_i) \right)^2 \lesssim d^{-\Omega(\alpha)}. \tag{7}$$

*Moreover, $f_t^1$ and $f_t^{\text{lin1}}$ are also close on the underlying data distribution $\mathcal{D}$. Namely, with high probability, for all $0 \le t \le T$ simultaneously, we have*

$$\mathbb{E}_{\boldsymbol{x} \sim \mathcal{D}} \left[ \min\{ (f_t^1(\boldsymbol{x}) - f_t^{\text{lin1}}(\boldsymbol{x}))^2, 1 \} \right] \lesssim d^{-\Omega(\alpha)} + \sqrt{\frac{\log T}{n}}. \tag{8}$$

Theorem 3.2 ensures that the neural network $f_t^1$ and the linear model $f_t^{\text{lin1}}$ make almost the same predictions in the early time of training. This agreement is not only on the training data, but also over the underlying input distribution $\mathcal{D}$. Note that this does not mean that $f_t^1$ and $f_t^{\text{lin1}}$ are the same on the entire space $\mathbb{R}^d$ – they might still differ significantly at low-density regions of $\mathcal{D}$. We also remark that our result has no assumption on the labels $\{y_i\}$ except they are bounded.

The width requirement in Theorem 3.2 is very mild as it only requires the width $m$ to be larger than $d^{1+\alpha}$ for some small constant $\alpha$. Note that the width is allowed to be much smaller than the number of samples $n$, which is usually the case in practice.

The agreement guaranteed in Theorem 3.2 is up to iteration $T = c \cdot \frac{d \log d}{\eta_1}$ (for some constant $c$). It turns out that for well-conditioned data, after $T$ iterations, a near optimal linear model will have been reached. This means that *the neural network in the early phase approximates a linear model all the way until the linear model converges to the optimum*. See Corollary 3.3 below.

**Corollary 3.3** (well-conditioned data). *Under the same setting as Theorem 3.2, and additionally assume that the data distribution $\mathcal{D}$'s covariance $\boldsymbol{\Sigma}$ satisfies $\lambda_{\min}(\boldsymbol{\Sigma}) = \Omega(1)$. Let $\boldsymbol{\beta}_* \in \mathbb{R}^{d+1}$ be the optimal parameter for the linear model that GD (6) converges to, and denote $f_*^{\text{lin1}}(\boldsymbol{x}) := f^{\text{lin1}}(\boldsymbol{x}; \boldsymbol{\beta}_*)$. Then with high probability, after $T = c \cdot \frac{d \log d}{\eta_1}$ iterations (for some universal constant $c$), we have*

$$\frac{1}{n} \sum_{i=1}^n \left( f_T^1(\boldsymbol{x}_i) - f_*^{\text{lin1}}(\boldsymbol{x}_i) \right)^2 \lesssim d^{-\Omega(\alpha)}, \ \mathbb{E}_{\boldsymbol{x} \sim \mathcal{D}} \left[ \min\{ (f_T^1(\boldsymbol{x}) - f_*^{\text{lin1}}(\boldsymbol{x}))^2, 1 \} \right] \lesssim d^{-\Omega(\alpha)} + \sqrt{\frac{\log T}{n}}.$$

### 3.1.1 Proof Sketch of Theorem 3.2

The proof of Theorem 3.2 consists of showing that the NTK matrix for the first layer at random initialization evaluated on the training data is close to the kernel matrix corresponding to the linear model (5), and that furthermore this agreement persists in the early phase of training up to iteration $T$. Specifically, the NTK matrix $\mathbf{\Theta}_1(\boldsymbol{W}) \in \mathbb{R}^{n \times n}$ at a given first-layer weight matrix $\boldsymbol{W}$, and the kernel matrix $\mathbf{\Theta}^{\mathrm{lin}1} \in \mathbb{R}^{n \times n}$ for the linear model (5) can be computed as:

$$\mathbf{\Theta}_1(\boldsymbol{W}) := \left(\phi'(\boldsymbol{X}\boldsymbol{W}^\top/\sqrt{d})\phi'(\boldsymbol{X}\boldsymbol{W}^\top/\sqrt{d})^\top/m\right) \odot (\boldsymbol{X}\boldsymbol{X}^\top/d), \ \ \mathbf{\Theta}^{\mathrm{lin}1} := (\zeta^2 \boldsymbol{X}\boldsymbol{X}^\top + \nu^2 \mathbf{1}\mathbf{1}^\top)/d.$$

We have the following result that bounds the difference between $\mathbf{\Theta}_1(\boldsymbol{W}(0))$ and $\mathbf{\Theta}^{\mathrm{lin}1}$ in spectral norm:

**Proposition 3.4.** *With high probability over the random initialization $\boldsymbol{W}(0)$ and the training data $\boldsymbol{X}$, we have $\left\|\mathbf{\Theta}_1(\boldsymbol{W}(0)) - \mathbf{\Theta}^{\mathrm{lin}1}\right\| \lesssim \frac{n}{d^{1+\alpha}}$.*

Notice that $\left\|\mathbf{\Theta}^{\mathrm{lin}1}\right\| = \Theta(\frac{n}{d})$ according to Claim 3.1. Thus the bound $\frac{n}{d^{1+\alpha}}$ in Proposition 3.4 is of smaller order. We emphasize that it is important to bound the spectral norm rather than the more naive Frobenius norm, since the latter would give $\left\|\mathbf{\Theta}_1(\boldsymbol{W}(0)) - \mathbf{\Theta}^{\mathrm{lin}1}\right\|_F \gtrsim \frac{n}{d}$, which is not useful. (See Figure 5 for a numerical verification.)

To prove Proposition 3.4, we first use the matrix Bernstein inequality to bound the perturbation of $\mathbf{\Theta}_1(\boldsymbol{W}(0))$ around its expectation with respect to $\boldsymbol{W}(0)$: $\left\|\mathbf{\Theta}_1(\boldsymbol{W}(0)) - \mathbb{E}_{\boldsymbol{W}(0)}[\mathbf{\Theta}_1(\boldsymbol{W}(0))]\right\| \lesssim \frac{n}{d^{1+\alpha}}$. Then we perform an entry-wise Taylor expansion of $\mathbb{E}_{\boldsymbol{W}(0)}[\mathbf{\Theta}_1(\boldsymbol{W}(0))]$, and it turns out that the top-order terms exactly constitute $\mathbf{\Theta}^{\mathrm{lin}1}$, and the rest can be bounded in spectral norm by $\frac{n}{d^{1+\alpha}}$.

After proving Proposition 3.4, in order to prove Theorem 3.2, we carefully track (i) the prediction difference between $f_t^1$ and $f_t^{\mathrm{lin}1}$, (ii) how much the weight matrix $\boldsymbol{W}$ move away from initialization, as well as (iii) how much the NTK changes. To prove the guarantee on the entire data distribution we further need to utilize tools from generalization theory. The full proof is given in Appendix D.

## 3.2 Training the Second Layer

Next we consider training the second layer weights $\boldsymbol{v}$, which corresponds to $\eta_1 = 0$ in (4). Denote by $f_t^2 : \mathbb{R}^d \to \mathbb{R}$ the network at iteration $t$ in this case. We will show that training the second layer is also close to training a simple linear model $f^{\mathrm{lin}2}(\boldsymbol{x}; \boldsymbol{\gamma}) := \boldsymbol{\gamma}^\top \boldsymbol{\psi}_2(\boldsymbol{x})$ in the early phase, where:

$$\boldsymbol{\psi}_2(\boldsymbol{x}) := \begin{bmatrix} \frac{1}{\sqrt{d}}\zeta\boldsymbol{x} \\ \frac{1}{\sqrt{2d}}\nu \\ \vartheta_0 + \vartheta_1\left(\frac{\|\boldsymbol{x}\|}{\sqrt{d}} - 1\right) + \vartheta_2\left(\frac{\|\boldsymbol{x}\|}{\sqrt{d}} - 1\right)^2 \end{bmatrix}, \quad \begin{cases} \zeta \text{ and } \nu \text{ are defined in (5)}, \\ \vartheta_0 = \mathbb{E}[\phi(g)], \\ \vartheta_1 = \mathbb{E}[g\phi'(g)], \\ \vartheta_2 = \mathbb{E}[(\frac{1}{2}g^3 - g)\phi'(g)]. \end{cases} \quad (9)$$

As usual, this linear model is trained with GD starting from zero:

$$\boldsymbol{\gamma}(0) = \mathbf{0}_{d+2}, \quad \boldsymbol{\gamma}(t+1) = \boldsymbol{\gamma}(t) - \eta_2 \nabla_{\boldsymbol{\gamma}} \frac{1}{2n} \sum_{i=1}^n (f^{\mathrm{lin}2}(\boldsymbol{x}_i; \boldsymbol{\gamma}(t)) - y_i)^2. \quad (10)$$

We denote by $f_t^{\mathrm{lin}2}$ the resulting model at iteration $t$.

Note that strictly speaking $f^{\mathrm{lin}2}(\boldsymbol{x}; \boldsymbol{\gamma})$ is not a linear model in $\boldsymbol{x}$ because the feature map $\boldsymbol{\psi}_2(\boldsymbol{x})$ contains a nonlinear feature depending on $\|\boldsymbol{x}\|$ in its last coordinate. Because $\frac{\|\boldsymbol{x}\|}{\sqrt{d}} \approx 1$ under our data assumption according to Claim 3.1, its effect might often be invisible. However, we emphasize that in general the inclusion of this norm-dependent feature is necessary, for example when the target function explicitly depends on the norm of the input. We illustrate this in Section 3.4.

Similar to Theorem 3.2, our main theorem for training the second layer is the following:

**Theorem 3.5** (main theorem for training the second layer). *Let $\alpha \in (0, \frac{1}{4})$ be a fixed constant. Suppose $n \gtrsim d^{1+\alpha}$ and $\begin{cases} m \gtrsim d^{1+\alpha}, \text{ if } \mathbb{E}[\phi(g)] = 0 \\ m \gtrsim d^{2+\alpha}, \text{ otherwise} \end{cases}$. Suppose $\begin{cases} \eta_2 \ll d/\log n, \text{ if } \mathbb{E}[\phi(g)] = 0 \\ \eta_2 \ll 1, \quad\quad\quad\quad \text{ otherwise} \end{cases}$ and $\eta_1 = 0$. Then there exists a universal constant $c > 0$ such that with high probability, for all*

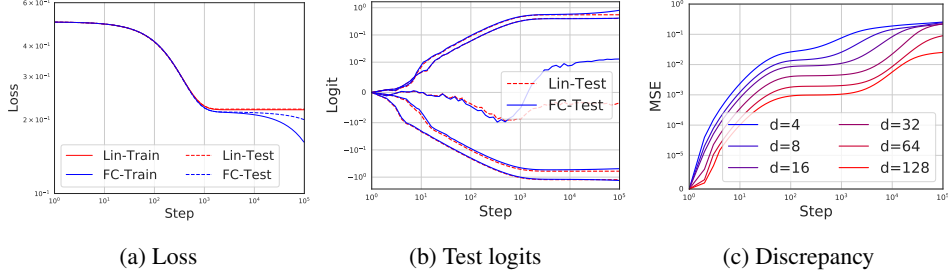

|  (a) Loss | (b) Test logits | (c) Discrepancy |

Figure 1: **Two-layer neural network learns a linear model early in training.** (a) Losses of a neural network and the corresponding linear model predicted by (11). Solid (dashed) lines represent the training (test) losses. We have $d = 50$, and use 20,000 training samples and 2,000 test samples. The neural network and the linear model are indistinguishable in the first 1,000 steps, after which linear learning finishes and the network continues to make progress. (b) Evolution of logits (i.e., outputs) of 5 random test examples. We see excellent agreement between the predictions of the neural network and the linear model in early time. (c) Discrepancy (in MSE) between the outputs of the network and the linear model for various values of $d$. As predicted, the discrepancy becomes smaller as $d$ increases.

$0 \le t \le T = c \cdot \frac{d \log d}{\eta_2}$ *simultaneously, we have*

$$\frac{1}{n} \sum_{i=1}^{n} \left( f_t^2(\boldsymbol{x}_i) - f_t^{\text{lin2}}(\boldsymbol{x}_i) \right)^2 \lesssim d^{-\Omega(\alpha)}, \quad \mathbb{E}_{\boldsymbol{x} \sim \mathcal{D}} \left[ \min\{ (f_t^2(\boldsymbol{x}) - f_t^{\text{lin2}}(\boldsymbol{x}))^2, 1 \} \right] \lesssim d^{-\Omega(\alpha)}.$$

Similar to Theorem 3.2, an important step in proving Theorem 3.5 is to prove that the NTK matrix for the second layer is close to the kernel for the linear model (9). Note that the theorem treats the case $\vartheta_0 = \mathbb{E}[\phi(g)] = 0$ differently. This is because when $\vartheta_0 \ne 0$, the second layer NTK has a large eigenvalue of size $\Theta(n)$, while when $\vartheta_0 = 0$, its largest eigenvalue is only $O(\frac{n \log n}{d})$.

We remark that if the data distribution is well-conditioned, we can also have a guarantee similar to Corollary 3.3.

### 3.3 Training Both Layers

Finally we consider the case where both layers are trained, in which $\eta_1 = \eta_2 = \eta > 0$ in (4). Since the NTK for training both layers is simply the sum of the first-layer NTK and the second-layer NTK, the corresponding linear model should have its kernel being the sum of the kernels for linear models (5) and (9), which can be derived easily:

$$f^{\text{lin}}(\boldsymbol{x}; \boldsymbol{\delta}) := \boldsymbol{\delta}^\top \boldsymbol{\psi}(\boldsymbol{x}), \quad \boldsymbol{\psi}(\boldsymbol{x}) := \begin{bmatrix} \sqrt{\frac{2}{d}} \zeta \boldsymbol{x} \\ \sqrt{\frac{3}{2d}} \nu \\ \vartheta_0 + \vartheta_1 (\frac{\|\boldsymbol{x}\|}{\sqrt{d}} - 1) + \vartheta_2 (\frac{\|\boldsymbol{x}\|}{\sqrt{d}} - 1)^2 \end{bmatrix}, \tag{11}$$

where the constants are from (9). Note that $\langle \boldsymbol{\psi}(\boldsymbol{x}), \boldsymbol{\psi}(\boldsymbol{x}') \rangle = \langle \boldsymbol{\psi}_1(\boldsymbol{x}), \boldsymbol{\psi}_1(\boldsymbol{x}') \rangle + \langle \boldsymbol{\psi}_2(\boldsymbol{x}), \boldsymbol{\psi}_2(\boldsymbol{x}') \rangle$.

Again, we can show that the neural network is close to the linear model (11) in early time. The guarantee is very similar to Theorems 3.2 and 3.5, so we defer the formal theorem to Appendix D; see Theorem D.1. Note that our result can be directly generalized to the case where $\eta_1 \ne \eta_2$, for which we just need to redefine the linear model using a weighted combination of the kernels for (5) and (9).

### 3.4 Empirical Verification

**Verifying the early-time agreement between neural network and linear model.** We verify our theory by training a two-layer neural network with erf activation and width 256 on synthetic data generated by $\boldsymbol{x} \sim \mathcal{N}(\boldsymbol{0}, \boldsymbol{I})$ and $y = \text{sign}(f^*(\boldsymbol{x}))$, where $f^*$ is a ground-truth two-layer erf network with width 5. In Figure 1a, we plot the training and test losses of the neural network (colored in

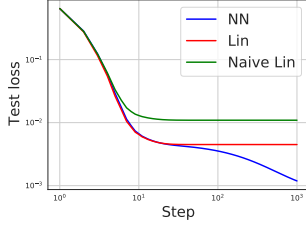

Figure 2: **The norm-dependent feature is necessary.** For the task of learning a norm-dependent function, test losses are shown for a neural network with ReLU activation, its corresponding linear model predicted by (11), and a naive linear model by resetting $\vartheta_1 = \vartheta_2 = 0$ in (11). Our predicted linear model is a much better approximation to the neural network than the naive linear model.

blue) and its corresponding linear model $f^{\text{lin}}$ (in red).[9] In the early training phase (up to 1,000 steps), the training/test losses of the network and the linear model are indistinguishable. After that, the optimal linear model is reached, and the network continues to make progress. In Figure 1b, we plot the evolution of the outputs (logits) of the network and the linear model on 5 random test examples, and we see excellent early-time agreement even on each individual sample. Finally, in Figure 1c, we vary the input dimension $d$, and for each case plot the mean squared error (MSE) of the discrepancies between the outputs of the network and the linear model. We see that the discrepancy indeed becomes smaller as $d$ increases, matching our theoretical prediction.

**The necessity of the norm-dependent feature.** We now illustrate the necessity of including the norm-dependent feature in (11) and (9) through an example of learning a norm-dependent function. We generate data from $\boldsymbol{x} \sim \mathcal{N}(\boldsymbol{0}, \boldsymbol{I})$ and $y = \frac{\|\boldsymbol{x}\|}{\sqrt{d}} + \text{ReLU}(\boldsymbol{a}^\top \boldsymbol{x})$ ($\|\boldsymbol{a}\| = O(1)$), and train a two-layer network with ReLU activation. We also train the corresponding linear model $f^{\text{lin}}$ (11) as well as a "naive linear model" which is identical to $f^{\text{lin}}$ except $\vartheta_1$ and $\vartheta_2$ are replaced with $0$. Figure 2 shows that $f^{\text{lin}}$ is indeed a much better approximation to the neural network than the naive linear model.

# 4 Extensions to Multi-Layer and Convolutional Neural Networks

In this section, we provide theoretical and empirical evidence supporting that the agreement between neural networks and linear models in the early phase of training may continue to hold for more complicated network architectures and datasets than what we analyzed in Section 3.

## 4.1 Theoretical Observations

**Multi-layer fully-connected (FC) neural networks.** For multi-layer FC networks, it was known that their infinite-width NTKs have the form $K(\boldsymbol{x}, \boldsymbol{x}') = h(\frac{\|\boldsymbol{x}\|^2}{d}, \frac{\|\boldsymbol{x}'\|^2}{d}, \frac{\langle \boldsymbol{x}, \boldsymbol{x}' \rangle}{d})$ ($\boldsymbol{x}, \boldsymbol{x}' \in \mathbb{R}^d$) for some function $h : \mathbb{R}^3 \to \mathbb{R}$ [Yang and Salman, 2019]. Let $\boldsymbol{\Theta}$ be the NTK matrix on the $n$ training data: $[\boldsymbol{\Theta}]_{i,j} = K(\boldsymbol{x}_i, \boldsymbol{x}_j)$. Under Assumption 3.1, we know from Claim 3.1 that $\frac{\|\boldsymbol{x}_i\|^2}{d} \approx 1$ and $\frac{\langle \boldsymbol{x}_i, \boldsymbol{x}_j \rangle}{d} \approx 0$ ($i \neq j$). Hence we can Taylor expand $h$ around $(1, 1, 0)$ for the off-diagonal entries of $\boldsymbol{\Theta}$ and around $(1, 1, 1)$ for the diagonal entries. Similar to our analysis of two-layer networks, we should be able to bound the higher-order components in the expansion, and only keep the simple ones like $\boldsymbol{X}\boldsymbol{X}^\top$, $\boldsymbol{1}\boldsymbol{1}^\top$, etc. This suggests that the early-time linear learning behavior which we showed for two-layer FC networks may persist in multi-layer FC networks.

**Convolutional neural networks (CNNs).** We consider a simple 1-dimensional CNN with one convolutional layer and without pooling (generalization to the commonly used 2-dimensional CNNs is straightforward):

$$f_{\text{CNN}}(\boldsymbol{x}; \boldsymbol{W}, \boldsymbol{V}) := \frac{1}{\sqrt{md}} \sum_{r=1}^{m} \boldsymbol{v}_r^\top \phi\left(\boldsymbol{w}_r * \boldsymbol{x}/\sqrt{q}\right). \tag{12}$$

Here $\boldsymbol{x} \in \mathbb{R}^d$ is the input, $\boldsymbol{W} = [\boldsymbol{w}_1, \ldots, \boldsymbol{w}_m]^\top \in \mathbb{R}^{m \times q}$ and $\boldsymbol{V} = [\boldsymbol{v}_1, \ldots, \boldsymbol{v}_m]^\top \in \mathbb{R}^{m \times d}$ contain the weights, where $m$ is the number of channels (or width), and $q \leq d$ is the filter size. All the weights are initialized i.i.d from $\mathcal{N}(0, 1)$. The convolution operator $*$ is defined as: for input $\boldsymbol{x} \in \mathbb{R}^d$

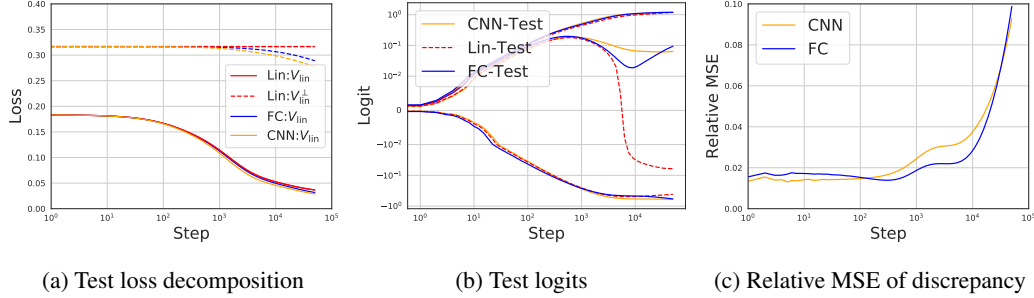

| (a) Test loss decomposition | (b) Test logits | (c) Relative MSE of discrepancy |

Figure 3: **Good agreement between 4-hidden-layer CNN/FC network and linear model on CIFAR-10 early in training.** (a) Decomposition of the test losses onto $V_{\text{lin}}$ (solid lines) and $V_{\text{lin}}^{\perp}$ (dashed lines) for CNN, FC and the corresponding linear model. (b) Three randomly selected test outputs for different models. (c) The relative MSE between the networks and the linear model. Note that we adjust the learning rates of CNN and FC so that their corresponding linear models are identical.

and filter $\boldsymbol{w} \in \mathbb{R}^q$, we have $\boldsymbol{w} * \boldsymbol{x} \in \mathbb{R}^d$ with $[\boldsymbol{w} * \boldsymbol{x}]_i := \sum_{j=1}^{q} [\boldsymbol{w}]_j [\boldsymbol{x}]_{i+j-1}$. We consider circular padding (as in Xiao et al. [2018], Li et al. [2019b]), so the indices in input should be understood as $[\boldsymbol{x}]_i = [\boldsymbol{x}]_{i+d}$.

We have the following result concerning the NTK of this CNN:

**Proposition 4.1.** *Let $\phi = \text{erf}$. Suppose $n \gtrsim d^{1+\alpha}$ and $q \gtrsim d^{\frac{1}{2}+2\alpha}$ for some constant $\alpha \in (0, \frac{1}{4})$. Consider $n$ datapoints $\boldsymbol{x}_1, \ldots, \boldsymbol{x}_n \overset{i.i.d.}{\sim} \text{Unif}(\{\pm 1\}^d)$. Then the corresponding NTK matrix $\boldsymbol{\Theta}_{\text{CNN}} \in \mathbb{R}^{n \times n}$ of the CNN (12) in the infinite-width limit ($m \to \infty$) satisfies $\left\| \boldsymbol{\Theta}_{\text{CNN}} - 2\zeta^2 \boldsymbol{X}\boldsymbol{X}^{\top}/d \right\| \lesssim \frac{n}{d^{1+\alpha}}$ with high probability, where $\zeta = \mathbb{E}[\phi'(g)]$.*

The proof is given in Appendix E. The above result shows that the NTK of a CNN can also be close to the (scaled) data kernel, which implies the linear learning behavior in the early time of training the CNN. Our empirical results will show that this behavior can even persist to multi-layer CNNs and real data beyond our analysis.

## 4.2 Empirical Results

We perform experiments on a binary classification task from CIFAR-10 ("cats" vs "horses") using a multi-layer FC network and a CNN. The numbers of training and test data are 10,000 and 2,000. The original size of the images is $32 \times 32 \times 3$, and we down-sample the images into size $8 \times 8 \times 3$ using a $4 \times 4$ average pooling. Then we train a 4-hidden-layer FC net and a 4-hidden-layer CNN with erf activation. To have finer-grained examination of the evolution of the losses, we decompose the residual of the predictions on test data (namely, $f_t(\boldsymbol{x}) - y$ for all test data collected as a vector in $\mathbb{R}^{2000}$) onto $V_{\text{lin}}$, the space spanned by the inputs (of dimension $d = 192$), and its complement $V_{\text{lin}}^{\perp}$ (of dimension $2000 - d$). For both networks, we observe in Figure 3a that the test losses of the networks and the linear model are almost identical up to 1,000 steps, and the networks start to make progress in $V_{\text{lin}}^{\perp}$ after that. In Figure 3b we plot the logit evolution of 3 random test datapoints and again observe good agreement in early time. In Figure 3c, we plot the relative MSE between the network and the linear model (i.e., $\mathbb{E}_{\boldsymbol{x}} \| f_t(\boldsymbol{x}) - f_t^{\text{lin}}(\boldsymbol{x}) \|^2 / \mathbb{E}_{\boldsymbol{x}} \| f_t^{\text{lin}}(\boldsymbol{x}) \|^2$ evaluated on test data). We observe that this quantity for either network is small in the first 1,000 steps and grows afterwards. The detailed setup and additional results for full-size CIFAR-10 and MNIST are deferred to Appendix A.

## 5 Conclusion

This work gave a novel theoretical result rigorously showing that gradient descent on a neural network learns a simple linear function in the early phase. While we mainly focused on two-layer fully-connected neural networks, we further provided theoretical and empirical evidence suggesting that this phenomenon continues to exist in more complicated models. Formally extending our result to those settings is a direction of future work. Another interesting direction is to study the dynamics of neural networks after the initial linear learning phase.

## Broader Impact

This work is theoretical and does not present any foreseeable societal consequence.

## Acknowledgments and Disclosure of Funding

WH was supported by NSF, ONR, Simons Foundation, Schmidt Foundation, Amazon Research, DARPA and SRC.

## Footnotes

[5]The scaling factors $\frac{1}{\sqrt{d}}$ and $\frac{1}{\sqrt{m}}$ are due to the NTK parameterization such that the weights can be initialized from $\mathcal{N}(0, 1)$. The standard parameterization can also be equivalently realized with the NTK parameterization by properly setting different learning rates in different layers [Lee et al., 2019], which we do allow here.

[6] Our results also hold for $\mathcal{N}(0, 1)$ initialization in the second layer. Here we use $\mathsf{Unif}(\{\pm 1\})$ for simplicity.

[7] Recall that a zero-mean random variable $X$ is $\sigma^2$-subgaussian if $\mathbb{E}[\exp(sX)] \leq \exp(\sigma^2 s^2/2) \, (\forall s \in \mathbb{R})$.

[8] We define $\phi'(0) = 1$ in this case.

[9]For $\phi = \text{erf}$, we have $\vartheta_0 = \vartheta_1 = \vartheta_2 = 0$, so $f^{\text{lin}}$ in (11) is a linear model in $\boldsymbol{x}$ without the nonlinear feature.

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
