[Supplementary Material]

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

 $\mathbf{W}(0)$ and the training data $\mathbf{X}$, we have $\left\|\mathbf{\Theta}_1(\mathbf{W}(0)) - \mathbf{\Theta}^{\text{lin1}}\right\| \lesssim \frac{n}{d^{1+\alpha}}$.*

Notice that $\left\|\mathbf{\Theta}^{\text{lin1}}\right\| = \Theta(\frac{n}{d})$ according to Claim 3.1. Thus the bound $\frac{n}{d^{1+\alpha}}$ in Proposition 3.4 is of smaller order. We emphasize that it is important to bound the spectral norm rather than the more naive Frobenius norm, since the latter would give $\left\|\mathbf{\Theta}_1(\mathbf{W}(0)) - \mathbf{\Theta}^{\text{lin1}}\right\|_F \gtrsim \frac{n}{d}$, which is not useful. (See Figure 5 for a numerical verification.)

To prove Proposition 3.4, we first use the matrix Bernstein inequality to bound the perturbation of $\mathbf{\Theta}_1(\mathbf{W}(0))$ around its expectation with respect to $\mathbf{W}(0)$: $\left\|\mathbf{\Theta}_1(\mathbf{W}(0)) - \mathbb{E}_{\mathbf{W}(0)}[\mathbf{\Theta}_1(\mathbf{W}(0))]\right\| \lesssim \frac{n}{d^{1+\alpha}}$. Then we perform an entry-wise Taylor expansion of $\mathbb{E}_{\mathbf{W}(0)}[\mathbf{\Theta}_1(\mathbf{W}(0))]$, and it turns out that the top-order terms exactly constitute $\mathbf{\Theta}^{\text{lin1}}$, and the rest can be bounded in spectral norm by $\frac{n}{d^{1+\alpha}}$.

After proving Proposition 3.4, in order to prove Theorem 3.2, we carefully track (i) the prediction difference between $f_t^1$ and $f_t^{\text{lin1}}$, (ii) how much the weight matrix $\mathbf{W}$ move away from initialization, as well as (iii) how much the NTK changes. To prove the guarantee on the entire data distribution we further need to utilize tools from generalization theory. The full proof is given in Appendix D.

## 3.2 Training the Second Layer

Next we consider training the second layer weights $\mathbf{v}$, which corresponds to $\eta_1 = 0$ in (4). Denote by $f_t^2 : \mathbb{R}^d \to \mathbb{R}$ the network at iteration $t$ in this case. We will show that training the second layer is also close to training a simple linear model $f^{\text{lin2}}(\mathbf{x}; \boldsymbol{\gamma}) := \boldsymbol{\gamma}^\top \boldsymbol{\psi}_2(\mathbf{x})$ in the early phase, where:

$$\boldsymbol{\psi}_2(\mathbf{x}) := \begin{bmatrix} \frac{1}{\sqrt{d}}\zeta\mathbf{x} \\ \frac{1}{\sqrt{2d}}\nu \\ \vartheta_0 + \vartheta_1(\frac{\|\mathbf{x}\|}{\sqrt{d}} - 1) + \vartheta_2(\frac{\|\mathbf{x}\|}{\sqrt{d}} - 1)^2 \end{bmatrix}, \quad \begin{cases} \zeta \text{ and } \nu \text{ are defined in (5),} \\ \vartheta_0 = \mathbb{E}[\phi(g)], \\ \vartheta_1 = \mathbb{E}[g\phi'(g)], \\ \vartheta_2 = \mathbb{E}[(\frac{1}{2}g^3 - g)\phi'(g)]. \end{cases} \tag{9}$$

As usual, this linear model is trained with GD starting from zero:

$$\boldsymbol{\gamma}(0) = \mathbf{0}_{d+2}, \quad \boldsymbol{\gamma}(t+1) = \boldsymbol{\gamma}(t) - \eta_2 \nabla_{\boldsymbol{\gamma}} \frac{1}{2n} \sum_{i=1}^n (f^{\text{lin2}}(\mathbf{x}_i; \boldsymbol{\gamma}(t)) - y_i)^2. \tag{10}$$

We denote by $f_t^{\text{lin2}}$ the resulting model at iteration $t$.

Note that strictly speaking $f^{\text{lin2}}(\mathbf{x}; \boldsymbol{\gamma})$ is not a linear model in $\mathbf{x}$ because the feature map $\boldsymbol{\psi}_2(\mathbf{x})$ contains a nonlinear feature depending on $\|\mathbf{x}\|$ in its last coordinate. Because $\frac{\|\mathbf{

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

 $x \sim \mathcal{N}(\mathbf{0}, I)$ and $y = \frac{\|x\|}{\sqrt{d}} + \text{ReLU}(a^\top x)$ ($\|a\| = O(1)$), and train a two-layer network with ReLU activation. We also train the corresponding linear model $f^{\text{lin}}$ (11) as well as a "naive linear model" which is identical to $f^{\text{lin}}$ except $\vartheta_1$ and $\vartheta_2$ are replaced with $0$. Figure 2 shows that $f^{\text{lin}}$ is indeed a much better approximation to the neural network than the naive linear model.

# 4 Extensions to Multi-Layer and Convolutional Neural Networks

In this section, we provide theoretical and empirical evidence supporting that the agreement between neural networks and linear models in the early phase of training may continue to hold for more complicated network architectures and datasets than what we analyzed in Section 3.

## 4.1 Theoretical Observations

**Multi-layer fully-connected (FC) neural networks.** For multi-layer FC networks, it was known that their infinite-width NTKs have the form $K(x, x') = h\left(\frac{\|x\|^2}{d}, \frac{\|x'\|^2}{d}, \frac{\langle x, x' \rangle}{d}\right)$ ($x, x' \in \mathbb{R}^d$) for some function $h : \mathbb{R}^3 \to \mathbb{R}$ [Yang and Salman, 2019]. Let $\Theta$ be the NTK matrix on the $n$ training data: $[\Theta]_{i,j} = K(x_i, x_j)$. Under Assumption 3.1, we know from Claim 3.1 that $\frac{\|x_i\|^2}{d} \approx 1$ and $\frac{\langle x_i, x_j \rangle}{d} \approx 0$ ($i \neq j$). Hence we can Taylor expand $h$ around $(1, 1, 0)$ for the off-diagonal entries of $\Theta$ and around $(1, 1, 1)$ for the diagonal entries. Similar to our analysis of two-layer networks, we should be able to bound the higher-order components in the expansion, and only keep the simple ones like $XX^\top$, $\mathbf{1}\mathbf{1}^\top$, etc. This suggests that the early-time linear learning behavior which we showed for two-layer FC networks may persist in multi-layer FC networks.

**Convolutional neural networks (CNNs).** We consider a simple 1-dimensional CNN with one convolutional layer and without pooling (generalization to the commonly used 2-dimensional CNNs is straightforward):

$$f_{\text{CNN}}(x; W, V) := \frac{1}{\sqrt{md}} \sum_{r=1}^{m} v_r^\top \phi\left(w_r * x / \sqrt{q}\right). \tag{12}$$

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

# A  Experiment Setup and Additional Plots

We provide additional plots and describe additional experiment details in this section.

In Figure 4, we repeat the same experiments in Figure 3 on the full-size ($32 \times 32 \times 3$) CIFAR-10 as well as MNIST datasets, using the same 4-hidden-layer FC and CNN architectures. For both datasets we take two classes and perform binary classification. We see very good early-time agreement except for CNN on CIFAR-10, where the agreement only lasts for a shorter time.

For the experiments in Figures 3 and 4, the FC network has width $512$ in each of the $4$ hidden layers, and the CNN uses circular padding and has $256$ channels in each of the $4$ hidden layers. For CIFAR-10 and MNIST images, we use standard data pre-processing, i.e., normalizing each image to have zero mean and unit variance. To ensure the initial outputs are always $0$, we subtract the function output at initialization for each datapoint (as discussed in Section 3). We train and test using the $\ell_2$ loss with $\pm 1$ labels. We use vanilla stochastic gradient descent with batch size $500$, and choose a small learning rate (roughly $\frac{0.01}{\|\mathrm{NTK}\|}$) so that we can better observe early time of training (similar to Nakkiran et al. [2019]).

We use the Neural Tangents Library [Novak et al., 2019] and JAX [Bradbury et al., 2018] for our experiments.

(a) Test loss  (b) Test logits  (c) Relative MSE of discrepancy

(d) Test loss decomposition  (e) Test logits  (f) Relative MSE of discrepancy

Figure 4: Replication of Figure 3 on full-size CIFAR-10 (top row) and MNIST (bottom row). In Figure 4a, there is no projection onto $V_{\mathrm{lin}}^{\perp}$ because the data dimension $32 \times 32 \times 3$ is larger than the number of test data 2,000.

# B  Additional Notation and Lemmas

We introduce some additional notation and lemmas that will be used in the proofs.

We use $\tilde{O}(\cdot)$ to hide poly-logarithmic factors in $n$ (the number of training datapoints). Denote by $\mathbb{1}_{\{E\}}$ the indicator function for an event $E$. For a vector $\boldsymbol{a}$, we let $\mathrm{diag}(\boldsymbol{a})$ be a diagonal matrix whose diagonal entries constitute $\boldsymbol{a}$. For a matrix $\boldsymbol{A}$, we use $\mathrm{vec}(\boldsymbol{A})$ to denote the vectorization of $\boldsymbol{A}$ in row-first order.

For a square matrix $\boldsymbol{A}$, we denote its diagonal and off-diagonal parts as $\boldsymbol{A}_{\mathrm{diag}}$ and $\boldsymbol{A}_{\mathrm{off}}$, respectively. Namely, we have $\boldsymbol{A} = \boldsymbol{A}_{\mathrm{diag}} + \boldsymbol{A}_{\mathrm{off}}$, where $[\boldsymbol{A}_{\mathrm{diag}}]_{i,j} = [\boldsymbol{A}]_{i,j} \, \mathbb{1}_{\{i=j\}}$ and $[\boldsymbol{A}_{\mathrm{off}}]_{i,j} = [\boldsymbol{A}]_{i,j} \, \mathbb{1}_{\{i \neq j\}}$. Equivalently, $\boldsymbol{A}_{\mathrm{diag}} = \boldsymbol{A} \odot \boldsymbol{I}$ and $\boldsymbol{A}_{\mathrm{off}} = \boldsymbol{A} \odot (\mathbf{1}\mathbf{1}^{\top} - \boldsymbol{I})$.

**Lemma B.1.** *For any matrix $\boldsymbol{A}$ and a submatrix $\boldsymbol{A}_1$ of $\boldsymbol{A}$, we have $\|\boldsymbol{A}_1\| \leq \|\boldsymbol{A}\|$.*

*Proof.* For simplicity we assume that $\boldsymbol{A}_1$ is in the top-left corner of $\boldsymbol{A}$, i.e. $\boldsymbol{A} = \begin{bmatrix} \boldsymbol{A}_1 & \boldsymbol{A}_2 \\ \boldsymbol{A}_3 & \boldsymbol{A}_4 \end{bmatrix}$. The same proof works when $\boldsymbol{A}_1$ is any other submatrix of $\boldsymbol{A}$.

By the definition of spectral norm, we have

$$
\begin{aligned}
\|\boldsymbol{A}\| &= \max_{\|\boldsymbol{x}\|=\|\boldsymbol{y}\|=1} \boldsymbol{x}^\top \boldsymbol{A} \boldsymbol{y} \\
&= \max_{\|\boldsymbol{x}\|=\|\boldsymbol{y}\|=1} \boldsymbol{x}^\top \begin{bmatrix} \boldsymbol{A}_1 & \boldsymbol{A}_2 \\ \boldsymbol{A}_3 & \boldsymbol{A}_4 \end{bmatrix} \boldsymbol{y} \\
&\geq \max_{\|\boldsymbol{x}_1\|=\|\boldsymbol{y}_1\|=1} [\boldsymbol{x}_1^\top, \boldsymbol{0}^\top] \begin{bmatrix} \boldsymbol{A}_1 & \boldsymbol{A}_2 \\ \boldsymbol{A}_3 & \boldsymbol{A}_4 \end{bmatrix} \begin{bmatrix} \boldsymbol{y}_1 \\ \boldsymbol{0} \end{bmatrix} \\
&= \max_{\|\boldsymbol{x}_1\|=\|\boldsymbol{y}_1\|=1} \boldsymbol{x}_1^\top \boldsymbol{A}_1 \boldsymbol{y}_1 \\
&= \|\boldsymbol{A}_1\|. \qquad\qquad\qquad\qquad\qquad\qquad\qquad \square
\end{aligned}
$$

**Lemma B.2.** *For any square matrix $\boldsymbol{A}$, we have $\|\boldsymbol{A}_{\mathrm{diag}}\| \leq \|\boldsymbol{A}\|$ and $\|\boldsymbol{A}_{\mathrm{off}}\| \leq 2\|\boldsymbol{A}\|$.*

*Proof.* From Lemma B.1 we know that $\left|[\boldsymbol{A}]_{i,i}\right| \leq \|\boldsymbol{A}\|$ for all $i$ since $[\boldsymbol{A}]_{i,i}$ can be viewed as a submatrix of $\boldsymbol{A}$. Thus we have

$$
\|\boldsymbol{A}_{\mathrm{diag}}\| = \max_i \left|[\boldsymbol{A}]_{i,i}\right| \leq \|\boldsymbol{A}\|.
$$

It follows that

$$
\|\boldsymbol{A}_{\mathrm{off}}\| = \|\boldsymbol{A} - \boldsymbol{A}_{\mathrm{diag}}\| \leq \|\boldsymbol{A}\| + \|\boldsymbol{A}_{\mathrm{diag}}\| \leq 2\|\boldsymbol{A}\|. \qquad\qquad \square
$$

**Lemma B.3** (Schur [1911])**.** *For any two positive semidefinite matrices $\boldsymbol{A}, \boldsymbol{B}$, we have*

$$
\|\boldsymbol{A} \odot \boldsymbol{B}\| \leq \|\boldsymbol{A}\| \cdot \max_i [\boldsymbol{B}]_{i,i}.
$$

# C  General Result on the Closeness between Two Dynamics

We present a general result that shows how the GD trajectory for a non-linear least squares problem can be simulated by a linear one. Later we will specialize this result to the settings considered in the paper.

We consider an objective function of the form:

$$
F(\boldsymbol{\theta}) = \frac{1}{2n} \|\boldsymbol{f}(\boldsymbol{\theta}) - \boldsymbol{y}\|^2,
$$

where $\boldsymbol{f} : \mathbb{R}^N \mapsto \mathbb{R}^n$ is a general differentiable function, and $\boldsymbol{y} \in \mathbb{R}^n$ satisfies $\|\boldsymbol{y}\| \leq \sqrt{n}$. We denote by $\boldsymbol{J} : \mathbb{R}^N \mapsto \mathbb{R}^{n \times N}$ the Jacobian map of $\boldsymbol{f}$. Then starting from some $\boldsymbol{\theta}(0) \in \mathbb{R}^N$, the GD updates for minimizing $F$ can be written as:

$$
\boldsymbol{\theta}(t+1) = \boldsymbol{\theta}(t) - \eta \nabla F(\boldsymbol{\theta}(t)) = \boldsymbol{\theta}(t) - \frac{1}{n} \eta \boldsymbol{J}(\boldsymbol{\theta}(t))^\top (\boldsymbol{f}(\boldsymbol{\theta}(t)) - \boldsymbol{y}).
$$

Consider another linear least squares problem:

$$
G(\boldsymbol{\omega}) = \frac{1}{2n} \|\boldsymbol{\Phi}\boldsymbol{\omega} - \boldsymbol{y}\|^2,
$$

where $\boldsymbol{\Phi} \in \mathbb{R}^{n \times M}$ is a fixed matrix. Its GD dynamics started from $\boldsymbol{\omega}(0) \in \mathbb{R}^M$ can be written as:

$$
\boldsymbol{\omega}(t+1) = \boldsymbol{\omega}(t) - \eta \nabla G(\boldsymbol{\omega}(t)) = \boldsymbol{\omega}(t) - \frac{1}{n} \eta \boldsymbol{\Phi}^\top (\boldsymbol{\Phi}\boldsymbol{\omega}(t) - \boldsymbol{y}).
$$

Let $\boldsymbol{K} := \boldsymbol{\Phi}\boldsymbol{\Phi}^\top$, and let

$$
\boldsymbol{u}(t) := \boldsymbol{f}(\boldsymbol{\theta}(t)),
$$

$$\boldsymbol{u}^{\mathrm{lin}}(t) := \boldsymbol{\Phi}\boldsymbol{\omega}(t),$$

which stand for the predictions of these two models at iteration $t$.

The linear dynamics admit a very simple analytical form, summarized below.

**Claim C.1.** *For all $t \geq 0$ we have $\boldsymbol{u}^{\mathrm{lin}}(t) - \boldsymbol{y} = \left(\boldsymbol{I} - \frac{1}{n}\eta\boldsymbol{K}\right)^t (\boldsymbol{u}^{\mathrm{lin}}(0) - \boldsymbol{y})$. As a consequence, if $\eta \leq \frac{2n}{\|\boldsymbol{K}\|}$, then we have $\left\|\boldsymbol{u}^{\mathrm{lin}}(t) - \boldsymbol{y}\right\| \leq \left\|\boldsymbol{u}^{\mathrm{lin}}(0) - \boldsymbol{y}\right\|$ for all $t \geq 0$.*

*Proof.* By definition we have $\boldsymbol{u}^{\mathrm{lin}}(t+1) = \boldsymbol{u}^{\mathrm{lin}}(t) - \frac{1}{n}\eta\boldsymbol{K}(\boldsymbol{u}^{\mathrm{lin}}(t) - \boldsymbol{y})$, which implies $\boldsymbol{u}^{\mathrm{lin}}(t+1) - \boldsymbol{y} = \left(\boldsymbol{I} - \frac{1}{n}\eta\boldsymbol{K}\right)(\boldsymbol{u}^{\mathrm{lin}}(t) - \boldsymbol{y})$. Thus the first statement follows directly. Then the second statement can be proved by noting that $\left\|\boldsymbol{I} - \frac{1}{n}\eta\boldsymbol{K}\right\| \leq 1$ when $\eta \leq \frac{2n}{\|\boldsymbol{K}\|}$. $\square$

We make the following assumption that connects these two problems:

**Assumption C.1.** *There exist $0 < \epsilon < \|\boldsymbol{K}\|, R > 0$ such that for any $\boldsymbol{\theta}, \boldsymbol{\theta}' \in \mathbb{R}^N$, as long as $\|\boldsymbol{\theta} - \boldsymbol{\theta}(0)\| \leq R$ and $\|\boldsymbol{\theta}' - \boldsymbol{\theta}(0)\| \leq R$, we have*

$$\left\|\boldsymbol{J}(\boldsymbol{\theta})\boldsymbol{J}(\boldsymbol{\theta}')^\top - \boldsymbol{K}\right\| \leq \epsilon.$$

Based on the above assumption, we have the following theorem showing the agreement between $\boldsymbol{u}(t)$ and $\boldsymbol{u}^{\mathrm{lin}}(t)$ as well as the parameter boundedness in early time.

**Theorem C.2.** *Suppose that the initializations are chosen so that $\boldsymbol{u}(0) = \boldsymbol{u}^{\mathrm{lin}}(0) = \boldsymbol{0}$, and that the learning rate satisfies $\eta \leq \frac{n}{\|\boldsymbol{K}\|}$. Suppose that Assumption C.1 is satisfied with $R^2\epsilon < n$. Then there exists a universal constant $c > 0$ such that for all $0 \leq t \leq c\frac{R^2}{\eta}$:*

- *(closeness of predictions) $\left\|\boldsymbol{u}(t) - \boldsymbol{u}^{\mathrm{lin}}(t)\right\| \lesssim \frac{\eta t\epsilon}{\sqrt{n}}$;*

- *(boundedness of parameter movement) $\|\boldsymbol{\theta}(t) - \boldsymbol{\theta}(0)\| \leq R, \|\boldsymbol{\omega}(t) - \boldsymbol{\omega}(0)\| \leq R$.*

*Proof.* We first prove the first two properties, and will prove the last property $\|\boldsymbol{\omega}(t) - \boldsymbol{\omega}(0)\| \leq R$ at the end.

We use induction to prove $\left\|\boldsymbol{u}(t) - \boldsymbol{u}^{\mathrm{lin}}(t)\right\| \lesssim \frac{\eta t\epsilon}{\sqrt{n}}$ and $\|\boldsymbol{\theta}(t) - \boldsymbol{\theta}(0)\| \leq R$. For $t = 0$, these statements are trivially true. Now suppose for some $1 \leq t \leq c\frac{R^2}{\eta}$ we have $\left\|\boldsymbol{u}(\tau) - \boldsymbol{u}^{\mathrm{lin}}(\tau)\right\| \lesssim \frac{\eta\tau\epsilon}{\sqrt{n}}$ and $\|\boldsymbol{\theta}(\tau) - \boldsymbol{\theta}(0)\| \leq R$ for $\tau = 0, 1, \ldots, t - 1$. We will now prove $\left\|\boldsymbol{u}(t) - \boldsymbol{u}^{\mathrm{lin}}(t)\right\| \lesssim \frac{\eta t\epsilon}{\sqrt{n}}$ and $\|\boldsymbol{\theta}(t) - \boldsymbol{\theta}(0)\| \leq R$ under these induction hypotheses.

Notice that from $\left\|\boldsymbol{u}(\tau) - \boldsymbol{u}^{\mathrm{lin}}(\tau)\right\| \lesssim \frac{\eta\tau\epsilon}{\sqrt{n}} \leq \frac{cR^2\epsilon}{\sqrt{n}} \lesssim \sqrt{n}$ and Claim C.1 we know $\|\boldsymbol{u}(\tau) - \boldsymbol{y}\| \lesssim \sqrt{n}$ for all $\tau < t$.

**Step 1: proving $\|\boldsymbol{\theta}(t) - \boldsymbol{\theta}(0)\| \leq R$.** We define

$$\boldsymbol{J}(\boldsymbol{\theta} \to \boldsymbol{\theta}') := \int_0^1 \boldsymbol{J}(\boldsymbol{\theta} + x(\boldsymbol{\theta}' - \boldsymbol{\theta}))dx.$$

We first prove $\|\boldsymbol{\theta}(t-1) - \boldsymbol{\theta}(0)\| \leq \frac{R}{2}$. If $t = 1$, this is trivially true. Now we assume $t \geq 2$. For each $0 \leq \tau < t - 1$, by the fundamental theorem for line integrals we have

$$\boldsymbol{u}(\tau + 1) - \boldsymbol{u}(\tau) = \boldsymbol{J}(\boldsymbol{\theta}(\tau) \to \boldsymbol{\theta}(\tau + 1)) \cdot (\boldsymbol{\theta}(\tau + 1) - \boldsymbol{\theta}(\tau))$$
$$= -\frac{\eta}{n}\boldsymbol{J}(\boldsymbol{\theta}(\tau) \to \boldsymbol{\theta}(\tau + 1))\boldsymbol{J}(\boldsymbol{\theta}(\tau))^\top (\boldsymbol{u}(\tau) - \boldsymbol{y}).$$

Let $\boldsymbol{E}(\tau) := \boldsymbol{J}(\boldsymbol{\theta}(\tau) \to \boldsymbol{\theta}(\tau + 1))\boldsymbol{J}(\boldsymbol{\theta}(\tau))^\top - \boldsymbol{K}$. Since $\|\boldsymbol{\theta}(\tau) - \boldsymbol{\theta}(0)\| \leq R$ and $\|\boldsymbol{\theta}(\tau + 1) - \boldsymbol{\theta}(0)\| \leq R$, from Assumption C.1 we know that $\|\boldsymbol{E}(\tau)\| \leq \epsilon$. We can write

$$\boldsymbol{u}(\tau + 1) - \boldsymbol{y} = \left(\boldsymbol{I} - \frac{\eta}{n}\boldsymbol{J}(\boldsymbol{\theta}(\tau) \to \boldsymbol{\theta}(\tau + 1))\boldsymbol{J}(\boldsymbol{\theta}(\tau))^\top\right)(\boldsymbol{u}(\tau) - \boldsymbol{y})$$
$$= \left(\boldsymbol{I} - \frac{\eta}{n}\boldsymbol{K}\right)(\boldsymbol{u}(\tau) - \boldsymbol{y}) - \frac{\eta}{n}\boldsymbol{E}(\tau)(\boldsymbol{u}(\tau) - \boldsymbol{y}). \tag{13}$$

It follows that

$$\|\boldsymbol{u}(\tau+1)-\boldsymbol{y}\|^2$$

$$\leq \left\|\left(\boldsymbol{I}-\frac{\eta}{n}\boldsymbol{K}\right)(\boldsymbol{u}(\tau)-\boldsymbol{y})\right\|^2 + 2\left\|\left(\boldsymbol{I}-\frac{\eta}{n}\boldsymbol{K}\right)(\boldsymbol{u}(\tau)-\boldsymbol{y})\right\| \cdot \left\|\frac{\eta}{n}\boldsymbol{E}(\tau)(\boldsymbol{u}(\tau)-\boldsymbol{y})\right\|$$

$$\quad + \left\|\frac{\eta}{n}\boldsymbol{E}(\tau)(\boldsymbol{u}(\tau)-\boldsymbol{y})\right\|^2$$

$$\leq \left\|\left(\boldsymbol{I}-\frac{\eta}{n}\boldsymbol{K}\right)(\boldsymbol{u}(\tau)-\boldsymbol{y})\right\|^2 + O\left(\sqrt{n}\cdot\frac{\eta}{n}\epsilon\sqrt{n} + \left(\frac{\eta}{n}\epsilon\sqrt{n}\right)^2\right)$$

$$= \left\|\left(\boldsymbol{I}-\frac{\eta}{n}\boldsymbol{K}\right)(\boldsymbol{u}(\tau)-\boldsymbol{y})\right\|^2 + O(\eta\epsilon) \qquad\qquad (\eta\epsilon\lesssim n)$$

$$= \|\boldsymbol{u}(\tau)-\boldsymbol{y}\|^2 - \frac{2\eta}{n}(\boldsymbol{u}(\tau)-\boldsymbol{y})^\top\boldsymbol{K}(\boldsymbol{u}(\tau)-\boldsymbol{y}) + \frac{\eta^2}{n^2}\|\boldsymbol{K}(\boldsymbol{u}(\tau)-\boldsymbol{y})\|^2 + O(\eta\epsilon)$$

$$\leq \|\boldsymbol{u}(\tau)-\boldsymbol{y}\|^2 - \frac{2\eta}{n}(\boldsymbol{u}(\tau)-\boldsymbol{y})^\top\boldsymbol{K}(\boldsymbol{u}(\tau)-\boldsymbol{y}) + \frac{\eta^2}{n^2}\|\boldsymbol{K}\|\cdot\left\|\boldsymbol{K}^{1/2}(\boldsymbol{u}(\tau)-\boldsymbol{y})\right\|^2 + O(\eta\epsilon)$$

$$\leq \|\boldsymbol{u}(\tau)-\boldsymbol{y}\|^2 - \frac{\eta}{n}(\boldsymbol{u}(\tau)-\boldsymbol{y})^\top\boldsymbol{K}(\boldsymbol{u}(\tau)-\boldsymbol{y}) + O(\eta\epsilon). \qquad (\tfrac{\eta^2\|\boldsymbol{K}\|}{n^2}\leq\tfrac{\eta}{n})$$

On the other hand, we have

$$\|\boldsymbol{\theta}(\tau+1)-\boldsymbol{\theta}(\tau)\|^2$$

$$= \frac{\eta^2}{n^2}\left\|\boldsymbol{J}(\boldsymbol{\theta}(\tau))^\top(\boldsymbol{u}(\tau)-\boldsymbol{y})\right\|^2$$

$$= \frac{\eta^2}{n^2}(\boldsymbol{u}(\tau)-\boldsymbol{y})^\top\boldsymbol{J}(\boldsymbol{\theta}(\tau))\boldsymbol{J}(\boldsymbol{\theta}(\tau))^\top(\boldsymbol{u}(\tau)-\boldsymbol{y}) \qquad\qquad (14)$$

$$\leq \frac{\eta^2}{n^2}\left((\boldsymbol{u}(\tau)-\boldsymbol{y})^\top\boldsymbol{K}(\boldsymbol{u}(\tau)-\boldsymbol{y}) + \|\boldsymbol{u}(\tau)-\boldsymbol{y}\|^2\left\|\boldsymbol{J}(\boldsymbol{\theta}(\tau))\boldsymbol{J}(\boldsymbol{\theta}(\tau))^\top-\boldsymbol{K}\right\|\right)$$

$$\leq \frac{\eta^2}{n^2}\left((\boldsymbol{u}(\tau)-\boldsymbol{y})^\top\boldsymbol{K}(\boldsymbol{u}(\tau)-\boldsymbol{y}) + O(n\epsilon)\right).$$

Combining the above two inequalities, we obtain

$$\|\boldsymbol{u}(\tau+1)-\boldsymbol{y}\|^2 - \|\boldsymbol{u}(\tau)-\boldsymbol{y}\|^2$$

$$\leq -\frac{\eta}{n}\left(\frac{n^2}{\eta^2}\|\boldsymbol{\theta}(\tau+1)-\boldsymbol{\theta}(\tau)\|^2 - O(n\epsilon)\right) + O(\eta\epsilon)$$

$$= -\frac{n}{\eta}\|\boldsymbol{\theta}(\tau+1)-\boldsymbol{\theta}(\tau)\|^2 + O(\eta\epsilon).$$

Taking sum over $\tau = 0, \ldots, t-2$, we get

$$\|\boldsymbol{u}(t-1)-\boldsymbol{y}\|^2 - \|\boldsymbol{u}(0)-\boldsymbol{y}\|^2 \leq -\frac{n}{\eta}\sum_{\tau=0}^{t-2}\|\boldsymbol{\theta}(\tau+1)-\boldsymbol{\theta}(\tau)\|^2 + O(\eta t\epsilon),$$

which implies

$$\frac{n}{\eta}\sum_{\tau=0}^{t-2}\|\boldsymbol{\theta}(\tau+1)-\boldsymbol{\theta}(\tau)\|^2 \leq \|\boldsymbol{y}\|^2 + O(\eta t\epsilon) \leq \|\boldsymbol{y}\|^2 + O(R^2\epsilon) = O(n).$$

Then by the Cauchy-Schwartz inequality we have

$$\|\boldsymbol{\theta}(t-1)-\boldsymbol{\theta}(0)\| \leq \sum_{\tau=0}^{t-2}\|\boldsymbol{\theta}(\tau+1)-\boldsymbol{\theta}(\tau)\| \leq \sqrt{(t-1)\sum_{\tau=0}^{t-2}\|\boldsymbol{\theta}(\tau+1)-\boldsymbol{\theta}(\tau)\|^2}$$

$$\leq \sqrt{t\cdot O(\eta)} \leq \sqrt{c\frac{R^2}{\eta}\cdot O(\eta)}.$$

Choosing $c$ sufficiently small, we can ensure $\|\boldsymbol{\theta}(t-1)-\boldsymbol{\theta}(0)\| \leq \frac{R}{2}$.

Now that we have proved $\|\boldsymbol{\theta}(t-1) - \boldsymbol{\theta}(0)\| \le \frac{R}{2}$, to prove $\|\boldsymbol{\theta}(t) - \boldsymbol{\theta}(0)\| \le R$ it suffices to bound the one-step deviation $\|\boldsymbol{\theta}(t) - \boldsymbol{\theta}(t-1)\|$ by $\frac{R}{2}$. Using the exact same method in (14), we have

$$\|\boldsymbol{\theta}(t) - \boldsymbol{\theta}(t-1)\| \le \frac{\eta}{n}\sqrt{n\|\boldsymbol{K}\| + O(n\epsilon)} \lesssim \eta\sqrt{\|\boldsymbol{K}\|/n} = \sqrt{\eta\|\boldsymbol{K}\|/n}\sqrt{\eta} \le \sqrt{c}R,$$

where we have used $\eta \le \frac{n}{\|\boldsymbol{K}\|}$ and $\eta \le \eta t \le cR^2$. Choosing $c$ sufficiently small, we can ensure $\|\boldsymbol{\theta}(t) - \boldsymbol{\theta}(t-1)\| \le \frac{R}{2}$. Therefore we conclude that $\|\boldsymbol{\theta}(t) - \boldsymbol{\theta}(0)\| \le R$.

**Step 2: proving $\|\boldsymbol{u}(t) - \boldsymbol{u}^{\mathrm{lin}}(t)\| \lesssim \frac{\eta t \epsilon}{\sqrt{n}}$.** Same as (13) we have

$$\boldsymbol{u}(t) - \boldsymbol{y} = \left(\boldsymbol{I} - \frac{\eta}{n}\boldsymbol{K}\right)(\boldsymbol{u}(t-1) - \boldsymbol{y}) - \frac{\eta}{n}\boldsymbol{E}(t-1)(\boldsymbol{u}(t-1) - \boldsymbol{y}),$$

where $\boldsymbol{E}(t-1) = \boldsymbol{J}(\boldsymbol{\theta}(t-1), \boldsymbol{\theta}(t))\boldsymbol{J}(\boldsymbol{\theta}(t-1))^\top - \boldsymbol{K}$. Since $\|\boldsymbol{\theta}(t-1) - \boldsymbol{\theta}(0)\| \le R$ and $\|\boldsymbol{\theta}(t) - \boldsymbol{\theta}(0)\| \le R$, we know from Assumption C.1 that $\|\boldsymbol{E}(t-1)\| \le \epsilon$. Moreover, from Claim C.1 we know

$$\boldsymbol{u}^{\mathrm{lin}}(t) - \boldsymbol{y} = \left(\boldsymbol{I} - \frac{\eta}{n}\boldsymbol{K}\right)(\boldsymbol{u}^{\mathrm{lin}}(t-1) - \boldsymbol{y}).$$

It follows that

$$\boldsymbol{u}(t) - \boldsymbol{u}^{\mathrm{lin}}(t) = \left(\boldsymbol{I} - \frac{\eta}{n}\boldsymbol{K}\right)(\boldsymbol{u}(t-1) - \boldsymbol{u}^{\mathrm{lin}}(t-1)) - \frac{\eta}{n}\boldsymbol{E}(t-1)(\boldsymbol{u}(t-1) - \boldsymbol{y}),$$

which implies

$$\begin{aligned}
\|\boldsymbol{u}(t) - \boldsymbol{u}^{\mathrm{lin}}(t)\| &\le \left\|\left(\boldsymbol{I} - \frac{\eta}{n}\boldsymbol{K}\right)(\boldsymbol{u}(t-1) - \boldsymbol{u}^{\mathrm{lin}}(t-1))\right\| + \left\|\frac{\eta}{n}\boldsymbol{E}(t-1)(\boldsymbol{u}(t-1) - \boldsymbol{y})\right\| \\
&\le \|\boldsymbol{u}(t-1) - \boldsymbol{u}^{\mathrm{lin}}(t-1)\| + O\left(\frac{\eta}{n}\epsilon\sqrt{n}\right) \\
&= \|\boldsymbol{u}(t-1) - \boldsymbol{u}^{\mathrm{lin}}(t-1)\| + O\left(\frac{\eta\epsilon}{\sqrt{n}}\right).
\end{aligned}$$

Therefore from $\|\boldsymbol{u}(t-1) - \boldsymbol{u}^{\mathrm{lin}}(t-1)\| \lesssim \frac{\eta(t-1)\epsilon}{\sqrt{n}}$ we know $\|\boldsymbol{u}(t) - \boldsymbol{u}^{\mathrm{lin}}(t)\| \lesssim \frac{\eta t \epsilon}{\sqrt{n}}$, completing the proof.

Finally, we prove the last statement in the theorem, i.e., $\|\boldsymbol{\omega}(t) - \boldsymbol{\omega}(0)\| \le R$. In fact we have already proved this – notice that we have proved $\|\boldsymbol{\theta}(t) - \boldsymbol{\theta}(0)\| \le R$ and that a special instance of this problem is when $\boldsymbol{\theta}(t) = \boldsymbol{\omega}(t)$, i.e., the two dynamics are the same. Applying our result on that problem instance, we obtain $\|\boldsymbol{\omega}(t) - \boldsymbol{\omega}(0)\| \le R$. □

# D Omitted Details in Section 3

In Appendix D.1, we present the formal theoretical guarantee (Theorem D.1) for the case of training both layers.

In Appendix D.2, we calculate the formulae of various Jacobians and NTKs that will be used in the analysis.

In Appendix D.3, we prove Theorem 3.2 (training the first layer).

In Appendix D.4, we prove Corollary 3.3 (training the first layer with well-conditioned data).

In Appendix D.5, we prove Theorem 3.5 (training the second layer).

In Appendix D.6, we prove Theorem D.1 (training both layers).

In Appendix D.7, we prove Claim 3.1 (data concentration properties).

## D.1 Guarantee for Training Both Layers

Now we state our guarantee for the case of training both layers, continuing from Section 3.3. Recall that the neural network weights $(\boldsymbol{W}(t), \boldsymbol{v}(t))$ are updated according to GD (4) with learning rate $\eta_1 = \eta_2 = \eta$. The linear model $f^{\mathrm{lin}}(\boldsymbol{x}; \delta)$ in (11) is also trained with GD:

$$\boldsymbol{\delta}(0) = \boldsymbol{0}_{d+2}, \quad \boldsymbol{\delta}(t+1) = \boldsymbol{\delta}(t) - \eta\nabla_{\boldsymbol{\delta}}\frac{1}{2n}\sum_{i=1}^{n}(f^{\mathrm{lin}}(\boldsymbol{x}_i; \boldsymbol{\delta}(t)) - y_i)^2.$$

We let $f_t$ and $f_t^{\text{lin}}$ be the neural network and the linear model at iteration $t$, i.e., $f_t(\boldsymbol{x}) := f(\boldsymbol{x}; \boldsymbol{W}(t), \boldsymbol{v}(t))$ and $f_t^{\text{lin}}(\boldsymbol{x}) := f^{\text{lin}}(\boldsymbol{x}; \boldsymbol{\delta}(t))$.

**Theorem D.1** (main theorem for training both layers). *Let $\alpha \in (0, \frac{1}{4})$ be a fixed constant. Suppose*
$n \gtrsim d^{1+\alpha}$ *and* $m \gtrsim d^{2+\alpha}$. *Suppose* $\begin{cases} \eta \ll d/\log n, & \text{if } \mathbb{E}[\phi(g)] = 0 \\ \eta \ll 1, & \text{otherwise} \end{cases}$ *. Then there exists a universal*
*constant $c > 0$ such that with high probability, for all $0 \le t \le T = c \cdot \frac{d\log d}{\eta}$ simultaneously, we have*

$$\frac{1}{n}\sum_{i=1}^{n} \left(f_t(\boldsymbol{x}_i) - f_t^{\text{lin}}(\boldsymbol{x}_i)\right)^2 \lesssim d^{-\Omega(\alpha)}, \quad \mathbb{E}_{\boldsymbol{x}\sim\mathcal{D}}\left[\min\{(f_t(\boldsymbol{x}) - f_t^{\text{lin}}(\boldsymbol{x}))^2, 1\}\right] \lesssim d^{-\Omega(\alpha)} + \sqrt{\frac{\log T}{n}}.$$

We remark that if the data distribution is well-conditioned, we can also have a guarantee similar to Corollary 3.3.

## D.2 Formulae of Jacobians and NTKs

We first calculate the Jacobian of the network outputs at the training data $\boldsymbol{X}$ with respect to the weights in the network. The Jacobian for the first layer is:

$$\boldsymbol{J}_1(\boldsymbol{W}, \boldsymbol{v}) := [\boldsymbol{J}_1(\boldsymbol{w}_1, v_1), \boldsymbol{J}_1(\boldsymbol{w}_2, v_2), \dots, \boldsymbol{J}_1(\boldsymbol{w}_m, v_m)] \in \mathbb{R}^{n\times md}, \tag{15}$$

where

$$\boldsymbol{J}_1(\boldsymbol{w}_r, v_r) := \frac{1}{\sqrt{md}}v_r\text{diag}\left(\phi'(\boldsymbol{X}\boldsymbol{w}_r/\sqrt{d})\right)\boldsymbol{X} \in \mathbb{R}^{n\times d}, \qquad r \in [m].$$

The Jacobian for the second layer is:

$$\boldsymbol{J}_2(\boldsymbol{W}) := \frac{1}{\sqrt{m}}\phi(\boldsymbol{X}\boldsymbol{W}^\top/\sqrt{d}) \in \mathbb{R}^{n\times m}. \tag{16}$$

Here we omit $\boldsymbol{v}$ in the notation since it does not affect the Jacobian. The Jacobian for both layers is simply $\boldsymbol{J}(\boldsymbol{W}, \boldsymbol{v}) := [\boldsymbol{J}_1(\boldsymbol{W}, \boldsymbol{v}), \boldsymbol{J}_2(\boldsymbol{W})] \in \mathbb{R}^{n\times(md+m)}$.

After calculating the Jacobians, we can calculate the NTK matrices for the first layer, the second layer, and both layers as follows:

$$\boldsymbol{\Theta}_1(\boldsymbol{W}, \boldsymbol{v}) := \boldsymbol{J}_1(\boldsymbol{W}, \boldsymbol{v})\boldsymbol{J}_1(\boldsymbol{W}, \boldsymbol{v})^\top = \frac{1}{m}\sum_{r=1}^{m}v_r^2\left(\phi'(\boldsymbol{X}\boldsymbol{w}_r/\sqrt{d})\phi'(\boldsymbol{X}\boldsymbol{w}_r/\sqrt{d})^\top\right)\odot\frac{\boldsymbol{X}\boldsymbol{X}^\top}{d},$$

$$\boldsymbol{\Theta}_2(\boldsymbol{W}) := \boldsymbol{J}_2(\boldsymbol{W})\boldsymbol{J}_2(\boldsymbol{W})^\top = \frac{1}{m}\phi(\boldsymbol{X}\boldsymbol{W}^\top/\sqrt{d})\phi(\boldsymbol{X}\boldsymbol{W}^\top/\sqrt{d})^\top,$$

$$\boldsymbol{\Theta}(\boldsymbol{W}, \boldsymbol{v}) := \boldsymbol{J}(\boldsymbol{W}, \boldsymbol{v})\boldsymbol{J}(\boldsymbol{W}, \boldsymbol{v})^\top = \boldsymbol{\Theta}_1(\boldsymbol{W}, \boldsymbol{v}) + \boldsymbol{\Theta}_2(\boldsymbol{W}).$$
$$\tag{17}$$

We also denote the expected NTK matrices at random initialization as:

$$\boldsymbol{\Theta}_1^* := \mathbb{E}_{\boldsymbol{w}\sim\mathcal{N}(\boldsymbol{0}, \boldsymbol{I}), v\sim\text{Unif}\{\pm 1\}}\left[v^2\left(\phi'(\boldsymbol{X}\boldsymbol{w}/\sqrt{d})\phi'(\boldsymbol{X}\boldsymbol{w}/\sqrt{d})^\top\right)\right]\odot\frac{\boldsymbol{X}\boldsymbol{X}^\top}{d}$$

$$= \mathbb{E}_{\boldsymbol{w}\sim\mathcal{N}(\boldsymbol{0}, \boldsymbol{I})}\left[\left(\phi'(\boldsymbol{X}\boldsymbol{w}/\sqrt{d})\phi'(\boldsymbol{X}\boldsymbol{w}/\sqrt{d})^\top\right)\right]\odot\frac{\boldsymbol{X}\boldsymbol{X}^\top}{d},$$

$$\boldsymbol{\Theta}_2^* := \mathbb{E}_{\boldsymbol{w}\sim\mathcal{N}(\boldsymbol{0}, \boldsymbol{I})}\left[\phi(\boldsymbol{X}\boldsymbol{w}/\sqrt{d})\phi(\boldsymbol{X}\boldsymbol{w}/\sqrt{d})^\top\right],$$

$$\boldsymbol{\Theta}^* := \boldsymbol{\Theta}_1^* + \boldsymbol{\Theta}_2^*.$$
$$\tag{18}$$

These are also the NTK matrices at infinite width ($m \to \infty$).

Next, for the three linear models (5), (9) and (11) defined in Section 3, denote their feature/Jacobian matrices by:

$$\boldsymbol{\Psi}_1 := [\boldsymbol{\psi}_1(\boldsymbol{x}_1), \dots, \boldsymbol{\psi}_1(\boldsymbol{x}_n)]^\top,$$
$$\boldsymbol{\Psi}_2 := [\boldsymbol{\psi}_2(\boldsymbol{x}_1), \dots, \boldsymbol{\psi}_2(\boldsymbol{x}_n)]^\top,$$
$$\boldsymbol{\Psi} := [\boldsymbol{\psi}(\boldsymbol{x}_1), \dots, \boldsymbol{\psi}(\boldsymbol{x}_n)]^\top.$$
$$\tag{19}$$

Figure 5: **Verification of Proposition 3.4/D.2.** We simulate the dependence of the spectral and Frobenius norms of $\boldsymbol{\Theta}_1(\boldsymbol{W}(0)){-}\boldsymbol{\Theta}^{\mathrm{lin1}}$ on $d$. We set $\phi = \mathrm{erf}$, $n = 10^4$ and $m = 2{\times}10^4$, and generate data from $\mathcal{N}(\boldsymbol{0}, \boldsymbol{I})$ for various $d$. We perform a linear least-squares fit on the log mean norms against $\log(d)$. Numerically we find $\left\| \boldsymbol{\Theta}_1(\boldsymbol{W}(0)) - \boldsymbol{\Theta}^{\mathrm{lin1}} \right\| \propto d^{-1.263}$ and $\left\| \boldsymbol{\Theta}_1(\boldsymbol{W}(0)) - \boldsymbol{\Theta}^{\mathrm{lin1}} \right\|_F \propto d^{-0.718}$.

Consequently, their corresponding kernel matrices are:

$$\boldsymbol{\Theta}^{\mathrm{lin1}} := \boldsymbol{\Psi}_1 \boldsymbol{\Psi}_1^\top = \frac{1}{d}(\zeta^2 \boldsymbol{X}\boldsymbol{X}^\top + \nu^2 \boldsymbol{1}\boldsymbol{1}^\top),$$

$$\boldsymbol{\Theta}^{\mathrm{lin2}} := \boldsymbol{\Psi}_2 \boldsymbol{\Psi}_2^\top = \frac{1}{d}\left( \zeta^2 \boldsymbol{X}\boldsymbol{X}^\top + \frac{1}{2}\nu^2 \boldsymbol{1}\boldsymbol{1}^\top \right) + \boldsymbol{q}\boldsymbol{q}^\top, \qquad (20)$$

$$\boldsymbol{\Theta}^{\mathrm{lin}} := \boldsymbol{\Psi}\boldsymbol{\Psi}^\top = \frac{1}{d}\left( 2\zeta^2 \boldsymbol{X}\boldsymbol{X}^\top + \frac{3}{2}\nu^2 \boldsymbol{1}\boldsymbol{1}^\top \right) + \boldsymbol{q}\boldsymbol{q}^\top.$$

Here the constants are defined in (9), and $\boldsymbol{q} \in \mathbb{R}^n$ is defined as $[\boldsymbol{q}]_i := \vartheta_0 + \vartheta_1\left(\frac{\|\boldsymbol{x}_i\|}{\sqrt{d}} - 1\right) + \vartheta_2\left(\frac{\|\boldsymbol{x}_i\|}{\sqrt{d}} - 1\right)^2$ for each $i \in [n]$.

### D.3   Proof of Theorem 3.2 (Training the First Layer)

For convenience we let $\boldsymbol{v} = \boldsymbol{v}(0)$ which is the fixed second layer. Since we have $v_r \in \{\pm 1\}$ ($\forall r \in [m]$), we can write the first-layer NTK matrix as

$$\boldsymbol{\Theta}_1(\boldsymbol{W}, \boldsymbol{v}) = \frac{1}{m}\sum_{r=1}^m \left( \phi'(\boldsymbol{X}\boldsymbol{w}_r/\sqrt{d})\phi'(\boldsymbol{X}\boldsymbol{w}_r/\sqrt{d})^\top \right) \odot \frac{\boldsymbol{X}\boldsymbol{X}^\top}{d}.$$

Because it does not depend on $\boldsymbol{v}$, we denote $\boldsymbol{\Theta}_1(\boldsymbol{W}) := \boldsymbol{\Theta}_1(\boldsymbol{W}, \boldsymbol{v})$ for convenience.

#### D.3.1   The NTK at Initialization

Now we prove Proposition 3.4, restated below:

**Proposition D.2** (restatement of Proposition 3.4). *With high probability over the random initialization $\boldsymbol{W}(0)$ and the training data $\boldsymbol{X}$, we have*

$$\left\| \boldsymbol{\Theta}_1(\boldsymbol{W}(0)) - \boldsymbol{\Theta}^{\mathrm{lin1}} \right\| \lesssim \frac{n}{d^{1+\alpha}}.$$

We perform a simulation to empirically verify Proposition D.2 in Figure 5. Here we fix $n$ and $m$ to be large and look at the dependence of $\left\| \boldsymbol{\Theta}_1(\boldsymbol{W}(0)) - \boldsymbol{\Theta}^{\mathrm{lin1}} \right\|$ on $d$. We find that $\left\| \boldsymbol{\Theta}_1(\boldsymbol{W}(0)) - \boldsymbol{\Theta}^{\mathrm{lin1}} \right\|$ indeed decays faster than $\frac{1}{d}$. In contrast, $\left\| \boldsymbol{\Theta}_1(\boldsymbol{W}(0)) - \boldsymbol{\Theta}^{\mathrm{lin1}} \right\|_F$ decays slower than $\frac{1}{d}$, indicating that bounding the Frobenius norm is insufficient.

To prove Proposition D.2, we will prove $\boldsymbol{\Theta}_1(\boldsymbol{W}(0))$ is close to its expectation $\boldsymbol{\Theta}_1^*$ (defined in (18)), and then prove $\boldsymbol{\Theta}_1^*$ is close to $\boldsymbol{\Theta}^{\mathrm{lin1}}$. We do these steps in the next two propositions.

**Proposition D.3.** *With high probability over the random initialization $\boldsymbol{W}(0)$ and the training data $\boldsymbol{X}$, we have*

$$\|\boldsymbol{\Theta}_1(\boldsymbol{W}(0)) - \boldsymbol{\Theta}_1^*\| \le \frac{n}{d^{1+\alpha}}.$$

*Proof.* For convenience we denote $\boldsymbol{W} = \boldsymbol{W}(0)$ and $\boldsymbol{\Theta}_1 = \boldsymbol{\Theta}_1(\boldsymbol{W}) = \boldsymbol{\Theta}_1(\boldsymbol{W}(0))$ in this proof.

From Claim 3.1 we know $\|\boldsymbol{X}\boldsymbol{X}^\top\| = O(n)$ with high probability. For the rest of the proof we will be conditioned on $\boldsymbol{X}$ and on Claim 3.1, and only consider the randomness in $\boldsymbol{W}$.

We define $\boldsymbol{\Theta}_1^{(r)} := \left(\phi'(\boldsymbol{X}\boldsymbol{w}_r/\sqrt{d})\phi'(\boldsymbol{X}\boldsymbol{w}_r/\sqrt{d})^\top\right) \odot \frac{\boldsymbol{X}\boldsymbol{X}^\top}{d}$ for each $r \in [m]$. Then we have $\boldsymbol{\Theta}_1 = \frac{1}{m}\sum_{r=1}^m \boldsymbol{\Theta}_1^{(r)}$. According to the initialization scheme (3), we know that $\boldsymbol{\Theta}_1^{(1)}, \boldsymbol{\Theta}_1^{(2)}, \ldots, \boldsymbol{\Theta}_1^{(m/2)}$ are independent, $\boldsymbol{\Theta}_1^{(m/2+1)}, \boldsymbol{\Theta}_1^{(m/2+2)}, \ldots, \boldsymbol{\Theta}_1^{(m)}$ are independent, and $\mathbb{E}[\boldsymbol{\Theta}_1^{(r)}] = \boldsymbol{\Theta}_1^*$ for all $r \in [m]$.

Next we will apply the matrix Bernstein inequality (Theorem 1.6.2 in Tropp [2015]) to bound $\|\boldsymbol{\Theta}_1 - \boldsymbol{\Theta}_1^*\|$. We will first consider the first half of independent neurons, i.e. $r \in [m/2]$. For each $r$ we have

$$
\begin{aligned}
\left\|\boldsymbol{\Theta}_1^{(r)}\right\| &= \left\|\text{diag}\left(\phi'(\boldsymbol{X}\boldsymbol{w}_r/\sqrt{d})\right) \cdot \frac{\boldsymbol{X}\boldsymbol{X}^\top}{d} \cdot \text{diag}\left(\phi'(\boldsymbol{X}\boldsymbol{w}_r/\sqrt{d})\right)\right\| \\
&\le \left\|\text{diag}\left(\phi'(\boldsymbol{X}\boldsymbol{w}_r/\sqrt{d})\right)\right\| \cdot \left\|\frac{\boldsymbol{X}\boldsymbol{X}^\top}{d}\right\| \cdot \left\|\text{diag}\left(\phi'(\boldsymbol{X}\boldsymbol{w}_r/\sqrt{d})\right)\right\| \\
&\le O(1) \cdot O(n/d) \cdot O(1) \\
&= O(n/d).
\end{aligned}
$$

Here we have used the boundedness of $\phi'(\cdot)$ (Assumption 3.2). Since $\boldsymbol{\Theta}_1^* = \mathbb{E}[\boldsymbol{\Theta}_1^{(r)}]$, it follows that

$$\|\boldsymbol{\Theta}_1^*\| \le O(n/d),$$

$$\left\|\boldsymbol{\Theta}_1^{(r)} - \boldsymbol{\Theta}_1^*\right\| \le O(n/d), \qquad \forall r \in [m/2]$$

$$\left\|\sum_{r=1}^{m/2}\mathbb{E}[(\boldsymbol{\Theta}_1^{(r)} - \boldsymbol{\Theta}_1^*)^2]\right\| \le \sum_{r=1}^{m/2}\left\|\mathbb{E}[(\boldsymbol{\Theta}_1^{(r)} - \boldsymbol{\Theta}_1^*)^2]\right\| \le O(mn^2/d^2).$$

Therefore, from the the matrix Bernstein inequality, for any $s \ge 0$ we have:

$$\Pr\left[\left\|\sum_{r=1}^{m/2}(\boldsymbol{\Theta}_1^{(r)} - \boldsymbol{\Theta}_1^*)\right\| \ge s\right] \le 2n \cdot \exp\left(\frac{-s^2/2}{O(mn^2/d^2 + sn/d)}\right).$$

Letting $s = \frac{m}{2} \cdot \frac{n}{d^{1+\alpha}}$, we obtain

$$
\begin{aligned}
\Pr\left[\left\|\sum_{r=1}^{m/2}(\boldsymbol{\Theta}_1^{(r)} - \boldsymbol{\Theta}_1^*)\right\| \ge \frac{m}{2} \cdot \frac{n}{d^{1+\alpha}}\right] &\le 2n \cdot \exp\left(-\Omega\left(\frac{m^2n^2/d^{2+2\alpha}}{mn^2/d^2 + mn^2/d^{2+\alpha}}\right)\right) \\
&= 2n \cdot \exp\left(-\Omega\left(\frac{m}{d^{2\alpha}}\right)\right) \\
&= d^{O(1)} \cdot e^{-\Omega(d^{1-\alpha})} \\
&\ll 1,
\end{aligned}
$$

where we have used $m = \Omega(d^{1+\alpha})$ and $n = d^{O(1)}$. Therefore with high probability we have

$$\left\|\sum_{r=1}^{m/2}(\boldsymbol{\Theta}_1^{(r)} - \boldsymbol{\Theta}_1^*)\right\| \le \frac{m}{2} \cdot \frac{n}{d^{1+\alpha}}.$$

Similarly, for the second half of the neurons we also have with high probability

$$\left\|\sum_{r=m/2+1}^{m}(\boldsymbol{\Theta}_1^{(r)} - \boldsymbol{\Theta}_1^*)\right\| \le \frac{m}{2} \cdot \frac{n}{d^{1+\alpha}}.$$

Finally, by the triangle inequality we have

$$\|\boldsymbol{\Theta}_1 - \boldsymbol{\Theta}_1^*\| = \frac{1}{m}\left\|\sum_{r=1}^{m}(\boldsymbol{\Theta}_1^{(r)} - \boldsymbol{\Theta}_1^*)\right\| \leq \frac{1}{m}\left(\frac{m}{2}\cdot\frac{n}{d^{1+\alpha}} + \frac{m}{2}\cdot\frac{n}{d^{1+\alpha}}\right) = \frac{n}{d^{1+\alpha}}$$

with high probability, completing the proof. $\qquad\square$

**Proposition D.4.** *With high probability over the training data $\boldsymbol{X}$, we have*

$$\left\|\boldsymbol{\Theta}_1^* - \boldsymbol{\Theta}^{\mathrm{lin1}}\right\| \lesssim \frac{n}{d^{1+\alpha}}.$$

*Proof.* We will be conditioned on the high probability events stated in Claim 3.1.

By the definition of $\boldsymbol{\Theta}_1^*$, we know

$$[\boldsymbol{\Theta}_1^*]_{i,j} = \frac{1}{d}\boldsymbol{x}_i^\top\boldsymbol{x}_j\cdot\mathbb{E}_{\boldsymbol{w}\sim\mathcal{N}(\boldsymbol{0},\boldsymbol{I})}\left[\phi'(\boldsymbol{w}^\top\boldsymbol{x}_i/\sqrt{d})\phi'(\boldsymbol{w}^\top\boldsymbol{x}_j/\sqrt{d})^\top\right], \qquad i,j\in[n].$$

We define

$$\Phi(a,b,c) := \mathbb{E}_{(z_1,z_2)\sim\mathcal{N}(\boldsymbol{0},\boldsymbol{\Lambda})}[\phi'(z_1)\phi'(z_2)], \text{ where } \boldsymbol{\Lambda} = \begin{pmatrix} a & c \\ c & b \end{pmatrix}, \quad a\geq 0, b\geq 0, |c|\leq\sqrt{ab}.$$

Then we can write

$$[\boldsymbol{\Theta}_1^*]_{i,j} = \frac{1}{d}\boldsymbol{x}_i^\top\boldsymbol{x}_j\cdot\Phi\left(\frac{\|\boldsymbol{x}_i\|^2}{d}, \frac{\|\boldsymbol{x}_j\|^2}{d}, \frac{\boldsymbol{x}_i^\top\boldsymbol{x}_j}{d}\right).$$

We consider the diagonal and off-diagonal entries of $\boldsymbol{\Theta}_1^*$ separately.

For $i\neq j$, from Claim 3.1 we know $\frac{\|\boldsymbol{x}_i\|^2}{d} = 1\pm\tilde{O}(\frac{1}{\sqrt{d}})$, $\frac{\|\boldsymbol{x}_j\|^2}{d} = 1\pm\tilde{O}(\frac{1}{\sqrt{d}})$ and $\frac{\boldsymbol{x}_i^\top\boldsymbol{x}_j}{d} = \pm\tilde{O}(\frac{1}{\sqrt{d}})$. Hence we apply Taylor expansion of $\Phi$ around $(1,1,0)$:

$$\Phi\left(\frac{\|\boldsymbol{x}_i\|^2}{d}, \frac{\|\boldsymbol{x}_j\|^2}{d}, \frac{\boldsymbol{x}_i^\top\boldsymbol{x}_j}{d}\right)$$

$$= \Phi(1,1,0) + c_1\left(\frac{\|\boldsymbol{x}_i\|^2}{d} - 1\right) + c_2\left(\frac{\|\boldsymbol{x}_j\|^2}{d} - 1\right) + c_3\frac{(\boldsymbol{x}_i^\top\boldsymbol{x}_j)^2}{d}$$

$$\pm O\left(\left(\frac{\|\boldsymbol{x}_i\|^2}{d} - 1\right)^2 + \left(\frac{\|\boldsymbol{x}_j\|^2}{d} - 1\right)^2 + \left(\frac{(\boldsymbol{x}_i^\top\boldsymbol{x}_j)^2}{d}\right)^2\right)$$

$$= \Phi(1,1,0) + c_1\left(\frac{\|\boldsymbol{x}_i\|^2}{d} - 1\right) + c_2\left(\frac{\|\boldsymbol{x}_j\|^2}{d} - 1\right) + c_3\frac{(\boldsymbol{x}_i^\top\boldsymbol{x}_j)^2}{d} \pm \tilde{O}\left(\frac{1}{d}\right).$$

Here $(c_1, c_2, c_3) := \nabla\Phi(1,1,0)$. Note that $\Phi(1,1,0)$ and all first and second order derivatives of $\Phi$ at $(1,1,0)$ exist and are bounded for activation $\phi$ that satisfies Assumption 3.2. In particular, we have $\Phi(1,1,0) = (\mathbb{E}[\phi'(g)])^2 = \zeta^2$ and $c_3 = (\mathbb{E}[g\phi'(g)])^2$. Using the above expansion, we can write

$$(\boldsymbol{\Theta}_1^*)_{\mathrm{off}} = \zeta^2\left(\frac{\boldsymbol{X}\boldsymbol{X}^\top}{d}\right)_{\mathrm{off}} + c_1\left(\mathrm{diag}(\boldsymbol{\epsilon})\cdot\frac{\boldsymbol{X}\boldsymbol{X}^\top}{d}\right)_{\mathrm{off}} + c_2\left(\frac{\boldsymbol{X}\boldsymbol{X}^\top}{d}\cdot\mathrm{diag}(\boldsymbol{\epsilon})\right)_{\mathrm{off}}$$
$$+ c_3\left(\frac{\boldsymbol{X}\boldsymbol{X}^\top}{d}\odot\frac{\boldsymbol{X}\boldsymbol{X}^\top}{d}\right)_{\mathrm{off}} + \boldsymbol{E}, \tag{21}$$

where $\boldsymbol{\epsilon}\in\mathbb{R}^n$ is defined as $[\boldsymbol{\epsilon}]_i = \frac{\|\boldsymbol{x}_i\|^2}{d} - 1$, and $[\boldsymbol{E}]_{i,j} = \pm\tilde{O}(\frac{1}{d})\cdot\frac{\boldsymbol{x}_i^\top\boldsymbol{x}_j}{d}\mathbb{1}_{\{i\neq j\}} = \pm\tilde{O}(\frac{1}{d^{1.5}})$.

Now we treat the terms in (21) separately. First, we have

$$\left\|\mathrm{diag}(\boldsymbol{\epsilon})\cdot\frac{\boldsymbol{X}\boldsymbol{X}^\top}{d}\right\| \leq \|\mathrm{diag}(\boldsymbol{\epsilon})\|\cdot\left\|\frac{\boldsymbol{X}\boldsymbol{X}^\top}{d}\right\| = \max_{i\in[n]}\left|[\boldsymbol{\epsilon}]_i\right|\cdot\left\|\frac{\boldsymbol{X}\boldsymbol{X}^\top}{d}\right\| \leq \tilde{O}\left(\frac{1}{\sqrt{d}}\right)\cdot O\left(\frac{n}{d}\right)$$

$$= \tilde{O}\left(\frac{n}{d^{1.5}}\right).$$

Similarly, we have $\left\| \frac{\boldsymbol{X}\boldsymbol{X}^\top}{d} \cdot \mathrm{diag}(\boldsymbol{\epsilon}) \right\| \leq \tilde{O}\left(\frac{n}{d^{1.5}}\right)$.

Next, for $\left(\frac{\boldsymbol{X}\boldsymbol{X}^\top}{d} \odot \frac{\boldsymbol{X}\boldsymbol{X}^\top}{d}\right)_{\mathrm{off}}$, we can use the 4th moment method in El Karoui [2010] to show that it is close to its mean. Specifically, the mean at each entry is $\mathbb{E}\left[\left(\frac{\boldsymbol{x}_i^\top \boldsymbol{x}_j}{d}\right)^2\right] = \frac{\mathrm{Tr}[\boldsymbol{\Sigma}^2]}{d^2}$ $(i \neq j)$, and the moment calculation in El Karoui [2010] shows the following bound on the error matrix $\boldsymbol{F} = \left(\frac{\boldsymbol{X}\boldsymbol{X}^\top}{d} \odot \frac{\boldsymbol{X}\boldsymbol{X}^\top}{d} - \frac{\mathrm{Tr}[\boldsymbol{\Sigma}^2]}{d^2}\mathbf{1}\mathbf{1}^\top\right)_{\mathrm{off}}$:

$$\mathbb{E}\left[\|\boldsymbol{F}\|^4\right] \leq \mathbb{E}\left[\mathrm{Tr}[\boldsymbol{F}^4]\right] \leq \tilde{O}\left(\frac{n^4}{d^6} + \frac{n^3}{d^4}\right) \leq \tilde{O}\left(\frac{n^4}{d^5}\right),$$

where we have used $n \gtrsim d$. Therefore by Markov inequality we know that with high probability, $\|\boldsymbol{F}\| \leq \tilde{O}\left(\frac{n}{d^{1.25}}\right)$.

For the final term $\boldsymbol{E}$ in (21), we have

$$\|\boldsymbol{E}\| \leq \|\boldsymbol{E}\|_F \leq \sqrt{n^2 \cdot \tilde{O}\left(\frac{1}{d^3}\right)} = \tilde{O}\left(\frac{n}{d^{1.5}}\right).$$

Put together, we can obtain the following bound regarding $(\boldsymbol{\Theta}_1^*)_{\mathrm{off}}$:

$$\begin{aligned}
&\left\|\left(\boldsymbol{\Theta}_1^* - \zeta^2 \frac{\boldsymbol{X}\boldsymbol{X}^\top}{d} - c_3 \frac{\mathrm{Tr}[\boldsymbol{\Sigma}^2]}{d^2}\mathbf{1}\mathbf{1}^\top\right)_{\mathrm{off}}\right\| \\
&\leq c_1 \cdot \tilde{O}\left(\frac{n}{d^{1.5}}\right) + c_2 \cdot \tilde{O}\left(\frac{n}{d^{1.5}}\right) + c_3 \cdot \tilde{O}\left(\frac{n}{d^{1.25}}\right) + \tilde{O}\left(\frac{n}{d^{1.5}}\right) \\
&= \tilde{O}\left(\frac{n}{d^{1.25}}\right).
\end{aligned} \tag{22}$$

Here we have used Lemma B.2 to bound the spectral norm of the off-diagonal part of a matrix by the spectral norm of the matrix itself. Notice $c_3 \frac{\mathrm{Tr}[\boldsymbol{\Sigma}^2]}{d} = (\mathbb{E}[g\phi'(g)])^2 \cdot \frac{\mathrm{Tr}[\boldsymbol{\Sigma}^2]}{d} = \nu^2$ (c.f. (5)). Hence (22) becomes

$$\left\|\left(\boldsymbol{\Theta}_1^* - \boldsymbol{\Theta}^{\mathrm{lin}1}\right)_{\mathrm{off}}\right\| = \tilde{O}\left(\frac{n}{d^{1.25}}\right). \tag{23}$$

For the diagonal entries of $\boldsymbol{\Theta}_1^*$, we have $[\boldsymbol{\Theta}_1^*]_{i,i} = \frac{\|\boldsymbol{x}_i\|^2}{d} \cdot \Phi\left(\frac{\|\boldsymbol{x}_i\|^2}{d}, \frac{\|\boldsymbol{x}_i\|^2}{d}, \frac{\|\boldsymbol{x}_i\|^2}{d}\right)$. We denote $\bar{\Phi}(a) := \Phi(a, a, a)$ $(a \geq 0)$. When $\phi$ is a smooth activation as in Assumption 3.2, we know that $\bar{\Phi}$ has bounded derivative, and thus we get

$$[\boldsymbol{\Theta}_1^*]_{i,i} = \frac{\|\boldsymbol{x}_i\|^2}{d} \cdot \bar{\Phi}\left(\frac{\|\boldsymbol{x}_i\|^2}{d}\right) = \left(1 \pm \tilde{O}\left(\frac{1}{\sqrt{d}}\right)\right) \cdot \left(\bar{\Phi}(1) \pm \tilde{O}\left(\frac{1}{\sqrt{d}}\right)\right) = \bar{\Phi}(1) \pm \tilde{O}\left(\frac{1}{\sqrt{d}}\right). \tag{24}$$

When $\phi$ is a piece-wise linear activation as in Assumption 3.2, $\bar{\Phi}(a)$ is a constant, so we have $\bar{\Phi}\left(\frac{\|\boldsymbol{x}_i\|^2}{d}\right) = \bar{\Phi}(1)$. Therefore (24) also holds. Notice that $\bar{\Phi}(1) = \mathbb{E}[(\phi'(g))^2] =: \gamma$. It follows from (24) that

$$\|(\boldsymbol{\Theta}_1^*)_{\mathrm{diag}} - \gamma \boldsymbol{I}\| = \tilde{O}\left(\frac{1}{\sqrt{d}}\right).$$

Also note that

$$\left\|\boldsymbol{\Theta}_{\mathrm{diag}}^{\mathrm{lin}1} - \zeta^2 \boldsymbol{I}\right\| = \tilde{O}\left(\frac{1}{\sqrt{d}}\right).$$

Therefore we obtain

$$\left\|\left(\boldsymbol{\Theta}_1^* - \boldsymbol{\Theta}^{\mathrm{lin}1}\right)_{\mathrm{diag}} - (\gamma - \zeta^2)\boldsymbol{I}\right\| = \tilde{O}\left(\frac{1}{\sqrt{d}}\right). \tag{25}$$

Combining the off-diagonal and diagonal approximations (23) and (25), we obtain

$$\left\| \mathbf{\Theta}_1^* - \mathbf{\Theta}^{\mathrm{lin}1} - (\gamma - \zeta^2)\mathbf{I} \right\| = \tilde{O}\left(\frac{n}{d^{1.25}}\right).$$

Finally, when $n \gtrsim d^{1+\alpha}$ $(0 < \alpha < \frac{1}{4})$, we have $\|\mathbf{I}\| = 1 \lesssim \frac{n}{d^{1+\alpha}}$. Hence we can discard the identity component above and get

$$\left\| \mathbf{\Theta}_1^* - \mathbf{\Theta}^{\mathrm{lin}1} \right\| = O\left(\frac{n}{d^{1+\alpha}}\right).$$

This completes the proof. $\qquad\qquad\qquad\qquad\qquad\qquad\qquad\qquad\qquad\qquad\qquad\qquad\square$

Combining Propositions D.3 and D.4 directly gives Proposition D.2.

### D.3.2 Agreement on Training Data

Now we prove the first part of Theorem 3.2, i.e., (7), which says that the neural network $f_t^1$ and the linear model $f_t^{\mathrm{lin}1}$ are close on the training data. We will use Theorem C.2, and the most important step is to verify Assumption C.1. To this end we prove the following Jacobian perturbation lemma.

**Lemma D.5** (Jacobian perturbation for the first layer). *If $\phi$ is a smooth activation as in Assumption 3.2, then with high probability over the training data $\mathbf{X}$, we have*

$$\left\| \mathbf{J}_1(\mathbf{W}, \mathbf{v}) - \mathbf{J}_1(\widetilde{\mathbf{W}}, \mathbf{v}) \right\| \lesssim \sqrt{\frac{n}{md}} \left\| \mathbf{W} - \widetilde{\mathbf{W}} \right\|_F, \qquad \forall \mathbf{W}, \widetilde{\mathbf{W}} \in \mathbb{R}^{m \times d}. \qquad (26)$$

*If $\phi$ is a piece-wise linear activation as in Assumption 3.2, then with high probability over the random initialization $\mathbf{W}(0)$ and the training data $\mathbf{X}$, we have*

$$\left\| \mathbf{J}_1(\mathbf{W}, \mathbf{v}) - \mathbf{J}_1(\mathbf{W}(0), \mathbf{v}) \right\| \lesssim \sqrt{\frac{n}{d}} \left( \frac{\|\mathbf{W} - \mathbf{W}(0)\|^{1/3}}{m^{1/6}} + \left(\frac{\log n}{m}\right)^{1/4} \right), \qquad \forall \mathbf{W} \in \mathbb{R}^{m \times d}. \tag{27}$$

*Proof.* Throughout the proof we will be conditioned on $\mathbf{X}$ and on the high-probability events in Claim 3.1.

By the definition of $\mathbf{J}_1(\mathbf{W}, \mathbf{v})$ in (15), we have

$$(\mathbf{J}_1(\mathbf{W}, \mathbf{v}) - \mathbf{J}_1(\widetilde{\mathbf{W}}, \mathbf{v}))(\mathbf{J}_1(\mathbf{W}, \mathbf{v}) - \mathbf{J}_1(\widetilde{\mathbf{W}}, \mathbf{v}))^\top$$
$$= \frac{1}{md} \left( \phi'\left(\mathbf{X}\mathbf{W}^\top/\sqrt{d}\right) - \phi'\left(\mathbf{X}\widetilde{\mathbf{W}}^\top/\sqrt{d}\right) \right) \left( \phi'\left(\mathbf{X}\mathbf{W}^\top/\sqrt{d}\right) - \phi'\left(\mathbf{X}\widetilde{\mathbf{W}}^\top/\sqrt{d}\right) \right)^\top \odot (\mathbf{X}\mathbf{X}^\top). \tag{28}$$

Then if $\phi$ is a smooth activation, we have with high probability,

$$\left\| \mathbf{J}_1(\mathbf{W}, \mathbf{v}) - \mathbf{J}_1(\widetilde{\mathbf{W}}, \mathbf{v}) \right\|^2$$
$$\leq \frac{1}{md} \left\| \phi'\left(\mathbf{X}\mathbf{W}^\top/\sqrt{d}\right) - \phi'\left(\mathbf{X}\widetilde{\mathbf{W}}^\top/\sqrt{d}\right) \right\|^2 \cdot \max_{i \in [n]} \|\mathbf{x}_i\|^2 \qquad \text{((28) and Lemma B.3)}$$
$$\lesssim \frac{1}{md} \left\| \phi'\left(\mathbf{X}\mathbf{W}^\top/\sqrt{d}\right) - \phi'\left(\mathbf{X}\widetilde{\mathbf{W}}^\top/\sqrt{d}\right) \right\|_F^2 \cdot d \qquad \text{(Claim 3.1)}$$
$$\lesssim \frac{1}{md} \left\| \mathbf{X}\mathbf{W}^\top/\sqrt{d} - \mathbf{X}\widetilde{\mathbf{W}}^\top/\sqrt{d} \right\|_F^2 \cdot d \qquad (\phi'' \text{ is bounded})$$
$$= \frac{1}{md} \left\| \mathbf{X}(\mathbf{W} - \widetilde{\mathbf{W}})^\top \right\|_F^2$$
$$\leq \frac{1}{md} \|\mathbf{X}\|^2 \left\| \mathbf{W} - \widetilde{\mathbf{W}} \right\|_F^2$$
$$\lesssim \frac{n}{md} \left\| \mathbf{W} - \widetilde{\mathbf{W}} \right\|_F^2. \qquad \text{(Claim 3.1)}$$

This proves (26).

Next we consider the case where $\phi$ is a piece-wise linear activation. From (28) and Lemma B.3 we have

$$\|\boldsymbol{J}_1(\boldsymbol{W}, \boldsymbol{v}) - \boldsymbol{J}_1(\boldsymbol{W}(0), \boldsymbol{v})\|^2 \leq \frac{1}{md} \|\boldsymbol{X}\boldsymbol{X}^\top\| \cdot \max_{i \in [n]} \left\|\phi'(\boldsymbol{W}\boldsymbol{x}_i/\sqrt{d}) - \phi'(\boldsymbol{W}(0)\boldsymbol{x}_i/\sqrt{d})\right\|^2$$

$$\lesssim \frac{n}{md} \cdot \max_{i \in [n]} \left\|\phi'(\boldsymbol{W}\boldsymbol{x}_i/\sqrt{d}) - \phi'(\boldsymbol{W}(0)\boldsymbol{x}_i/\sqrt{d})\right\|^2. \tag{29}$$

For each $i \in [n]$, let

$$M_i = \{r \in [m] : \text{sign}(\boldsymbol{w}_r^\top \boldsymbol{x}_i) \neq \text{sign}(\boldsymbol{w}_r(0)^\top \boldsymbol{x}_i)\}$$

Since $\phi'$ is a step function that only depends on the sign of the input, we have

$$\left\|\phi'(\boldsymbol{W}\boldsymbol{x}_i/\sqrt{d}) - \phi'(\boldsymbol{W}(0)\boldsymbol{x}_i/\sqrt{d})\right\|^2 \lesssim |M_i|, \qquad \forall i \in [n]. \tag{30}$$

Therefore we need to bound $|M_i|$, i.e. how many coordinates in $\boldsymbol{W}\boldsymbol{x}_i$ and $\boldsymbol{W}(0)\boldsymbol{x}_i$ differ in sign for each $i \in [n]$.

Let $\lambda > 0$ be a parameter whose value will be determined later. For each $i \in [n]$, define

$$N_i := \{r \in [m] : |\boldsymbol{w}_r(0)^\top \boldsymbol{x}_i| \leq \lambda \|\boldsymbol{x}_i\|\}.$$

We have

$$|N_i| = \sum_{r=1}^m \mathbb{1}_{\{|\boldsymbol{w}_r(0)^\top \boldsymbol{x}_i| \leq \lambda \|\boldsymbol{x}_i\|\}} = 2 \sum_{r=1}^{m/2} \mathbb{1}_{\{|\boldsymbol{w}_r(0)^\top \boldsymbol{x}_i| \leq \lambda \|\boldsymbol{x}_i\|\}},$$

where the second equality is due to the symmetric initialization (3). Since $\frac{\boldsymbol{w}_r(0)^\top \boldsymbol{x}_i}{\|\boldsymbol{x}_i\|} \sim \mathcal{N}(0,1)$, we have $\mathbb{E}\left[\mathbb{1}_{\{|\boldsymbol{w}_r(0)^\top \boldsymbol{x}_i| \leq \lambda \|\boldsymbol{x}_i\|\}}\right] = \Pr[|g| \leq \lambda] \leq \frac{2\lambda}{\sqrt{2\pi}}$. Also note that $\boldsymbol{w}_1(0), \ldots, \boldsymbol{w}_{m/2}(0)$ are independent. Then by Hoeffding's inequality we know that with probability at least $1 - \delta$,

$$|N_i| \leq \sqrt{\frac{2}{\pi}} \lambda m + O\left(\sqrt{m \log \frac{1}{\delta}}\right).$$

Taking a union bound over all $i \in [n]$, we know that with high probability,

$$|N_i| \lesssim \lambda m + \sqrt{m \log n}, \qquad \forall i \in [n]. \tag{31}$$

By definition, if $r \in M_i$ but $r \notin N_i$, we must have $\left|\boldsymbol{w}_r^\top \boldsymbol{x}_i - \boldsymbol{w}_r(0)^\top \boldsymbol{x}_i\right| \geq \left|\boldsymbol{w}_r(0)^\top \boldsymbol{x}_i\right| > \lambda \|\boldsymbol{x}_i\|$. This leads to

$$\|(\boldsymbol{W} - \boldsymbol{W}(0))\boldsymbol{x}_i\|^2 = \sum_{r=1}^m \left|(\boldsymbol{w}_r - \boldsymbol{w}_r(0))^\top \boldsymbol{x}_i\right|^2 \geq \sum_{r \in M_i \setminus N_i} \left|(\boldsymbol{w}_r - \boldsymbol{w}_r(0))^\top \boldsymbol{x}_i\right|^2$$

$$\geq \sum_{r \in M_i \setminus N_i} \lambda^2 \|\boldsymbol{x}_i\|^2 \gtrsim \sum_{r \in M_i \setminus N_i} \lambda^2 d = \lambda^2 d \, |M_i \setminus N_i|$$

Thus we have

$$|M_i \setminus N_i| \lesssim \frac{\|(\boldsymbol{W} - \boldsymbol{W}(0))\boldsymbol{x}_i\|^2}{\lambda^2 d} \leq \frac{\|\boldsymbol{W} - \boldsymbol{W}(0)\|^2 \|\boldsymbol{x}_i\|^2}{\lambda^2 d} \lesssim \frac{\|\boldsymbol{W} - \boldsymbol{W}(0)\|^2}{\lambda^2}, \qquad \forall i \in [n]. \tag{32}$$

Combining (31) and (32) we obtain

$$|M_i| \lesssim \lambda m + \sqrt{m \log n} + \frac{\|\boldsymbol{W} - \boldsymbol{W}(0)\|^2}{\lambda^2}, \qquad \forall i \in [n].$$

Letting $\lambda = \left(\frac{\|\boldsymbol{W} - \boldsymbol{W}(0)\|^2}{m}\right)^{1/3}$, we get

$$|M_i| \lesssim m^{2/3} \|\boldsymbol{W} - \boldsymbol{W}(0)\|^{2/3} + \sqrt{m \log n}, \qquad \forall i \in [n]. \tag{33}$$

Finally, we combine (29), (30) and (33) to obtain

$$\|J_1(W,v) - J_1(W(0),v)\|^2 \lesssim \frac{n}{md}\left(m^{2/3}\|W - W(0)\|^{2/3} + \sqrt{m\log n}\right)$$
$$= \frac{n}{d}\left(\frac{\|W - W(0)\|^{2/3}}{m^{1/3}} + \sqrt{\frac{\log n}{m}}\right).$$

This proves (27). $\qquad\qquad\square$

The next lemma verifies Assumption C.1 for the case of training the first layer.

**Lemma D.6.** *Let $R = \sqrt{d\log d}$. With high probability over the random initialization $W(0)$ and the training data $X$, for all $W, \widetilde{W} \in \mathbb{R}^{m\times d}$ such that $\|W - W(0)\|_F \le R$ and $\left\|\widetilde{W} - W(0)\right\|_F \le R$, we have*

$$\left\|J_1(W,v)J_1(\widetilde{W},v)^\top - \Theta^{\mathrm{lin1}}\right\| \lesssim \frac{n}{d^{1+\frac{\alpha}{7}}}.$$

*Proof.* This proof is conditioned on all the high-probability events we have shown.

Now consider $W, \widetilde{W} \in \mathbb{R}^{m\times d}$ such that $\|W - W(0)\|_F \le R$ and $\left\|\widetilde{W} - W(0)\right\|_F \le R$. If $\phi$ is a smooth activation, from Lemma D.5 we have

$$\|J_1(W,v) - J_1(W(0),v)\| \lesssim \sqrt{\frac{n}{md}}\|W - W(0)\|_F \le \sqrt{\frac{n}{md}}\cdot\sqrt{d\log d} \lesssim \sqrt{\frac{n\log d}{d^{1+\alpha}}} \ll \sqrt{\frac{n}{d^{1+\frac{\alpha}{2}}}},$$

where we have used $m \gtrsim d^{1+\alpha}$. If $\phi$ is a piece-wise linear activation, from Lemma D.5 we have

$$\|J_1(W,v) - J_1(W(0),v)\| \lesssim \sqrt{\frac{n}{d}}\left(\frac{\|W - W(0)\|^{1/3}}{m^{1/6}} + \left(\frac{\log n}{m}\right)^{1/4}\right)$$
$$\le \sqrt{\frac{n}{d}}\left(\frac{(d\log d)^{1/6}}{m^{1/6}} + \left(\frac{\log n}{m}\right)^{1/4}\right)$$
$$\lesssim \sqrt{\frac{n}{d}}\cdot\frac{(d\log d)^{1/6}}{d^{1/6+\alpha/6}}$$
$$\ll \frac{\sqrt{n}}{d^{\frac{1}{2}+\frac{\alpha}{7}}}.$$

Hence we always have $\|J_1(W,v) - J_1(W(0),v)\| \le \frac{\sqrt{n}}{d^{\frac{1}{2}+\frac{\alpha}{7}}}$. Similarly, we have $\left\|J_1(\widetilde{W},v) - J_1(W(0),v)\right\| \le \frac{\sqrt{n}}{d^{\frac{1}{2}+\frac{\alpha}{7}}}$.

Note that from Proposition D.2 and Claim 3.1 we know

$$\left\|J_1(W(0),v)J_1(W(0),v)^\top\right\| \lesssim \left\|\Theta^{\mathrm{lin1}}\right\| + \frac{n}{d^{1+\alpha}} \lesssim \frac{n}{d} + \frac{n}{d^{1+\alpha}} \lesssim \frac{n}{d},$$

which implies $\|J_1(W(0),v)\| \lesssim \sqrt{\frac{n}{d}}$. It follows that $\|J_1(W,v)\| \lesssim \sqrt{\frac{n}{d}} + \frac{\sqrt{n}}{d^{\frac{1}{2}+\frac{\alpha}{7}}} \lesssim \sqrt{\frac{n}{d}}$ and $\left\|J_1(\widetilde{W},v)\right\| \lesssim \sqrt{\frac{n}{d}}$. Then we have

$$\left\|J_1(W,v)J_1(\widetilde{W},v)^\top - J_1(W(0),v)J_1(W(0),v)^\top\right\|$$
$$\le \|J_1(W,v)\|\cdot\left\|J_1(\widetilde{W},v) - J_1(W(0),v)\right\| + \|J_1(W(0),v)\|\cdot\|J_1(W,v) - J_1(W(0),v)\|$$
$$\lesssim \sqrt{\frac{n}{d}}\cdot\frac{\sqrt{n}}{d^{\frac{1}{2}+\frac{\alpha}{7}}} + \sqrt{\frac{n}{d}}\cdot\frac{\sqrt{n}}{d^{\frac{1}{2}+\frac{\alpha}{7}}}$$
$$\lesssim \frac{n}{d^{1+\frac{\alpha}{7}}}.$$

Combining the above inequality with Proposition D.2, we obtain

$$\left\|J_1(W,v)J_1(\widetilde{W},v)^\top - \Theta^{\mathrm{lin1}}\right\| \lesssim \frac{n}{d^{1+\frac{\alpha}{7}}} + \frac{n}{d^{1+\alpha}} \lesssim \frac{n}{d^{1+\frac{\alpha}{7}}},$$

completing the proof. $\qquad\qquad\square$

Finally, we can instantiate Theorem C.2 to conclude the proof of (7):

**Proposition D.7.** *There exists a universal constant $c > 0$ such that with high probability, for all $0 \le t \le T = c \cdot \frac{d \log d}{\eta_1}$ simultaneously, we have:*

- $\frac{1}{n} \sum_{i=1}^n (f_t^1(\boldsymbol{x}_i) - f_t^{\mathrm{lin1}}(\boldsymbol{x}_i))^2 \le d^{-\frac{\alpha}{4}}$;

- $\|\boldsymbol{W}(t) - \boldsymbol{W}(0)\|_F \le \sqrt{d \log d}$, $\|\boldsymbol{\beta}(t)\| \le \sqrt{d \log d}$.

*Proof.* Let $R = \sqrt{d \log d}$ and $\epsilon = C \frac{n}{d^{1 + \frac{\alpha}{7}}}$ for a sufficiently large universal constant $C > 0$. From Lemma D.6 we know that Assumption C.1 is satisfied with parameters $\epsilon$ and $R$. (Note that $\epsilon \ll \frac{n}{d} \lesssim \|\boldsymbol{\Theta}^{\mathrm{lin1}}\|$.) Also we have $R^2 \epsilon \ll n$, and $\eta_1 \ll d \lesssim \frac{n}{\|\boldsymbol{\Theta}^{\mathrm{lin1}}\|}$. Therefore, we can apply Theorem C.2 and obtain for all $0 \le t \le T$:

$$\sqrt{\sum_{i=1}^n (f_t^1(\boldsymbol{x}_i) - f_t^{\mathrm{lin1}}(\boldsymbol{x}_i))^2} \lesssim \frac{\eta_1 t \epsilon}{\sqrt{n}} \lesssim \frac{d \log d \cdot \frac{n}{d^{1 + \frac{\alpha}{7}}}}{\sqrt{n}} = \frac{\sqrt{n} \log d}{d^{\frac{\alpha}{7}}} \ll \frac{\sqrt{n}}{d^{\frac{\alpha}{8}}},$$

which implies

$$\frac{1}{n} \sum_{i=1}^n (f_t^1(\boldsymbol{x}_i) - f_t^{\mathrm{lin1}}(\boldsymbol{x}_i))^2 \le d^{-\frac{\alpha}{4}}.$$

Furthermore, Theorem C.2 also tells us $\|\boldsymbol{W}(t) - \boldsymbol{W}(0)\|_F \le \sqrt{d \log d}$ and $\|\boldsymbol{\beta}(t)\| \le \sqrt{d \log d}$. $\quad\square$

### D.3.3 Agreement on Distribution

Now we prove the second part of Theorem 3.2, (8), which guarantees the agreement between $f_t^1$ and $f_t^{\mathrm{lin1}}$ on the entire distribution $\mathcal{D}$. As usual, we will be conditioned on all the high-probability events unless otherwise noted.

Given the initialization $(\boldsymbol{W}(0), \boldsymbol{v})$ (recall that $\boldsymbol{v} = \boldsymbol{v}(0)$ is always fixed), we define an auxiliary model $f^{\mathrm{aux1}}(\boldsymbol{x}; \boldsymbol{W})$ which is the first-order Taylor approximation of the neural network $f(\boldsymbol{x}; \boldsymbol{W}, \boldsymbol{v})$ around $\boldsymbol{W}(0)$:

$$
\begin{aligned}
f^{\mathrm{aux1}}(\boldsymbol{x}; \boldsymbol{W}) &:= f(\boldsymbol{x}; \boldsymbol{W}(0), \boldsymbol{v}) + \langle \boldsymbol{W} - \boldsymbol{W}(0), \nabla_{\boldsymbol{W}} f(\boldsymbol{x}; \boldsymbol{W}(0), \boldsymbol{v}) \rangle \\
&= \langle \boldsymbol{W} - \boldsymbol{W}(0), \nabla_{\boldsymbol{W}} f(\boldsymbol{x}; \boldsymbol{W}(0), \boldsymbol{v}) \rangle \\
&= \langle \mathrm{vec}\,(\boldsymbol{W} - \boldsymbol{W}(0)), \boldsymbol{\rho}_1(\boldsymbol{x}) \rangle,
\end{aligned}
$$

where $\boldsymbol{\rho}_1(\boldsymbol{x}) := \nabla_{\boldsymbol{W}} f(\boldsymbol{x}; \boldsymbol{W}(0), \boldsymbol{v})$. Above we have used $f(\boldsymbol{x}; \boldsymbol{W}(0), \boldsymbol{v}) = 0$ according to the symmetric initialization (3). We also denote $f_t^{\mathrm{aux1}}(\boldsymbol{x}) := f^{\mathrm{aux1}}(\boldsymbol{x}; \boldsymbol{W}(t))$ for all $t$.

For all models, we write their predictions on all training datapoints concisely as $f_t^1(\boldsymbol{X}), f_t^{\mathrm{lin1}}(\boldsymbol{X}), f_t^{\mathrm{aux1}}(\boldsymbol{X}) \in \mathbb{R}^n$. From Proposition D.7 we know that $f_t^1$ and $f_t^{\mathrm{lin1}}$ make similar predictions on $\boldsymbol{X}$ (for all $t \le T$ simultaneously):

$$\left\| f_t^1(\boldsymbol{X}) - f_t^{\mathrm{lin1}}(\boldsymbol{X}) \right\| \le \frac{\sqrt{n}}{d^{\frac{\alpha}{8}}}. \tag{34}$$

We can also related the predictions of $f_t^1$ and $f_t^{\mathrm{aux1}}$ by the fundamental theorem for line integrals:

$$
\begin{aligned}
f_t^1(\boldsymbol{X}) &= f_t^1(\boldsymbol{X}) - f_0^1(\boldsymbol{X}) = \boldsymbol{J}_1(\boldsymbol{W}(0) \to \boldsymbol{W}(t), \boldsymbol{v}) \cdot \mathrm{vec}\,(\boldsymbol{W}(t) - \boldsymbol{W}(0)), \\
f_t^{\mathrm{aux1}}(\boldsymbol{X}) &= f_t^{\mathrm{aux1}}(\boldsymbol{X}) - f_0^{\mathrm{aux1}}(\boldsymbol{X}) = \boldsymbol{J}_1(\boldsymbol{W}(0), \boldsymbol{v}) \cdot \mathrm{vec}\,(\boldsymbol{W}(t) - \boldsymbol{W}(0)),
\end{aligned}
\tag{35}
$$

where $\boldsymbol{J}_1(\boldsymbol{W}(0) \to \boldsymbol{W}(t), \boldsymbol{v}) := \int_0^1 \boldsymbol{J}_1(\boldsymbol{W}(0) + x(\boldsymbol{W}(t) - \boldsymbol{W}(0)), \boldsymbol{v}) dx$. Since $\|\boldsymbol{W}(t) - \boldsymbol{W}(0)\|_F \le \sqrt{d \log d}$ according to Proposition D.7, we can use Lemma D.5 in the same way as in the proof of Lemma D.6 and obtain

$$\|\boldsymbol{J}_1(\boldsymbol{W}(0) \to \boldsymbol{W}(t), \boldsymbol{v}) - \boldsymbol{J}_1(\boldsymbol{W}(0), \boldsymbol{v})\| \le \frac{\sqrt{n}}{d^{\frac{1}{2} + \frac{\alpha}{7}}}.$$

Then it follows from (35) that

$$\left\| f_t^1(\boldsymbol{X}) - f_t^{\mathrm{aux1}}(\boldsymbol{X}) \right\| = \left\| (\boldsymbol{J}_1(\boldsymbol{W}(0) \to \boldsymbol{W}(t), \boldsymbol{v}) - \boldsymbol{J}_1(\boldsymbol{W}(0), \boldsymbol{v})) \cdot \mathrm{vec}\left( \boldsymbol{W}(t) - \boldsymbol{W}(0) \right) \right\|$$
$$\leq \frac{\sqrt{n}}{d^{\frac{1}{2}+\frac{\alpha}{7}}} \cdot \sqrt{d \log d}$$
$$\leq \frac{\sqrt{n}}{d^{\frac{\alpha}{8}}}.$$

$$(36)$$

Combining (34) and (36) we know

$$\left\| f_t^{\mathrm{aux1}}(\boldsymbol{X}) - f_t^{\mathrm{lin1}}(\boldsymbol{X}) \right\| \lesssim \frac{\sqrt{n}}{d^{\frac{\alpha}{8}}}.$$

This implies

$$\frac{1}{n} \sum_{i=1}^{n} \min\left\{ \left( f_t^{\mathrm{aux1}}(\boldsymbol{x}_i) - f_t^{\mathrm{lin1}}(\boldsymbol{x}_i) \right)^2, 1 \right\} \leq \frac{1}{n} \sum_{i=1}^{n} \left( f_t^{\mathrm{aux1}}(\boldsymbol{x}_i) - f_t^{\mathrm{lin1}}(\boldsymbol{x}_i) \right)^2 \lesssim d^{-\frac{\alpha}{4}}.$$

Next we will translate these guarantees on the training data to the distribution $\mathcal{D}$ using Rademacher complexity. Note that the model $f_t^{\mathrm{aux1}}(\boldsymbol{x}) - f_t^{\mathrm{lin1}}(\boldsymbol{x})$ is by definition linear in the feature $\begin{bmatrix} \boldsymbol{\rho}_1(\boldsymbol{x}) \\ \boldsymbol{\psi}_1(\boldsymbol{x}) \end{bmatrix}$, and it belongs to the following function class (for all $t \leq T$):

$$\mathcal{F} := \left\{ \boldsymbol{x} \mapsto \boldsymbol{a}^\top \begin{bmatrix} \boldsymbol{\rho}_1(\boldsymbol{x}) \\ \boldsymbol{\psi}_1(\boldsymbol{x}) \end{bmatrix} : \|\boldsymbol{a}\| \leq 2\sqrt{d \log d} \right\}.$$

This is because we have $\|\mathrm{vec}\left( \boldsymbol{W}(t) - \boldsymbol{W}(0) \right)\| \leq \sqrt{d \log d}$ and $\|\boldsymbol{\beta}(t)\| \leq \sqrt{d \log d}$ for all $t \leq T$. Using the well-known bound on the empirical Rademacher complexity of a linear function class with bounded $\ell_2$ norm (see e.g. Bartlett and Mendelson [2002]), we can bound the empirical Rademacher complexity of the function class $\mathcal{F}$:

$$\hat{\mathcal{R}}_{\boldsymbol{X}}(\mathcal{F}) := \frac{1}{n} \mathbb{E}_{\varepsilon_1, \dots, \varepsilon_n \overset{\mathrm{i.i.d.}}{\sim} \mathsf{Unif}(\{\pm 1\})} \left[ \sup_{h \in \mathcal{F}} \sum_{i=1}^{n} \varepsilon_i h(\boldsymbol{x}_i) \right]$$
$$\lesssim \frac{\sqrt{d \log d}}{n} \sqrt{ \sum_{i=1}^{n} \left( \|\boldsymbol{\rho}_1(\boldsymbol{x}_i)\|^2 + \|\boldsymbol{\psi}_1(\boldsymbol{x}_i)\|^2 \right) }$$

$$(37)$$

$$= \frac{\sqrt{d \log d}}{n} \sqrt{ \mathrm{Tr}[\boldsymbol{\Theta}_1(\boldsymbol{W}(0), \boldsymbol{v})] + \mathrm{Tr}[\boldsymbol{\Theta}^{\mathrm{lin1}}] }.$$

Since $\phi'$ is bounded and $\frac{\|\boldsymbol{x}_i\|^2}{d} = O(1)$ ($\forall i \in [n]$), we can bound

$$\mathrm{Tr}[\boldsymbol{\Theta}_1(\boldsymbol{W}(0), \boldsymbol{v})] = \sum_{i=1}^{n} \frac{1}{m} \sum_{r=1}^{m} \phi'\left( \boldsymbol{w}_r(0)^\top \boldsymbol{x}_i / \sqrt{d} \right)^2 \cdot \frac{\|\boldsymbol{x}_i\|^2}{d} \lesssim n,$$

and

$$\mathrm{Tr}[\boldsymbol{\Theta}^{\mathrm{lin1}}] = \sum_{i=1}^{n} \left( \zeta^2 \frac{\|\boldsymbol{x}_i\|^2}{d} + \frac{\nu^2}{d} \right) \lesssim n.$$

Therefore we have

$$\hat{\mathcal{R}}_{\boldsymbol{X}}(\mathcal{F}) \lesssim \frac{\sqrt{d \log d}}{n} \sqrt{n} = \sqrt{\frac{d \log d}{n}}.$$

Now using the standard generalization bound via Rademacher complexity (see e.g. Mohri et al. [2012]), and noticing that the function $z \mapsto \min\{z^2, 1\}$ is 2-Lipschitz and bounded in $[0, 1]$, we have with high probability, for all $t \leq T$ simultaneously,

$$\mathbb{E}_{\boldsymbol{x} \sim \mathcal{D}} \left[ \min\left\{ \left( f_t^{\mathrm{aux1}}(\boldsymbol{x}) - f_t^{\mathrm{lin1}}(\boldsymbol{x}) \right)^2, 1 \right\} \right]$$

$$\leq \frac{1}{n} \sum_{i=1}^{n} \min \left\{ \left( f_t^{\mathrm{aux1}}(\boldsymbol{x}_i) - f_t^{\mathrm{lin1}}(\boldsymbol{x}_i) \right)^2, 1 \right\} + O\left( \sqrt{\frac{d \log d}{n}} \right) + O\left( \frac{1}{\sqrt{n}} \right)$$

$$\lesssim d^{-\frac{\alpha}{4}} + \sqrt{\frac{d \log d}{d^{1+\alpha}}} \qquad\qquad\qquad (n \gtrsim d^{1+\alpha})$$

$$\lesssim d^{-\frac{\alpha}{4}}. \tag{38}$$

Therefore we have shown that $f_t^{\mathrm{aux1}}$ and $f_t^{\mathrm{lin1}}$ are close on the distribution $\mathcal{D}$ for all $t \leq T$. To complete the proof, we need to show that $f_t^1$ and $f_t^{\mathrm{aux1}}$ are close on $\mathcal{D}$. For this, we take an imaginary set of test datapoints $\widetilde{\boldsymbol{x}}_1, \ldots, \widetilde{\boldsymbol{x}}_n \overset{\mathrm{i.i.d.}}{\sim} \mathcal{D}$, which are independent of the training samples. Let $\widetilde{\boldsymbol{X}} \in \mathbb{R}^{n \times d}$ be the corresponding test data matrix. Since the test data are from the same distribution $\mathcal{D}$, the concentration properties in Claim 3.1 still hold, and the Jacobian perturbation bounds in Lemma D.5 hold as well. Hence we can apply the exact same arguments in (36) and obtain with high probability for all $t \leq T$,

$$\left\| f_t^1(\widetilde{\boldsymbol{X}}) - f_t^{\mathrm{aux1}}(\widetilde{\boldsymbol{X}}) \right\| \leq \frac{\sqrt{n}}{d^{\frac{\alpha}{8}}},$$

which implies

$$\frac{1}{n} \sum_{i=1}^{n} \min \left\{ (f_t^1(\widetilde{\boldsymbol{x}}_i) - f_t^{\mathrm{aux1}}(\widetilde{\boldsymbol{x}}_i))^2, 1 \right\} \leq d^{-\frac{\alpha}{4}}.$$

Now notice that $f_t^1$ and $f_t^{\mathrm{aux1}}$ are independent of $\widetilde{\boldsymbol{X}}$. Thus, by Hoeffding inequality, for each $t$, with probability at least $1 - \delta$ we have

$$\mathbb{E}_{\boldsymbol{x} \sim \mathcal{D}} \left[ \min \left\{ (f_t^1(\boldsymbol{x}) - f_t^{\mathrm{aux1}}(\boldsymbol{x}))^2, 1 \right\} \right]$$

$$\leq \frac{1}{n} \sum_{i=1}^{n} \min \left\{ (f_t^1(\widetilde{\boldsymbol{x}}_i) - f_t^{\mathrm{aux1}}(\widetilde{\boldsymbol{x}}_i))^2, 1 \right\} + O\left( \sqrt{\frac{\log \frac{1}{\delta}}{n}} \right)$$

$$\lesssim d^{-\frac{\alpha}{4}} + \sqrt{\frac{\log \frac{1}{\delta}}{n}}.$$

Then letting $\delta = \frac{1}{100T}$ and taking a union bound over $t \leq T$, we obtain that with high probability, for all $t \leq T$ simultaneously,

$$\mathbb{E}_{\boldsymbol{x} \sim \mathcal{D}} \left[ \min \left\{ (f_t^1(\boldsymbol{x}) - f_t^{\mathrm{aux1}}(\boldsymbol{x}))^2, 1 \right\} \right] \lesssim d^{-\frac{\alpha}{4}} + \sqrt{\frac{\log T}{n}}. \tag{39}$$

Therefore we have proved that $f_t^1$ and $f_t^{\mathrm{aux1}}$ are close on $\mathcal{D}$. Finally, combining (38) and (39), we know that with high probability, for all $t \leq T$,

$$\mathbb{E}_{\boldsymbol{x} \sim \mathcal{D}} \left[ \min \left\{ (f_t^1(\boldsymbol{x}) - f_t^{\mathrm{lin1}}(\boldsymbol{x}))^2, 1 \right\} \right] \lesssim d^{-\frac{\alpha}{4}} + \sqrt{\frac{\log T}{n}}.$$

Here we have used $\min\{(a+b)^2, 1\} \leq 2(\min\{a^2, 1\} + \min\{b^2, 1\})$ ($\forall a, b \in \mathbb{R}$). Therefore we have finished the proof of (8). The proof of Theorem 3.2 is done.

### D.4  Proof of Corollary 3.3 (Training the First Layer, Well-Conditioned Data)

*Proof of Corollary 3.3.* We continue to adopt the notation in Appendix D.3.3 to use $f_t^1(\boldsymbol{X})$, $f_t^{\mathrm{lin1}}(\boldsymbol{X})$, etc. to represent the predictions of a model on all $n$ training datapoints. Given Theorem 3.2, it suffices to prove that $f_T^{\mathrm{lin1}}$ and $f_*^{\mathrm{lin1}}$ are close in the following sense:

$$\frac{1}{n} \sum_{i=1}^{n} \left( f_T^{\mathrm{lin1}}(\boldsymbol{x}_i) - f_*^{\mathrm{lin1}}(\boldsymbol{x}_i) \right)^2 \lesssim d^{-\Omega(\alpha)}, \tag{40}$$

$$\mathbb{E}_{\boldsymbol{x} \sim \mathcal{D}} \left[ \min\{ (f_T^{\mathrm{lin1}}(\boldsymbol{x}) - f_*^{\mathrm{lin1}}(\boldsymbol{x}))^2, 1 \} \right] \lesssim d^{-\Omega(\alpha)}. \tag{41}$$

According to the linear dynamics (6), we have the following relation (see Claim C.1):

$$f_T^{\mathrm{lin}1}(\boldsymbol{X}) - \boldsymbol{y} = \left(\boldsymbol{I} - \tfrac{1}{n}\eta_1\boldsymbol{\Theta}^{\mathrm{lin}1}\right)^T (-\boldsymbol{y}),$$

$$f_*^{\mathrm{lin}1}(\boldsymbol{X}) - \boldsymbol{y} = \lim_{t\to\infty} \left(\boldsymbol{I} - \tfrac{1}{n}\eta_1\boldsymbol{\Theta}^{\mathrm{lin}1}\right)^t (-\boldsymbol{y}) =: \left(\boldsymbol{I} - \tfrac{1}{n}\eta_1\boldsymbol{\Theta}^{\mathrm{lin}1}\right)^\infty (-\boldsymbol{y}).$$

From the well-conditioned data assumption, it is easy to see that $\boldsymbol{\Theta}^{\mathrm{lin}1}$'s non-zero eigenvalues are all $\Omega(\frac{n}{d})$ with high probability. As a consequence, in all the non-zero eigen-directions of $\boldsymbol{\Theta}^{\mathrm{lin}1}$, the corresponding eigenvalues of $\left(\boldsymbol{I} - \tfrac{1}{n}\eta_1\boldsymbol{\Theta}^{\mathrm{lin}1}\right)^T$ are at most $\left(1 - \tfrac{1}{n}\eta_1 \cdot \Omega(\frac{n}{d})\right)^T \le \exp\left(-\Omega\left(\frac{\eta_1 T}{d}\right)\right) = \exp\left(-\Omega(\log d)\right) = d^{-\Omega(1)}$. This implies

$$\left\| f_T^{\mathrm{lin}1}(\boldsymbol{X}) - f_*^{\mathrm{lin}1}(\boldsymbol{X}) \right\| \le \left\| \left(\boldsymbol{I} - \tfrac{1}{n}\eta_1\boldsymbol{\Theta}^{\mathrm{lin}1}\right)^T - \left(\boldsymbol{I} - \tfrac{1}{n}\eta_1\boldsymbol{\Theta}^{\mathrm{lin}1}\right)^\infty \right\| \cdot \|\boldsymbol{y}\| \lesssim d^{-\Omega(1)}\sqrt{n},$$

which completes the proof of (40).

To prove (41), we further apply the standard Rademacher complexity argument (similar to Appendix D.3.3). For this we just need to bound the $\ell_2$ norm of the parameters, $\|\boldsymbol{\beta}(T)\|$ and $\|\boldsymbol{\beta}_*\|$. From Proposition D.7, we already have $\|\boldsymbol{\beta}(T)\| \le \sqrt{d\log d}$. Regarding $\boldsymbol{\beta}_*$, we can directly write down its expression

$$\boldsymbol{\beta}_* = (\boldsymbol{\Psi}_1^\top\boldsymbol{\Psi}_1)^\dagger\boldsymbol{\Psi}_1^\top\boldsymbol{y}.$$

Here $\boldsymbol{\Psi}_1$ is the feature matrix defined in (19), and $^\dagger$ stands for the Moore–Penrose pseudo-inverse. Recall that $\boldsymbol{\Theta}^{\mathrm{lin}1} = \boldsymbol{\Psi}_1\boldsymbol{\Psi}_1^\top$. Notice that every non-zero singular value of $(\boldsymbol{\Psi}_1^\top\boldsymbol{\Psi}_1)^\dagger\boldsymbol{\Psi}_1^\top$ is the inverse of a non-zero singular value of $\boldsymbol{\Psi}_1$, and that every non-zero singular value of $\boldsymbol{\Psi}_1$ is $\Omega(\sqrt{\frac{n}{d}})$. This implies $\left\|(\boldsymbol{\Psi}_1^\top\boldsymbol{\Psi}_1)^\dagger\boldsymbol{\Psi}_1^\top\right\| \lesssim \sqrt{\frac{d}{n}}$. Hence we have

$$\|\boldsymbol{\beta}_*\| \lesssim \sqrt{\tfrac{d}{n}}\sqrt{n} = \sqrt{d}.$$

Therefore we can apply the standard Rademacher complexity argument and conclude the proof of (41). □

### D.5 Proof of Theorem 3.5 (Training the Second Layer)

Since the first layer is kept fixed in this case, we let $\boldsymbol{W} = \boldsymbol{W}(0)$ for notational convenience. Similar to the proof of Theorem 3.2 in Appendix D.3, we still divide the proof into 3 parts: analyzing the NTK at initialization (which is also the NTK throughout training in this case), proving the agreement on training data, and proving the agreement on the distribution.

It is easy to see from the definition of $\boldsymbol{\Theta}^{\mathrm{lin}2}$ in (20) and Claim 3.1 that if $\vartheta_0 \neq 0$, then $\left\|\boldsymbol{\Theta}^{\mathrm{lin}2}\right\| = O(n)$ with high probability, and if $\vartheta_0 = 0$, then $\left\|\boldsymbol{\Theta}^{\mathrm{lin}2}\right\| = O(\frac{n\log n}{d})$ with high probability. As we will see in the proof, this is why we distinguish these two cases in Theorem 3.5.

#### D.5.1 The NTK at Initialization

**Proposition D.8.** *With high probability over the random initialization $\boldsymbol{W}$ and the training data $\boldsymbol{X}$, we have*

$$\left\|\boldsymbol{\Theta}_2(\boldsymbol{W}) - \boldsymbol{\Theta}^{\mathrm{lin}2}\right\| \lesssim \frac{n}{d^{1+\frac{\alpha}{3}}}.$$

To prove Proposition D.8, we will prove $\boldsymbol{\Theta}_2(\boldsymbol{W})$ is close to its expectation $\boldsymbol{\Theta}_2^*$ (defined in (18)), and then prove $\boldsymbol{\Theta}_2^*$ is close to $\boldsymbol{\Theta}^{\mathrm{lin}2}$. We do these steps in the next two propositions.

**Proposition D.9.** *With high probability over the training data $\boldsymbol{X}$, we have*

$$\left\|\boldsymbol{\Theta}_2^* - \boldsymbol{\Theta}^{\mathrm{lin}2}\right\| \lesssim \frac{n}{d^{1+\alpha}}.$$

*Proof.* We will be conditioned on the high probability events stated in Claim 3.1.

By the definition of $\boldsymbol{\Theta}_2^*$, we know

$$[\boldsymbol{\Theta}_2^*]_{i,j} = \mathbb{E}_{\boldsymbol{w}\sim\mathcal{N}(\boldsymbol{0},\boldsymbol{I})}\left[\phi(\boldsymbol{w}^\top\boldsymbol{x}_i/\sqrt{d})\phi(\boldsymbol{w}^\top\boldsymbol{x}_j/\sqrt{d})^\top\right], \qquad i,j \in [n].$$

We define

$$\Gamma(a,b,c) := \mathbb{E}_{(z_1,z_2)\sim\mathcal{N}(\mathbf{0},\mathbf{\Lambda})}[\phi(z_1)\phi(z_2)], \text{ where } \mathbf{\Lambda} = \begin{pmatrix} a^2 & c \\ c & b^2 \end{pmatrix}, \quad a \geq 0, b \geq 0, |c| \leq ab.$$

Then we can write

$$[\mathbf{\Theta}_2^*]_{i,j} = \Gamma\left(\frac{\|\boldsymbol{x}_i\|}{\sqrt{d}}, \frac{\|\boldsymbol{x}_j\|}{\sqrt{d}}, \frac{\boldsymbol{x}_i^\top \boldsymbol{x}_j}{d}\right).$$

Denote $e_i := \frac{\|\boldsymbol{x}_i\|}{\sqrt{d}} - 1$ and $s_{i,j} := \frac{\boldsymbol{x}_i^\top \boldsymbol{x}_j}{d}$. Below we consider the diagonal and off-diagonal entries of $\mathbf{\Theta}_2^*$ separately.

For $i \neq j$, we do a Taylor expansion of $\Gamma$ around $(1,1,0)$:

$$[\mathbf{\Theta}_2^*]_{i,j}$$

$$= \Gamma(1,1,0) + \nabla\Gamma(1,1,0)^\top \begin{bmatrix} e_i \\ e_j \\ s_{i,j} \end{bmatrix} + \frac{1}{2}[e_i, e_j, s_{i,j}] \cdot \nabla^2\Gamma(1,1,0) \cdot \begin{bmatrix} e_i \\ e_j \\ s_{i,j} \end{bmatrix} + O(|e_i|^3 + |e_j|^3 + |s_{i,j}|^3)$$

$$= \vartheta_0^2 + \vartheta_0\vartheta_1(e_i + e_j) + \zeta^2 s_{i,j} + \vartheta_0\vartheta_2(e_i^2 + e_j^2) + \vartheta_1^2 e_i e_j + \frac{1}{2}\vartheta_1^2 s_{i,j}^2 + \gamma s_{i,j}(e_i + e_j) \pm \tilde{O}\left(\frac{1}{d^{3/2}}\right)$$

$$= (\vartheta_0 + \vartheta_1 e_i + \vartheta_2 e_i^2)(\vartheta_0 + \vartheta_1 e_j + \vartheta_2 e_j^2) - \vartheta_1\vartheta_2(e_i e_j^2 + e_i^2 e_j) - \vartheta_2^2 e_i^2 e_j^2 + \zeta^2 s_{i,j} + \frac{1}{2}\vartheta_1^2 s_{i,j}^2$$

$$\qquad + \gamma s_{i,j}(e_i + e_j) \pm \tilde{O}\left(\frac{1}{d^{3/2}}\right)$$

$$= [\boldsymbol{q}]_i [\boldsymbol{q}]_j \pm \tilde{O}\left(\frac{1}{d^{3/2}}\right) \pm \tilde{O}\left(\frac{1}{d^2}\right) + \zeta^2 s_{i,j} + \frac{1}{2}\vartheta_1^2 s_{i,j}^2 + \gamma s_{i,j}(e_i + e_j) \pm \tilde{O}\left(\frac{1}{d^{3/2}}\right)$$

$$= [\boldsymbol{q}]_i [\boldsymbol{q}]_j + \zeta^2 s_{i,j} + \frac{1}{2}\vartheta_1^2 s_{i,j}^2 + \gamma s_{i,j}(e_i + e_j) \pm \tilde{O}\left(\frac{1}{d^{3/2}}\right).$$

Here $\zeta, \vartheta_0, \vartheta_1, \vartheta_2$ are defined in (9), and $\gamma$ is the $(1,3)$-th entry in the Hessian $\nabla^2\Gamma(1,1,0)$ whose specific value is not important to us. Recall that $[\boldsymbol{q}]_i = \vartheta_0 + \vartheta_1 e_i + \vartheta_2 e_i^2$.

On the other hand, by the definition (20) we have

$$\left[\mathbf{\Theta}^{\text{lin2}}\right]_{i,j} = \zeta^2 s_{i,j} + \frac{\nu^2}{2d} + [\boldsymbol{q}]_i [\boldsymbol{q}]_j.$$

It follows that

$$\left[\mathbf{\Theta}_2^* - \mathbf{\Theta}^{\text{lin2}}\right]_{i,j} = \frac{1}{2}\vartheta_1^2 s_{i,j}^2 - \frac{\nu^2}{2d} + \gamma s_{i,j}(e_i + e_j) \pm \tilde{O}\left(\frac{1}{d^{3/2}}\right)$$

$$= \frac{1}{2}\vartheta_1^2\left(s_{i,j}^2 - \frac{\text{Tr}[\mathbf{\Sigma}^2]}{d^2}\right) + \gamma s_{i,j}(e_i + e_j) \pm \tilde{O}\left(\frac{1}{d^{3/2}}\right).$$

Here we have used the definition of $\nu$ in (5). In the proof of Proposition D.4, we have proved that all the error terms above contribute to at most $\tilde{O}(\frac{n}{d^{1.25}})$ in spectral norm. Using the analysis there we get

$$\left\|(\mathbf{\Theta}_2^* - \mathbf{\Theta}^{\text{lin2}})_{\text{off}}\right\| = \tilde{O}\left(\frac{n}{d^{1.25}}\right).$$

Regarding the diagonal entries, it is easy to see that all the diagonal entries in $\mathbf{\Theta}_2^*$ and $\mathbf{\Theta}^{\text{lin2}}$ are $O(1)$, which implies

$$\left\|(\mathbf{\Theta}_2^* - \mathbf{\Theta}^{\text{lin2}})_{\text{diag}}\right\| = O(1).$$

Therefore we have

$$\left\|\mathbf{\Theta}_2^* - \mathbf{\Theta}^{\text{lin2}}\right\| = \tilde{O}\left(\frac{n}{d^{1.25}}\right) + O(1) = O\left(\frac{n}{d^{1+\alpha}}\right),$$

since $n \gtrsim d^{1+\alpha}$ ($0 < \alpha < \frac{1}{4}$). $\qquad\square$

**Proposition D.10.** *With high probability over the random initialization $\boldsymbol{W}$ and the training data $\boldsymbol{X}$, we have*

$$\|\boldsymbol{\Theta}_2(\boldsymbol{W}) - \boldsymbol{\Theta}_2^*\| \lesssim \frac{n}{d^{1+\frac{\alpha}{3}}}.$$

*Proof.* For convenience we denote $\boldsymbol{\Theta}_2 = \boldsymbol{\Theta}_2(\boldsymbol{W})$ in the proof. We will be conditioned on $\boldsymbol{X}$ and on Claim 3.1, and only consider the randomness in $\boldsymbol{W}$. From Proposition D.9 we know that

$$\|\boldsymbol{\Theta}_2^*\| = \begin{cases} \tilde{O}(n/d), & \text{if } \vartheta_0 = \mathbb{E}[\phi(g)] = 0 \\ O(n), & \text{otherwise} \end{cases}.$$

Define $\boldsymbol{\Theta}_2^{(r)} := \phi(\boldsymbol{X}\boldsymbol{w}_r/\sqrt{d})\phi(\boldsymbol{X}\boldsymbol{w}_r/\sqrt{d})^\top$ for each $r \in [m]$. We have $\boldsymbol{\Theta}_2 = \frac{1}{m}\sum_{r=1}^m \boldsymbol{\Theta}_2^{(r)}$. According to the initialization scheme (3), we know that $\boldsymbol{\Theta}_2^{(1)}, \boldsymbol{\Theta}_2^{(2)}, \ldots, \boldsymbol{\Theta}_2^{(m/2)}$ are independent, $\boldsymbol{\Theta}_2^{(m/2+1)}, \boldsymbol{\Theta}_2^{(m/2+2)}, \ldots, \boldsymbol{\Theta}_2^{(m)}$ are independent, and $\mathbb{E}[\boldsymbol{\Theta}_2^{(r)}] = \boldsymbol{\Theta}_2^*$ for all $r \in [m]$.

Since the matrices $\boldsymbol{\Theta}_2^{(r)}$ are possibly unbounded, we will use a variant of the matrix Bernstein inequality for unbounded matrices, which can be found as Proposition 4.1 in Klochkov and Zhivotovskiy [2020]. There are two main steps in order to use this inequality: (i) showing that $\left\|\boldsymbol{\Theta}_2^{(r)} - \boldsymbol{\Theta}_2^*\right\|$ is a sub-exponential random variable for each $r$ and bounding its sub-exponential norm; (ii) bounding the variance $\left\|\sum_{r=1}^{m/2}\mathbb{E}[(\boldsymbol{\Theta}_2^{(r)} - \boldsymbol{\Theta}_2^*)^2]\right\|$. For the first step, we have

$$\begin{aligned}
\left\|\boldsymbol{\Theta}_2^{(r)} - \boldsymbol{\Theta}_2^*\right\| &\leq \left\|\boldsymbol{\Theta}_2^{(r)}\right\| + \|\boldsymbol{\Theta}_2^*\| \\
&= \left\|\phi(\boldsymbol{X}\boldsymbol{w}_r/\sqrt{d})\right\|^2 + O(n) \\
&\lesssim \|\phi(\boldsymbol{0}_n)\|^2 + \left\|\boldsymbol{X}\boldsymbol{w}_r/\sqrt{d}\right\|^2 + n && (\phi \text{ is Lipschitz}) \\
&\lesssim n + \frac{\|\boldsymbol{X}\|^2\|\boldsymbol{w}_r\|^2}{d} \\
&\lesssim n + \frac{n}{d}\|\boldsymbol{w}_r\|^2.
\end{aligned}$$

Since $\|\boldsymbol{w}_r\|^2$ is a $\chi^2$ random variable with $d$ degrees of freedom, it has sub-exponential norm $O(d)$, which implies that the random variable $\left\|\boldsymbol{\Theta}_2^{(r)} - \boldsymbol{\Theta}_2^*\right\|$ has sub-exponential norm $O(n)$.

Next we bound the variance. Let $B > 0$ be a threshold to be determined. We have:

$$\begin{aligned}
&\left\|\mathbb{E}[(\boldsymbol{\Theta}_2^{(r)} - \boldsymbol{\Theta}_2^*)^2]\right\| \\
={}& \left\|\mathbb{E}[(\boldsymbol{\Theta}_2^{(r)})^2] - (\boldsymbol{\Theta}_2^*)^2\right\| \\
\leq{}& \left\|\mathbb{E}[(\boldsymbol{\Theta}_2^{(r)})^2]\right\| + \|\boldsymbol{\Theta}_2^*\|^2 \\
={}& \left\|\mathbb{E}_{\boldsymbol{w}\sim\mathcal{N}(\boldsymbol{0},\boldsymbol{I})}\left[\left\|\phi(\boldsymbol{X}\boldsymbol{w}/\sqrt{d})\right\|^2 \phi(\boldsymbol{X}\boldsymbol{w}/\sqrt{d})\phi(\boldsymbol{X}\boldsymbol{w}/\sqrt{d})^\top\right]\right\| + \|\boldsymbol{\Theta}_2^*\|^2 \\
\leq{}& \left\|\mathbb{E}_{\boldsymbol{w}\sim\mathcal{N}(\boldsymbol{0},\boldsymbol{I})}\left[\mathbb{1}_{\{\|\phi(\boldsymbol{X}\boldsymbol{w}/\sqrt{d})\|\leq B\}}\left\|\phi(\boldsymbol{X}\boldsymbol{w}/\sqrt{d})\right\|^2 \phi(\boldsymbol{X}\boldsymbol{w}/\sqrt{d})\phi(\boldsymbol{X}\boldsymbol{w}/\sqrt{d})^\top\right]\right\| \\
&+ \left\|\mathbb{E}_{\boldsymbol{w}\sim\mathcal{N}(\boldsymbol{0},\boldsymbol{I})}\left[\mathbb{1}_{\{\|\phi(\boldsymbol{X}\boldsymbol{w}/\sqrt{d})\|>B\}}\left\|\phi(\boldsymbol{X}\boldsymbol{w}/\sqrt{d})\right\|^2 \phi(\boldsymbol{X}\boldsymbol{w}/\sqrt{d})\phi(\boldsymbol{X}\boldsymbol{w}/\sqrt{d})^\top\right]\right\| + \|\boldsymbol{\Theta}_2^*\|^2 \\
\leq{}& B^2 \left\|\mathbb{E}_{\boldsymbol{w}\sim\mathcal{N}(\boldsymbol{0},\boldsymbol{I})}\left[\phi(\boldsymbol{X}\boldsymbol{w}/\sqrt{d})\phi(\boldsymbol{X}\boldsymbol{w}/\sqrt{d})^\top\right]\right\| \\
&+ \mathbb{E}_{\boldsymbol{w}\sim\mathcal{N}(\boldsymbol{0},\boldsymbol{I})}\left[\mathbb{1}_{\{\|\phi(\boldsymbol{X}\boldsymbol{w}/\sqrt{d})\|>B\}}\left\|\phi(\boldsymbol{X}\boldsymbol{w}/\sqrt{d})\right\|^4\right] + \|\boldsymbol{\Theta}_2^*\|^2 \\
={}& B^2 \|\boldsymbol{\Theta}_2^*\| + \mathbb{E}_{\boldsymbol{w}\sim\mathcal{N}(\boldsymbol{0},\boldsymbol{I})}\left[\mathbb{1}_{\{\|\phi(\boldsymbol{X}\boldsymbol{w}/\sqrt{d})\|>B\}}\left\|\phi(\boldsymbol{X}\boldsymbol{w}/\sqrt{d})\right\|^4\right] + \|\boldsymbol{\Theta}_2^*\|^2
\end{aligned}$$

$$\leq B^2 \left\|\boldsymbol{\Theta}_2^*\right\| + \left\|\boldsymbol{\Theta}_2^*\right\|^2 + \sqrt{\mathbb{E}_{\boldsymbol{w}\sim\mathcal{N}(\boldsymbol{0},\boldsymbol{I})}\left[\mathbb{1}_{\left\{\left\|\phi(\boldsymbol{X}\boldsymbol{w}/\sqrt{d})\right\|>B\right\}}\right] \cdot \mathbb{E}_{\boldsymbol{w}\sim\mathcal{N}(\boldsymbol{0},\boldsymbol{I})}\left[\left\|\phi(\boldsymbol{X}\boldsymbol{w}/\sqrt{d})\right\|^8\right]}$$

<div align="center">(Cauchy-Schwarz inequality)</div>

$$= B^2 \left\|\boldsymbol{\Theta}_2^*\right\| + \left\|\boldsymbol{\Theta}_2^*\right\|^2 + \sqrt{\Pr_{\boldsymbol{w}\sim\mathcal{N}(\boldsymbol{0},\boldsymbol{I})}\left[\left\|\phi(\boldsymbol{X}\boldsymbol{w}/\sqrt{d})\right\|>B\right] \cdot \mathbb{E}_{\boldsymbol{w}\sim\mathcal{N}(\boldsymbol{0},\boldsymbol{I})}\left[\left\|\phi(\boldsymbol{X}\boldsymbol{w}/\sqrt{d})\right\|^8\right]}.$$

Note that $\left|\left\|\phi(\boldsymbol{X}\boldsymbol{w}/\sqrt{d})\right\| - \left\|\phi(\boldsymbol{X}\boldsymbol{w}'/\sqrt{d})\right\|\right| \lesssim \left\|\boldsymbol{X}\boldsymbol{w}/\sqrt{d} - \boldsymbol{X}\boldsymbol{w}'/\sqrt{d}\right\| \leq \frac{\|\boldsymbol{X}\|}{\sqrt{d}}\|\boldsymbol{w}-\boldsymbol{w}'\| \lesssim \sqrt{\frac{n}{d}}\|\boldsymbol{w}-\boldsymbol{w}'\|$ for all $\boldsymbol{w}, \boldsymbol{w}' \in \mathbb{R}^d$. Then by the standard Lipschitz concentration bound for Gaussian variables (see e.g. Wainwright [2019]) we know that for any $s > 0$:

$$\Pr_{\boldsymbol{w}\sim\mathcal{N}(\boldsymbol{0},\boldsymbol{I})}\left[\left\|\phi(\boldsymbol{X}\boldsymbol{w}/\sqrt{d})\right\| > M + s\right] \leq e^{-\Omega\left(\frac{s^2}{n/d}\right)},$$

where $M := \mathbb{E}_{\boldsymbol{w}\sim\mathcal{N}(\boldsymbol{0},\boldsymbol{I})}\left[\left\|\phi(\boldsymbol{X}\boldsymbol{w}/\sqrt{d})\right\|\right]$ which can be bounded as

$$\begin{aligned}
M^2 &\leq \mathbb{E}_{\boldsymbol{w}\sim\mathcal{N}(\boldsymbol{0},\boldsymbol{I})}\left[\left\|\phi(\boldsymbol{X}\boldsymbol{w}/\sqrt{d})\right\|^2\right] \\
&\lesssim \mathbb{E}_{\boldsymbol{w}\sim\mathcal{N}(\boldsymbol{0},\boldsymbol{I})}\left[\left\|\phi(\boldsymbol{0}_n)\right\|^2 + \left\|\boldsymbol{X}\boldsymbol{w}/\sqrt{d}\right\|^2\right] \\
&\lesssim n + \frac{n}{d}\mathbb{E}_{\boldsymbol{w}\sim\mathcal{N}(\boldsymbol{0},\boldsymbol{I})}\left[\|\boldsymbol{w}\|^2\right] \\
&\lesssim n.
\end{aligned}$$

Thus, letting $\frac{s^2}{n/d} = C\log n$ for a sufficiently large universal constant $C > 0$, we know that with probability at least $1 - n^{-10}$ over $\boldsymbol{w} \sim \mathcal{N}(\boldsymbol{0},\boldsymbol{I})$,

$$\left\|\phi(\boldsymbol{X}\boldsymbol{w}/\sqrt{d})\right\| \leq M + s \lesssim \sqrt{n} + \sqrt{\frac{n}{d}\log n} \lesssim \sqrt{n}.$$

Hence we pick the threshold $B = C'\sqrt{n}$ which is the upper bound above, where $C' > 0$ is a universal constant.

We can also bound

$$\begin{aligned}
&\mathbb{E}_{\boldsymbol{w}\sim\mathcal{N}(\boldsymbol{0},\boldsymbol{I})}\left[\left\|\phi(\boldsymbol{X}\boldsymbol{w}/\sqrt{d})\right\|^8\right] \\
&= \mathbb{E}_{\boldsymbol{w}\sim\mathcal{N}(\boldsymbol{0},\boldsymbol{I})}\left[\left(\sum_{i=1}^n \phi(\boldsymbol{x}_i^\top \boldsymbol{w}/\sqrt{d})^2\right)^4\right] \\
&\lesssim \mathbb{E}_{\boldsymbol{w}\sim\mathcal{N}(\boldsymbol{0},\boldsymbol{I})}\left[\left(\sum_{i=1}^n \left(\phi(0)^2 + (\boldsymbol{x}_i^\top \boldsymbol{w}/\sqrt{d})^2\right)\right)^4\right]
\end{aligned}$$

<div align="center">($\phi$ is Lipschitz & Cauchy-Schwartz inequality)</div>

$$\begin{aligned}
&\lesssim \mathbb{E}_{\boldsymbol{w}\sim\mathcal{N}(\boldsymbol{0},\boldsymbol{I})}\left[\left(n + \sum_{i=1}^n (\boldsymbol{x}_i^\top \boldsymbol{w}/\sqrt{d})^2\right)^4\right] && (|\phi(0)| = O(1)) \\
&\lesssim n^4 + \mathbb{E}_{\boldsymbol{w}\sim\mathcal{N}(\boldsymbol{0},\boldsymbol{I})}\left[\left(\sum_{i=1}^n (\boldsymbol{x}_i^\top \boldsymbol{w}/\sqrt{d})^2\right)^4\right] && \text{(Jensen's inequality)} \\
&= n^4 + n^4\mathbb{E}_{\boldsymbol{w}\sim\mathcal{N}(\boldsymbol{0},\boldsymbol{I})}\left[\left(\frac{1}{n}\sum_{i=1}^n (\boldsymbol{x}_i^\top \boldsymbol{w}/\sqrt{d})^2\right)^4\right] \\
&\leq n^4 + n^4\mathbb{E}_{\boldsymbol{w}\sim\mathcal{N}(\boldsymbol{0},\boldsymbol{I})}\left[\frac{1}{n}\sum_{i=1}^n (\boldsymbol{x}_i^\top \boldsymbol{w}/\sqrt{d})^8\right] && \text{(Jensen's inequality)}
\end{aligned}$$

$$= n^4 + n^3 \sum_{i=1}^{n} \mathbb{E}_{x \sim \mathcal{N}(0, \|\boldsymbol{x}_i\|^2/d)}[x^8]$$

$$\lesssim n^4. \qquad\qquad (\|\boldsymbol{x}_i\|^2/d = O(1))$$

Combining all the above, we get

$$\left\| \mathbb{E}[(\boldsymbol{\Theta}_2^{(r)} - \boldsymbol{\Theta}_2^*)^2] \right\|$$

$$\leq B^2 \|\boldsymbol{\Theta}_2^*\| + \|\boldsymbol{\Theta}_2^*\|^2 + \sqrt{ \Pr_{\boldsymbol{w} \sim \mathcal{N}(\mathbf{0}, \boldsymbol{I})} \left[ \left\| \phi(\boldsymbol{X}\boldsymbol{w}/\sqrt{d}) \right\| > B \right] \cdot \mathbb{E}_{\boldsymbol{w} \sim \mathcal{N}(\mathbf{0}, \boldsymbol{I})} \left[ \left\| \phi(\boldsymbol{X}\boldsymbol{w}/\sqrt{d}) \right\|^8 \right] }$$

$$\lesssim n \|\boldsymbol{\Theta}_2^*\| + \|\boldsymbol{\Theta}_2^*\|^2 + \sqrt{n^{-10} \cdot n^4}$$

$$= n \|\boldsymbol{\Theta}_2^*\| + \|\boldsymbol{\Theta}_2^*\|^2 + n^{-3}.$$

We will discuss two cases separately.

**Case 1: $\vartheta_0 \neq 0$.** Recall that in this case Theorem 3.5 assumes $m \gtrsim d^{2+\alpha}$.

Since $\|\boldsymbol{\Theta}_2^*\| = O(n)$, we have $\left\| \mathbb{E}[(\boldsymbol{\Theta}_2^{(r)} - \boldsymbol{\Theta}_2^*)^2] \right\| \lesssim n^2$ which implies

$$\left\| \sum_{r=1}^{m/2} \mathbb{E}[(\boldsymbol{\Theta}_2^{(r)} - \boldsymbol{\Theta}_2^*)^2] \right\| \lesssim mn^2.$$

Applying Proposition 4.1 in Klochkov and Zhivotovskiy [2020], we know that for any $u \gg \max\{n \log m, n\sqrt{m}\} = n\sqrt{m}$,

$$\Pr \left[ \left\| \sum_{r=1}^{m/2} (\boldsymbol{\Theta}_2^{(r)} - \boldsymbol{\Theta}_2^*) \right\| > u \right] \lesssim n \cdot \exp\left( -\Omega\left( \min\left\{ \frac{u^2}{mn^2}, \frac{u}{n \log m} \right\} \right) \right).$$

Let $u = m \cdot \frac{n}{d^{1+\frac{\alpha}{3}}}$. We can verify $u \gg n\sqrt{m}$ since $m \gtrsim d^{2+\alpha}$. Then we have

$$\Pr \left[ \left\| \sum_{r=1}^{m/2} (\boldsymbol{\Theta}_2^{(r)} - \boldsymbol{\Theta}_2^*) \right\| > m \cdot \frac{n}{d^{1+\frac{\alpha}{3}}} \right] \lesssim n \cdot \exp\left( -\Omega\left( \min\left\{ \frac{m}{d^{2+\frac{2\alpha}{3}}}, \frac{m}{d^{1+\frac{\alpha}{3}} \log m} \right\} \right) \right) \ll 1.$$

Similarly, for the second half of the neurons we also have $\left\| \sum_{r=m/2+1}^{m} (\boldsymbol{\Theta}_2^{(r)} - \boldsymbol{\Theta}_2^*) \right\| \leq m \cdot \frac{n}{d^{1+\frac{\alpha}{3}}}$ with high probability. Therefore we have with high probability,

$$\|\boldsymbol{\Theta}_2 - \boldsymbol{\Theta}_2^*\| \lesssim \frac{n}{d^{1+\frac{\alpha}{3}}}.$$

**Case 2: $\vartheta_0 = 0$.** Recall that in this case Theorem 3.5 assumes $m \gtrsim d^{1+\alpha}$.

Since $\|\boldsymbol{\Theta}_2^*\| = \tilde{O}(n/d)$, we have $\left\| \mathbb{E}[(\boldsymbol{\Theta}_2^{(r)} - \boldsymbol{\Theta}_2^*)^2] \right\| \lesssim n \cdot \tilde{O}(n/d) + \tilde{O}((n/d)^2) + n^{-3} = \tilde{O}(n^2/d)$ which implies

$$\left\| \sum_{r=1}^{m/2} \mathbb{E}[(\boldsymbol{\Theta}_2^{(r)} - \boldsymbol{\Theta}_2^*)^2] \right\| \leq \tilde{O}(mn^2/d) \lesssim \frac{mn^2}{d^{1-\frac{\alpha}{10}}}.$$

Applying Proposition 4.1 in Klochkov and Zhivotovskiy [2020], we know that for any $u \gg \max\{n \log m, n\sqrt{m/d^{1-\frac{\alpha}{10}}}\} = n\sqrt{m/d^{1-\frac{\alpha}{10}}}$,

$$\Pr \left[ \left\| \sum_{r=1}^{m/2} (\boldsymbol{\Theta}_2^{(r)} - \boldsymbol{\Theta}_2^*) \right\| > u \right] \lesssim n \cdot \exp\left( -\Omega\left( \min\left\{ \frac{u^2}{mn^2/d^{1-\frac{\alpha}{10}}}, \frac{u}{n \log m} \right\} \right) \right).$$

Let $u = m \cdot \frac{n}{d^{1+\frac{\alpha}{3}}}$. We can verify $u \gg n\sqrt{m/d^{1-\frac{\alpha}{10}}}$ since $m \gtrsim d^{1+\alpha}$. Then we have

$$\Pr\left[\left\|\sum_{r=1}^{m/2}(\mathbf{\Theta}_2^{(r)} - \mathbf{\Theta}_2^*)\right\| > m \cdot \frac{n}{d^{1+\frac{\alpha}{3}}}\right] \lesssim n \cdot \exp\left(-\Omega\left(\min\left\{\frac{m}{d^{1+0.77\alpha}}, \frac{m}{d^{1+\frac{\alpha}{3}}\log m}\right\}\right)\right) \ll 1.$$

Similarly, for the second half of the neurons we also have $\left\|\sum_{r=m/2+1}^{m}(\mathbf{\Theta}_2^{(r)} - \mathbf{\Theta}_2^*)\right\| \le m \cdot \frac{n}{d^{1+\frac{\alpha}{3}}}$ with high probability. Therefore we have with high probability,

$$\|\mathbf{\Theta}_2 - \mathbf{\Theta}_2^*\| \lesssim \frac{n}{d^{1+\frac{\alpha}{3}}}.$$

The proof is completed. $\qquad\qquad\square$

Combining Propositions D.9 and D.10 directly gives Proposition D.8.

### D.5.2   Agreement on Training Data

To prove the agreement between $f_t^2$ and $f_t^{\mathrm{lin}2}$ on training data for all $t \le T = c \cdot \frac{d\log d}{\eta_2}$, we still apply Theorem C.2. This case is much easier than training the first layer (Appendix D.3.2), since the Jacobian for the second layer does not change during training, and thus Proposition D.8 already verifies Assumption C.1. Therefore we can directly instantiate Theorem C.2 with $\epsilon = C\frac{n}{d^{1+\frac{\alpha}{3}}}$ (for a sufficiently large constant $C$) and $R = \sqrt{d\log d}$, which gives (notice that the choice of $\eta_2$ in Theorem 3.5 also satisfies the condition in Theorem C.2)

$$\sqrt{\sum_{i=1}^{n}(f_t^2(\mathbf{x}_i) - f_t^{\mathrm{lin}2}(\mathbf{x}_i))^2} \lesssim \frac{\eta_2 t\epsilon}{\sqrt{n}} \lesssim \frac{d\log d \cdot \frac{n}{d^{1+\frac{\alpha}{3}}}}{\sqrt{n}} = \frac{\sqrt{n}\log d}{d^{\frac{\alpha}{3}}} \ll \frac{\sqrt{n}}{d^{\frac{\alpha}{4}}},$$

i.e.,

$$\frac{1}{n}\sum_{i=1}^{n}(f_t^2(\mathbf{x}_i) - f_t^{\mathrm{lin}2}(\mathbf{x}_i))^2 \le d^{-\frac{\alpha}{2}}.$$

This proves the first part in Theorem 3.5.

Note that Theorem C.2 also tells us $\|\mathbf{v}(t) - \mathbf{v}(0)\| \le \sqrt{d\log d}$ and $\|\boldsymbol{\gamma}(t)\| \le \sqrt{d\log d}$, which will be useful for proving the guarantee on the distribution $\mathcal{D}$.

### D.5.3   Agreement on Distribution

Now we prove the second part in Theorem 3.5, which is the agreement between $f_t^2$ and $f_t^{\mathrm{lin}2}$ on the distribution $\mathcal{D}$. The proof is similar to the case of training the first layer (Appendix D.3.3), but our case here is again simpler. In particular, we do not need to define an auxiliary model anymore because $f(\mathbf{x}; \mathbf{W}, \mathbf{v})$ is already linear in the parameters $\mathbf{v}$. Now that $f_t^2 - f_t^{\mathrm{lin}2}$ is a linear model (in some feature space) with bounded parameters, we can bound the Rademacher complexity of the linear function class it belongs to, similar to Appendix D.3.3. Similar to (37), we can bound the Rademacher complexity by

$$\frac{\sqrt{d\log d}}{n}\sqrt{\mathrm{Tr}[\mathbf{\Theta}_2(\mathbf{W})] + \mathrm{Tr}[\mathbf{\Theta}^{\mathrm{lin}2}]}.$$

Next we bound the above two traces. First, we have

$$[\mathbf{\Theta}_2(\mathbf{W})]_{i,i} = \frac{1}{m}\sum_{r=1}^{m}\phi(\mathbf{x}_i^\top\mathbf{w}_r/\sqrt{d})^2 \lesssim \frac{1}{m}\sum_{r=1}^{m}\left(\phi(0)^2 + (\mathbf{x}_i^\top\mathbf{w}_r/\sqrt{d})^2\right)$$

$$= 1 + \frac{1}{dm}\sum_{r=1}^{m}(\mathbf{x}_i^\top\mathbf{w}_r)^2 \lesssim 1 + \frac{1}{dm}(dm + \log n) \lesssim 1$$

with high probability for all $i \in [n]$ together. Here we have used the standard tail bound for $\chi^2$ random variables and a union bound over $i \in [n]$. Hence we have $\text{Tr}[\boldsymbol{\Theta}_2(\boldsymbol{W})] \lesssim n$. For the second trace, we have

$$\text{Tr}[\boldsymbol{\Theta}^{\text{lin}2}] = \sum_{i=1}^{n} \left( \zeta^2 \frac{\|\boldsymbol{x}_i\|^2}{d} + \frac{\nu^2}{2d} + [\boldsymbol{q}]_i^2 \right) \lesssim n$$

with high probability. Therefore we can bound the Rademacher complexity by $\sqrt{\frac{d \log d}{n}}$. Then we can conclude the agreement guarantee on the distribution $\mathcal{D}$, i.e., for all $t \leq T$ simultaneously,

$$
\begin{aligned}
&\mathbb{E}_{\boldsymbol{x} \sim \mathcal{D}} \left[ \min \left\{ \left( f_t^2(\boldsymbol{x}) - f_t^{\text{lin}2}(\boldsymbol{x}) \right)^2, 1 \right\} \right] \\
&\leq \frac{1}{n} \sum_{i=1}^{n} \min \left\{ \left( f_t^2(\boldsymbol{x}_i) - f_t^{\text{lin}2}(\boldsymbol{x}_i) \right)^2, 1 \right\} + O\left( \sqrt{\frac{d \log d}{n}} \right) + O\left( \frac{1}{\sqrt{n}} \right) \\
&\lesssim d^{-\frac{\alpha}{2}} + \sqrt{\frac{d \log d}{d^{1+\alpha}}} \qquad\qquad\qquad\qquad\qquad\qquad\qquad (n \gtrsim d^{1+\alpha}) \\
&\lesssim d^{-\frac{\alpha}{2}}.
\end{aligned}
$$

This completes the proof of Theorem 3.5.

### D.6 Proof of Theorem D.1 (Training Both Layers)

The proof for training both layers follows the same ideas in the proofs for training the first layer only and the second layer only. In fact, most technical components needed in the proof were already developed in the previous proofs. The only new component is a Jacobian perturbation bound for the case of training both layers, Lemma D.12 (analog of Lemma D.5 for training the first layer).

As before, we proceed in three steps.

#### D.6.1 The NTK at Initialization

**Proposition D.11.** *With high probability over the random initialization $(\boldsymbol{W}(0), \boldsymbol{v}(0))$ and the training data $\boldsymbol{X}$, we have*

$$\left\| \boldsymbol{\Theta}(\boldsymbol{W}(0), \boldsymbol{v}(0)) - \boldsymbol{\Theta}^{\text{lin}} \right\| \lesssim \frac{n}{d^{1+\frac{\alpha}{3}}}.$$

*Proof.* This is a direct corollary of Propositions D.2 and D.8, given that $\boldsymbol{\Theta}(\boldsymbol{W}(0), \boldsymbol{v}(0)) = \boldsymbol{\Theta}_1(\boldsymbol{W}(0), \boldsymbol{v}(0)) + \boldsymbol{\Theta}(\boldsymbol{W}(0))$ ((17)) and $\boldsymbol{\Theta}^{\text{lin}} = \boldsymbol{\Theta}^{\text{lin}1} + \boldsymbol{\Theta}^{\text{lin}2}$ ((20)). □

#### D.6.2 Agreement on Training Data

The proof for the agreement on training data is similar to the case of training the first layer only (Appendix D.3.2). We will again apply Theorem C.2. For this we need a new Jacobian perturbation lemma to replace Lemma D.5, since both layers are allowed to move now.

**Lemma D.12** (Jacobian perturbation for both layers). *If $\phi$ is a smooth activation as in Assumption 3.2, then with high probability over the training data $\boldsymbol{X}$, we have*

$$\left\| \boldsymbol{J}_1(\boldsymbol{W}, \boldsymbol{v}) - \boldsymbol{J}_1(\boldsymbol{W}(0), \boldsymbol{v}(0)) \right\| \lesssim \sqrt{\tfrac{n}{md}} \|\boldsymbol{W} - \boldsymbol{W}(0)\|_F + \sqrt{\tfrac{n}{m}} \|\boldsymbol{v} - \boldsymbol{v}(0)\|, \quad \forall \boldsymbol{W}, \boldsymbol{v}. \quad (42)$$

*If $\phi$ is a piece-wise linear activation as in Assumption 3.2, then with high probability over the random initialization $\boldsymbol{W}(0)$ and the training data $\boldsymbol{X}$, we have*

$$\left\| \boldsymbol{J}_1(\boldsymbol{W}, \boldsymbol{v}) - \boldsymbol{J}_1(\boldsymbol{W}(0), \boldsymbol{v}(0)) \right\| \lesssim \sqrt{\tfrac{n}{d}} \left( \frac{\|\boldsymbol{W} - \boldsymbol{W}(0)\|^{1/3}}{m^{1/6}} + \left( \frac{\log n}{m} \right)^{1/4} \right) + \sqrt{\tfrac{n}{md}} \|\boldsymbol{v} - \boldsymbol{v}(0)\|,$$

$$\forall \boldsymbol{W}, \boldsymbol{v}. \quad (43)$$

*Furthermore, with high probability over the training data $\boldsymbol{X}$, we have*

$$\left\| \boldsymbol{J}_2(\boldsymbol{W}) - \boldsymbol{J}_2(\widetilde{\boldsymbol{W}}) \right\| \lesssim \sqrt{\frac{n}{md}} \left\| \boldsymbol{W} - \widetilde{\boldsymbol{W}} \right\|_F, \qquad \forall \boldsymbol{W}, \widetilde{\boldsymbol{W}}. \quad (44)$$

*Proof.* We will be conditioned on $\boldsymbol{X}$ and on the high-probability events in Claim 3.1.

We first consider the first-layer Jacobian. By the definition of $\boldsymbol{J}_1(\boldsymbol{W}, \boldsymbol{v})$ in (15), we have

$$
\begin{aligned}
&(\boldsymbol{J}_1(\boldsymbol{W}, \boldsymbol{v}) - \boldsymbol{J}_1(\boldsymbol{W}(0), \boldsymbol{v}(0)))(\boldsymbol{J}_1(\boldsymbol{W}, \boldsymbol{v}) - \boldsymbol{J}_1(\boldsymbol{W}(0), \boldsymbol{v}(0)))^\top \\
&= \frac{1}{md}\Bigg( \left( \phi'\left(\boldsymbol{X}\boldsymbol{W}^\top/\sqrt{d}\right) \operatorname{diag}(\boldsymbol{v}) - \phi'\left(\boldsymbol{X}\boldsymbol{W}(0)^\top/\sqrt{d}\right) \operatorname{diag}(\boldsymbol{v}(0)) \right) \\
&\qquad \cdot \left( \phi'\left(\boldsymbol{X}\boldsymbol{W}^\top/\sqrt{d}\right) \operatorname{diag}(\boldsymbol{v}) - \phi'\left(\boldsymbol{X}\boldsymbol{W}(0)^\top/\sqrt{d}\right) \operatorname{diag}(\boldsymbol{v}(0)) \right)^\top \Bigg) \odot (\boldsymbol{X}\boldsymbol{X}^\top).
\end{aligned}
\tag{45}
$$

Then if $\phi$ is a smooth activation, we have with high probability,

$$
\begin{aligned}
&\|\boldsymbol{J}_1(\boldsymbol{W}, \boldsymbol{v}) - \boldsymbol{J}_1(\boldsymbol{W}(0), \boldsymbol{v}(0))\|^2 \\
&\leq \frac{1}{md}\left\| \phi'\left(\boldsymbol{X}\boldsymbol{W}^\top/\sqrt{d}\right) \operatorname{diag}(\boldsymbol{v}) - \phi'\left(\boldsymbol{X}\boldsymbol{W}(0)^\top/\sqrt{d}\right) \operatorname{diag}(\boldsymbol{v}(0)) \right\|^2 \cdot \max_{i \in [n]} \|\boldsymbol{x}_i\|^2 \\
&\hspace{9cm} ((45) \text{ and Lemma B.3}) \\
&\lesssim \frac{1}{m}\left\| \phi'\left(\boldsymbol{X}\boldsymbol{W}^\top/\sqrt{d}\right) \operatorname{diag}(\boldsymbol{v}) - \phi'\left(\boldsymbol{X}\boldsymbol{W}(0)^\top/\sqrt{d}\right) \operatorname{diag}(\boldsymbol{v}(0)) \right\|^2 \quad (\text{Claim 3.1}) \\
&\lesssim \frac{1}{m}\left\| \phi'\left(\boldsymbol{X}\boldsymbol{W}^\top/\sqrt{d}\right) \operatorname{diag}(\boldsymbol{v}(0)) - \phi'\left(\boldsymbol{X}\boldsymbol{W}(0)^\top/\sqrt{d}\right) \operatorname{diag}(\boldsymbol{v}(0)) \right\|^2 \\
&\quad + \frac{1}{m}\left\| \phi'\left(\boldsymbol{X}\boldsymbol{W}^\top/\sqrt{d}\right) \operatorname{diag}(\boldsymbol{v}) - \phi'\left(\boldsymbol{X}\boldsymbol{W}^\top/\sqrt{d}\right) \operatorname{diag}(\boldsymbol{v}(0)) \right\|^2 \\
&\leq \frac{1}{m}\left\| \phi'\left(\boldsymbol{X}\boldsymbol{W}^\top/\sqrt{d}\right) - \phi'\left(\boldsymbol{X}\boldsymbol{W}(0)^\top/\sqrt{d}\right) \right\|_F^2 \cdot \|\operatorname{diag}(\boldsymbol{v}(0))\|^2 \\
&\quad + \frac{1}{m}\left\| \phi'\left(\boldsymbol{X}\boldsymbol{W}^\top/\sqrt{d}\right) \operatorname{diag}(\boldsymbol{v} - \boldsymbol{v}(0)) \right\|_F^2 \\
&\leq \frac{n}{md}\|\boldsymbol{W} - \boldsymbol{W}(0)\|_F^2 + \frac{1}{m}\left\| \phi'\left(\boldsymbol{X}\boldsymbol{W}^\top/\sqrt{d}\right) \operatorname{diag}(\boldsymbol{v} - \boldsymbol{v}(0)) \right\|_F^2 \\
&\hspace{5cm} (\text{using the proof of Lemma D.5, and } \|\operatorname{diag}(\boldsymbol{v}(0))\| = 1) \\
&\leq \frac{n}{md}\|\boldsymbol{W} - \boldsymbol{W}(0)\|_F^2 + \frac{n}{m}\|\boldsymbol{v} - \boldsymbol{v}(0)\|^2. \hspace{3cm} (\phi' \text{ is bounded})
\end{aligned}
$$

This proves (42).

If $\phi$ is a piece-wise linear activation, then with high probability,

$$
\begin{aligned}
&\|\boldsymbol{J}_1(\boldsymbol{W}, \boldsymbol{v}) - \boldsymbol{J}_1(\boldsymbol{W}(0), \boldsymbol{v}(0))\|^2 \\
&\leq \frac{1}{md}\|\boldsymbol{X}\boldsymbol{X}^\top\| \cdot \max_{i \in [n]}\left\| \operatorname{diag}(\boldsymbol{v})\phi'(\boldsymbol{W}\boldsymbol{x}_i/\sqrt{d}) - \operatorname{diag}(\boldsymbol{v}(0))\phi'(\boldsymbol{W}(0)\boldsymbol{x}_i/\sqrt{d}) \right\|^2 \\
&\hspace{9cm} ((45) \text{ and Lemma B.3}) \\
&\leq \frac{n}{md} \cdot \max_{i \in [n]}\left\| \operatorname{diag}(\boldsymbol{v})\phi'(\boldsymbol{W}\boldsymbol{x}_i/\sqrt{d}) - \operatorname{diag}(\boldsymbol{v}(0))\phi'(\boldsymbol{W}(0)\boldsymbol{x}_i/\sqrt{d}) \right\|^2 \quad (\text{Claim 3.1}) \\
&\lesssim \frac{n}{md} \cdot \max_{i \in [n]}\left\| \operatorname{diag}(\boldsymbol{v}(0))\phi'(\boldsymbol{W}\boldsymbol{x}_i/\sqrt{d}) - \operatorname{diag}(\boldsymbol{v}(0))\phi'(\boldsymbol{W}(0)\boldsymbol{x}_i/\sqrt{d}) \right\|^2 \\
&\quad + \frac{n}{md} \cdot \max_{i \in [n]}\left\| \operatorname{diag}(\boldsymbol{v})\phi'(\boldsymbol{W}\boldsymbol{x}_i/\sqrt{d}) - \operatorname{diag}(\boldsymbol{v}(0))\phi'(\boldsymbol{W}\boldsymbol{x}_i/\sqrt{d}) \right\|^2 \\
&\leq \frac{n}{md} \cdot \max_{i \in [n]}\left\| \phi'(\boldsymbol{W}\boldsymbol{x}_i/\sqrt{d}) - \phi'(\boldsymbol{W}(0)\boldsymbol{x}_i/\sqrt{d}) \right\|^2 + \frac{n}{md} \cdot \max_{i \in [n]}\left\| \operatorname{diag}(\boldsymbol{v} - \boldsymbol{v}(0))\phi'(\boldsymbol{W}\boldsymbol{x}_i/\sqrt{d}) \right\|^2 \\
&\hspace{9cm} (\|\operatorname{diag}(\boldsymbol{v}(0))\| = 1) \\
&\lesssim \frac{n}{d}\left( \frac{\|\boldsymbol{W} - \boldsymbol{W}(0)\|^{2/3}}{m^{1/3}} + \sqrt{\frac{\log n}{m}} \right) + \frac{n}{md}\|\boldsymbol{v} - \boldsymbol{v}(0)\|^2. \\
&\hspace{5cm} (\text{using the proof of Lemma D.5, and } \phi' \text{ is bounded})
\end{aligned}
$$

This proves (43).

For the second-layer Jacobian, we have with high probability,

$$
\begin{aligned}
\left\| \boldsymbol{J}_2(\boldsymbol{W}) - \boldsymbol{J}_2(\widetilde{\boldsymbol{W}}) \right\| &= \frac{1}{\sqrt{m}} \left\| \phi(\boldsymbol{X}\boldsymbol{W}^\top/\sqrt{d}) - \phi(\boldsymbol{X}\widetilde{\boldsymbol{W}}^\top/\sqrt{d}) \right\| \\
&\leq \frac{1}{\sqrt{m}} \left\| \boldsymbol{X}(\boldsymbol{W} - \widetilde{\boldsymbol{W}})^\top/\sqrt{d} \right\|_F \qquad (\phi' \text{ is bounded}) \\
&\leq \frac{\|\boldsymbol{X}\|}{\sqrt{md}} \left\| \boldsymbol{W} - \widetilde{\boldsymbol{W}} \right\|_F \\
&\leq \sqrt{\frac{n}{md}} \left\| \boldsymbol{W} - \widetilde{\boldsymbol{W}} \right\|_F,
\end{aligned}
$$

completing the proof of (44). □

Based on Lemma D.12, we can now verify Assumption C.1 for the case of training both layers:

**Lemma D.13.** *Let $R = \sqrt{d \log d}$. With high probability over the random initialization and the training data, for all $(\boldsymbol{W}, \boldsymbol{v})$ and $(\widetilde{\boldsymbol{W}}, \widetilde{\boldsymbol{v}})$ such that $\|\boldsymbol{W} - \boldsymbol{W}(0)\|_F \leq R$, $\left\|\widetilde{\boldsymbol{W}} - \boldsymbol{W}(0)\right\|_F \leq R$, $\|\boldsymbol{v} - \boldsymbol{v}(0)\| \leq R$ and $\|\widetilde{\boldsymbol{v}} - \boldsymbol{v}(0)\| \leq R$, we have*

$$
\left\| \boldsymbol{J}(\boldsymbol{W}, \boldsymbol{v})\boldsymbol{J}(\widetilde{\boldsymbol{W}}, \widetilde{\boldsymbol{v}})^\top - \boldsymbol{\Theta}^{\mathrm{lin}} \right\| \lesssim \frac{n}{d^{1+\frac{\alpha}{3}}}.
$$

*Proof.* This proof is conditioned on all the high-probability events we have shown.

Now consider $(\boldsymbol{W}, \boldsymbol{v})$ and $(\widetilde{\boldsymbol{W}}, \widetilde{\boldsymbol{v}})$ which satisfy the conditions stated in the lemma.

If $\phi$ is a smooth activation, from Lemma D.12 we know

$$
\begin{aligned}
\|\boldsymbol{J}_1(\boldsymbol{W}, \boldsymbol{v}) - \boldsymbol{J}_1(\boldsymbol{W}(0), \boldsymbol{v}(0))\| &\lesssim \sqrt{\frac{n}{md}} \|\boldsymbol{W} - \boldsymbol{W}(0)\|_F + \sqrt{\frac{n}{m}} \|\boldsymbol{v} - \boldsymbol{v}(0)\| \\
&\leq \sqrt{\frac{n}{md}} \cdot \sqrt{d \log d} + \sqrt{\frac{n}{m}} \cdot \sqrt{d \log d} \\
&\lesssim \sqrt{\frac{nd \log d}{m}} \\
&\lesssim \sqrt{\frac{n \log d}{d^{1+\alpha}}} \\
&\ll \sqrt{\frac{n}{d^{1+\frac{2\alpha}{3}}}},
\end{aligned}
$$

where we have used $m \gtrsim d^{2+\alpha}$. If $\phi$ is a piece-wise linear activation, from Lemma D.12 we have

$$
\begin{aligned}
\|\boldsymbol{J}_1(\boldsymbol{W}, \boldsymbol{v}) - \boldsymbol{J}_1(\boldsymbol{W}(0), \boldsymbol{v}(0))\| &\lesssim \sqrt{\frac{n}{d}} \left( \frac{\|\boldsymbol{W} - \boldsymbol{W}(0)\|^{1/3}}{m^{1/6}} + \left( \frac{\log n}{m} \right)^{1/4} \right) + \sqrt{\frac{n}{md}} \|\boldsymbol{v} - \boldsymbol{v}(0)\| \\
&\leq \sqrt{\frac{n}{d}} \left( \frac{(d \log d)^{1/6}}{m^{1/6}} + \left( \frac{\log n}{m} \right)^{1/4} \right) + \sqrt{\frac{n \log d}{m}} \\
&\lesssim \sqrt{\frac{n}{d}} \cdot \frac{(d \log d)^{1/6}}{d^{1/3 + \alpha/6}} + \sqrt{\frac{n \log d}{d^{2+\alpha}}} \\
&\ll \frac{\sqrt{n}}{d^{\frac{2}{3}}}.
\end{aligned}
$$

Hence in either case have $\|\boldsymbol{J}_1(\boldsymbol{W}, \boldsymbol{v}) - \boldsymbol{J}_1(\boldsymbol{W}(0), \boldsymbol{v}(0))\| \leq \frac{\sqrt{n}}{d^{\frac{1}{2}+\frac{\alpha}{3}}}$. Similarly, we have $\left\| \boldsymbol{J}_1(\widetilde{\boldsymbol{W}}, \widetilde{\boldsymbol{v}}) - \boldsymbol{J}_1(\boldsymbol{W}(0), \boldsymbol{v}(0)) \right\| \leq \frac{\sqrt{n}}{d^{\frac{1}{2}+\frac{\alpha}{3}}}$.

Also, we know from Proposition D.2 that $\|\boldsymbol{J}_1(\boldsymbol{W}(0),\boldsymbol{v}(0))\| \lesssim \sqrt{\frac{n}{d}}$. It follows that $\|\boldsymbol{J}_1(\boldsymbol{W},\boldsymbol{v})\| \lesssim \sqrt{\frac{n}{d}}$ and $\left\|\boldsymbol{J}_1(\widetilde{\boldsymbol{W}},\widetilde{\boldsymbol{v}})\right\| \lesssim \sqrt{\frac{n}{d}}$. Then we have

$$\left\|\boldsymbol{J}_1(\boldsymbol{W},\boldsymbol{v})\boldsymbol{J}_1(\widetilde{\boldsymbol{W}},\widetilde{\boldsymbol{v}})^\top - \boldsymbol{J}_1(\boldsymbol{W}(0),\boldsymbol{v}(0))\boldsymbol{J}_1(\boldsymbol{W}(0),\boldsymbol{v}(0))^\top\right\|$$

$$\leq \|\boldsymbol{J}_1(\boldsymbol{W},\boldsymbol{v})\| \cdot \left\|\boldsymbol{J}_1(\widetilde{\boldsymbol{W}},\widetilde{\boldsymbol{v}}) - \boldsymbol{J}_1(\boldsymbol{W}(0),\boldsymbol{v}(0))\right\| + \|\boldsymbol{J}_1(\boldsymbol{W}(0),\boldsymbol{v}(0))\| \cdot \|\boldsymbol{J}_1(\boldsymbol{W},\boldsymbol{v}) - \boldsymbol{J}_1(\boldsymbol{W}(0),\boldsymbol{v}(0))\|$$

$$\lesssim \sqrt{\frac{n}{d}} \cdot \frac{\sqrt{n}}{d^{\frac{1}{2}+\frac{\alpha}{3}}} + \sqrt{\frac{n}{d}} \cdot \frac{\sqrt{n}}{d^{\frac{1}{2}+\frac{\alpha}{3}}}$$

$$\lesssim \frac{n}{d^{1+\frac{\alpha}{3}}}.$$

Next we look at the second-layer Jacobian. From Lemma D.12 we know $\|\boldsymbol{J}_2(\boldsymbol{W}) - \boldsymbol{J}_2(\boldsymbol{W}(0))\| \lesssim \sqrt{\frac{n}{md}} \cdot \sqrt{d \log d} \lesssim \sqrt{\frac{n \log d}{d^{2+\alpha}}} \ll \frac{\sqrt{n}}{d^{1+\frac{\alpha}{3}}}$. Similarly we have $\left\|\boldsymbol{J}_2(\widetilde{\boldsymbol{W}}) - \boldsymbol{J}_2(\boldsymbol{W}(0))\right\| \ll \frac{\sqrt{n}}{d^{1+\frac{\alpha}{3}}}$. Also, from Proposition D.8 we know $\|\boldsymbol{J}_2(\boldsymbol{W}(0))\| \lesssim \sqrt{n}$, which implies $\|\boldsymbol{J}_2(\boldsymbol{W})\| \lesssim \sqrt{n}$ and $\left\|\boldsymbol{J}_2(\widetilde{\boldsymbol{W}})\right\| \lesssim \sqrt{n}$. It follows that

$$\left\|\boldsymbol{J}_2(\boldsymbol{W})\boldsymbol{J}_2(\widetilde{\boldsymbol{W}})^\top - \boldsymbol{J}_2(\boldsymbol{W}(0))\boldsymbol{J}_2(\boldsymbol{W}(0))^\top\right\|$$

$$\leq \|\boldsymbol{J}_2(\boldsymbol{W})\| \cdot \left\|\boldsymbol{J}_2(\widetilde{\boldsymbol{W}}) - \boldsymbol{J}_2(\boldsymbol{W}(0))\right\| + \|\boldsymbol{J}_2(\boldsymbol{W}(0))\| \cdot \|\boldsymbol{J}_2(\boldsymbol{W}) - \boldsymbol{J}_2(\boldsymbol{W}(0))\|$$

$$\lesssim \sqrt{n} \cdot \frac{\sqrt{n}}{d^{1+\frac{\alpha}{3}}} + \sqrt{n} \cdot \frac{\sqrt{n}}{d^{1+\frac{\alpha}{3}}}$$

$$\lesssim \frac{n}{d^{1+\frac{\alpha}{3}}}.$$

Combining the above auguments for two layers, we obtain

$$\left\|\boldsymbol{J}(\boldsymbol{W},\boldsymbol{v})\boldsymbol{J}(\widetilde{\boldsymbol{W}},\widetilde{\boldsymbol{v}})^\top - \boldsymbol{J}(\boldsymbol{W}(0),\boldsymbol{v}(0))\boldsymbol{J}(\boldsymbol{W}(0),\boldsymbol{v}(0))^\top\right\|$$

$$= \left\|\boldsymbol{J}_1(\boldsymbol{W},\boldsymbol{v})\boldsymbol{J}_1(\widetilde{\boldsymbol{W}},\widetilde{\boldsymbol{v}})^\top + \boldsymbol{J}_2(\boldsymbol{W})\boldsymbol{J}_2(\widetilde{\boldsymbol{W}})^\top \right.$$

$$\left. - \boldsymbol{J}_1(\boldsymbol{W}(0),\boldsymbol{v}(0))\boldsymbol{J}_1(\boldsymbol{W}(0),\boldsymbol{v}(0))^\top - \boldsymbol{J}_2(\boldsymbol{W}(0))\boldsymbol{J}_2(\boldsymbol{W}(0))^\top\right\|$$

$$\leq \left\|\boldsymbol{J}_1(\boldsymbol{W},\boldsymbol{v})\boldsymbol{J}_1(\widetilde{\boldsymbol{W}},\widetilde{\boldsymbol{v}})^\top - \boldsymbol{J}_1(\boldsymbol{W}(0),\boldsymbol{v}(0))\boldsymbol{J}_1(\boldsymbol{W}(0),\boldsymbol{v}(0))^\top\right\|$$

$$+ \left\|\boldsymbol{J}_2(\boldsymbol{W})\boldsymbol{J}_2(\widetilde{\boldsymbol{W}})^\top - \boldsymbol{J}_2(\boldsymbol{W}(0))\boldsymbol{J}_2(\boldsymbol{W}(0))^\top\right\|$$

$$\lesssim \frac{n}{d^{1+\frac{\alpha}{3}}} + \frac{n}{d^{1+\frac{\alpha}{3}}}$$

$$\lesssim \frac{n}{d^{1+\frac{\alpha}{3}}}.$$

Combining the above inequality with Proposition D.11, the proof is finished. $\qquad\square$

Finally, we can apply Theorem C.2 with $R = \sqrt{d \log d}$ and $\epsilon = O(\frac{n}{d^{1+\frac{\alpha}{3}}})$, and obtain that for all $t \leq T$:

$$\sqrt{\sum_{i=1}^n (f_t(\boldsymbol{x}_i) - f^{\mathrm{lin}}(\boldsymbol{x}_i))^2} \lesssim \frac{\eta t \epsilon}{\sqrt{n}} \lesssim \frac{d \log d \cdot \frac{n}{d^{1+\frac{\alpha}{3}}}}{\sqrt{n}} = \frac{\sqrt{n} \log d}{d^{\frac{\alpha}{3}}} \ll \frac{\sqrt{n}}{d^{\frac{\alpha}{4}}},$$

i.e.,

$$\frac{1}{n}\sum_{i=1}^n (f_t(\boldsymbol{x}_i) - f_t^{\mathrm{lin}}(\boldsymbol{x}_i))^2 \leq d^{-\frac{\alpha}{2}}.$$

This proves the first part in Theorem D.1.

Note that Theorem C.2 also tells us $\|\boldsymbol{W}(t) - \boldsymbol{W}(0)\| \leq \sqrt{d \log d}$, $\|\boldsymbol{v}(t) - \boldsymbol{v}(0)\| \leq \sqrt{d \log d}$ and $\|\boldsymbol{\delta}(t)\| \leq \sqrt{d \log d}$, which will be useful for proving the guarantee on the distribution $\mathcal{D}$.

### D.6.3 Agreement on Distribution

The proof for the second part of Theorem D.1 is basically identical to the case of training the first layer (Appendix D.3.3), so we will only sketch the differences here to avoid repetition.

Recall that in Appendix D.3.3 we define an auxiliary model which is the first-order approximation of the network around initialization. Here since we are training both layers, we need to modify the definition of the auxiliary model to incorporate deviation from initialization in both layers:

$$f^{\text{aux}}(\boldsymbol{x}; \boldsymbol{W}, \boldsymbol{v}) := \langle \boldsymbol{W} - \boldsymbol{W}(0), \nabla_{\boldsymbol{W}} f(\boldsymbol{x}; \boldsymbol{W}(0), \boldsymbol{v}(0)) \rangle + \langle \boldsymbol{v} - \boldsymbol{v}(0), \nabla_{\boldsymbol{v}} f(\boldsymbol{x}; \boldsymbol{W}(0), \boldsymbol{v}(0)) \rangle.$$

Then we denote $f_t^{\text{aux}}(\boldsymbol{x}) := f^{\text{aux}}(\boldsymbol{x}; \boldsymbol{W}(t), \boldsymbol{v}(t))$.

There are two more minor changes to Appendix D.3.3:

1. When proving $f_t$ and $f_t^{\text{aux}}$ are close on both training data and imaginary test data, we need to bound a Jacobian perturbation. In Appendix D.3.3 this step is done using Lemma D.5. Now we simply need to use Lemma D.12 instead and note that $\|\boldsymbol{W}(t) - \boldsymbol{W}(0)\| \leq \sqrt{d \log d}$ and $\|\boldsymbol{v}(t) - \boldsymbol{v}(0)\| \leq \sqrt{d \log d}$.

2. Instead of (37), the empirical Rademacher complexity of the function class that each $f_t^{\text{aux}} - f_t^{\text{lin}}$ lies in will be

$$\frac{\sqrt{d \log d}}{n} \sqrt{\text{Tr}[\boldsymbol{\Theta}(\boldsymbol{W}(0), \boldsymbol{v}(0))] + \text{Tr}[\boldsymbol{\Theta}^{\text{lin}}]}$$
$$= \frac{\sqrt{d \log d}}{n} \sqrt{\text{Tr}[\boldsymbol{\Theta}_1(\boldsymbol{W}(0), \boldsymbol{v}(0))] + \text{Tr}[\boldsymbol{\Theta}_2(\boldsymbol{W}(0))] + \text{Tr}[\boldsymbol{\Theta}^{\text{lin1}}] + \text{Tr}[\boldsymbol{\Theta}^{\text{lin2}}]}.$$

   In Appendices D.3.3 and D.5.3, we have shown that the above 4 traces are all $O(n)$ with high probability. Hence we get the same Rademacher complexity bound as before.

Modulo these differences, the proof proceeds the same as Appendix D.3.3. Therefore we conclude the proof of Theorem D.1.

### D.7 Proof of Claim 3.1

*Proof of Claim 3.1.* According to Assumption 3.1, we have $\boldsymbol{x}_i = \boldsymbol{\Sigma}^{1/2} \bar{\boldsymbol{x}}_i$ where $\mathbb{E}[\bar{\boldsymbol{x}}_i] = \boldsymbol{0}$, $\mathbb{E}[\bar{\boldsymbol{x}}_i \bar{\boldsymbol{x}}_i^\top] = \boldsymbol{I}$, and $\bar{\boldsymbol{x}}_i$'s entries are independent and $O(1)$-subgaussian.

By Hanson-Wright inequality (specifically, Theorem 2.1 in Rudelson and Vershynin [2013]), we have for any $t \geq 0$,

$$\Pr\left[ \left| \left\| \boldsymbol{\Sigma}^{1/2} \bar{\boldsymbol{x}}_i \right\| - \|\boldsymbol{\Sigma}^{1/2}\|_F \right| > t \right] \leq 2 \exp\left( -\Omega\left( \frac{t^2}{\left\| \boldsymbol{\Sigma}^{1/2} \right\|^2} \right) \right),$$

i.e.,

$$\Pr\left[ \left| \|\boldsymbol{x}_i\| - \sqrt{d} \right| > t \right] \leq 2 \exp\left( -\Omega\left( t^2 \right) \right).$$

Let $t = C\sqrt{\log n}$ for a sufficiently large constant $C > 0$. Taking a union bound over all $i \in [n]$, we obtain that with high probability, $\|\boldsymbol{x}_i\| = \sqrt{d} \pm O(\sqrt{\log n})$ for all $i \in [n]$ simultaneously. This proves the first property in Claim 3.1.

For $i \neq j$, we have $\langle \boldsymbol{x}_i, \boldsymbol{x}_j \rangle = \bar{\boldsymbol{x}}_i^\top \boldsymbol{\Sigma} \bar{\boldsymbol{x}}_j$. Conditioned on $\bar{\boldsymbol{x}}_j$, we know that $\bar{\boldsymbol{x}}_i^\top \boldsymbol{\Sigma} \bar{\boldsymbol{x}}_j$ is zero-mean and $O(\|\boldsymbol{\Sigma} \bar{\boldsymbol{x}}_j\|^2)$-subgaussian, which means for any $t \geq 0$,

$$\Pr\left[ \left| \bar{\boldsymbol{x}}_i^\top \boldsymbol{\Sigma} \bar{\boldsymbol{x}}_j \right| > t \,\Big|\, \bar{\boldsymbol{x}}_j \right] \leq 2 \exp\left( -\frac{t^2}{\|\boldsymbol{\Sigma} \bar{\boldsymbol{x}}_j\|^2} \right).$$

Since we have shown that $\|\boldsymbol{\Sigma} \bar{\boldsymbol{x}}_j\|^2 \lesssim \|\boldsymbol{x}_j\|^2 \lesssim \sqrt{d} + \sqrt{\log n} \lesssim \sqrt{d}$ with probability at least $1 - n^{-10}$, we have

$$\Pr\left[ \left| \bar{\boldsymbol{x}}_i^\top \boldsymbol{\Sigma} \bar{\boldsymbol{x}}_j \right| > t \right] \leq n^{-10} + 2 \exp\left( -\Omega\left( \frac{t^2}{d} \right) \right).$$

Then we can take $t = C\sqrt{d\log n}$ and apply a union bound over $i, j$, which gives $|\langle \boldsymbol{x}_i, \boldsymbol{x}_j \rangle| \lesssim \sqrt{d\log n}$ for all $i \neq j$ with high probability. This completes the proof of the second statement in Claim 3.1.

Finally, for $\boldsymbol{X}\boldsymbol{X}^\top$, we can use standard covariance concentration (see, e.g., Lemma A.6 in Du et al. [2020]) to obtain $0.9\boldsymbol{\Sigma} \preceq \frac{1}{n}\boldsymbol{X}^\top\boldsymbol{X} \preceq 1.1\boldsymbol{\Sigma}$ with high probability. This implies $\|\boldsymbol{X}\boldsymbol{X}^\top\| = \|\boldsymbol{X}^\top\boldsymbol{X}\| = \Theta(n)$. $\qquad\square$

# E Omitted Details in Section 4

*Proof of Proposition 4.1.* For an input $\boldsymbol{x} \in \mathbb{R}^d$ and an index $k \in [d]$, we let $[\boldsymbol{x}]_{k:k+q}$ be the patch of size $q$ starting from index $k$, i.e., $[\boldsymbol{x}]_{k:k+q} := \left[[\boldsymbol{x}]_k, [\boldsymbol{x}]_{k+1}, \ldots, [\boldsymbol{x}]_{k+q-1}\right]^\top \in \mathbb{R}^q$.

For two datapoints $\boldsymbol{x}_i$ and $\boldsymbol{x}_j$ $(i, j \in [n])$ and a location $k \in [d]$, we define

$$\rho_{i,j,k} := \frac{\left\langle [\boldsymbol{x}_i]_{k:k+q}, [\boldsymbol{x}_j]_{k:k+q} \right\rangle}{q}$$

which is a local correlation between $\boldsymbol{x}_i$ and $\boldsymbol{x}_j$.

Now we calculate the infinite-width NTK matrix $\boldsymbol{\Theta}_{\mathsf{CNN}}$, which is also the expectation of a finite-width NTK matrix with respect to the randomly initialized weights $(\boldsymbol{W}, \boldsymbol{V})$. We divide the NTK matrix into two components corresponding to two layers: $\boldsymbol{\Theta}_{\mathsf{CNN}} = \boldsymbol{\Theta}_{\mathsf{CNN}}^{(1)} + \boldsymbol{\Theta}_{\mathsf{CNN}}^{(2)}$, and consider the two layers separately.

**Step 1: the second-layer NTK.** Since the CNN model (12) is linear in the second layer weights, it is easy to derive the formula for the second-layer NTK:

$$
\begin{aligned}
\left[\boldsymbol{\Theta}_{\mathsf{CNN}}^{(2)}\right]_{i,j} &= \frac{1}{d}\mathbb{E}_{\boldsymbol{w}\sim\mathcal{N}(\boldsymbol{0}_q, \boldsymbol{I}_q)}\left[\phi(\boldsymbol{w}*\boldsymbol{x}_i/\sqrt{q})^\top\phi(\boldsymbol{w}*\boldsymbol{x}_j/\sqrt{q})\right] \\
&= \frac{1}{d}\mathbb{E}_{\boldsymbol{w}\sim\mathcal{N}(\boldsymbol{0}_q, \boldsymbol{I}_q)}\left[\sum_{k=1}^d \phi([\boldsymbol{w}*\boldsymbol{x}_i]_k/\sqrt{q})\phi([\boldsymbol{w}*\boldsymbol{x}_j]_k/\sqrt{q})\right] \\
&= \frac{1}{d}\mathbb{E}_{\boldsymbol{w}\sim\mathcal{N}(\boldsymbol{0}_q, \boldsymbol{I}_q)}\left[\sum_{k=1}^d \phi\left(\left\langle \boldsymbol{w}, [\boldsymbol{x}_i]_{k:k+q}\right\rangle/\sqrt{q}\right)\phi\left(\left\langle \boldsymbol{w}, [\boldsymbol{x}_j]_{k:k+q}\right\rangle/\sqrt{q}\right)\right] \\
&= \frac{1}{d}\sum_{k=1}^d P(\rho_{i,j,k}),
\end{aligned}
$$

where

$$P(\rho) := \mathbb{E}_{(z_1, z_2)\sim\mathcal{N}(\boldsymbol{0}, \boldsymbol{\Lambda})}[\phi(z_1)\phi(z_2)], \text{ where } \boldsymbol{\Lambda} = \begin{pmatrix} 1 & \rho \\ \rho & 1 \end{pmatrix}, \quad |\rho| \leq 1.$$

Note that we have used the property $\left\|[\boldsymbol{x}_j]_{k:k+q}\right\| = \left\|[\boldsymbol{x}_j]_{k:k+q}\right\| = \sqrt{q}$ since the data are from the hypercube $\{\pm 1\}^d$.

For $i \neq j$, we can do a Taylor expansion of $P$ around 0: $P(\rho) = \zeta^2\rho \pm O(|\rho|^3)$. Here since $\phi = \mathrm{erf}$ is an odd function, all the even-order terms in the expansion vanish. Therefore we have

$$\left[\boldsymbol{\Theta}_{\mathsf{CNN}}^{(2)}\right]_{i,j} = \frac{1}{d}\sum_{k=1}^d(\zeta^2\rho_{i,j,k} \pm O(|\rho_{i,j,k}|^3)) = \frac{1}{d}\zeta^2\boldsymbol{x}_i^\top\boldsymbol{x}_j \pm \frac{1}{d}\sum_{k=1}^d O(|\rho_{i,j,k}|^3).$$

Next we bound the error term $\frac{1}{d}\sum_{k=1}^d |\rho_{i,j,k}|^3$ for all $i \neq j$. For each $i, j, k$ $(i \neq j)$, since $\boldsymbol{x}_i, \boldsymbol{x}_j \overset{\text{i.i.d.}}{\sim} \mathsf{Unif}(\{\pm 1\}^d)$, by Hoeffding's inequality we know that with probability $1 - \delta$, we have $|\rho_{i,j,k}| \lesssim \sqrt{\frac{\log\frac{1}{\delta}}{q}}$. Taking a union bound, we know that with high probability, for all $i, j, k$ $(i \neq j)$

we have $|\rho_{i,j,k}| = \tilde{O}(q^{-1/2})$. Now we will be conditioned on this happening. Then we write

$$\sum_{k=1}^{d} |\rho_{i,j,k}|^3 = \sum_{k=1}^{q} |\rho_{i,j,k}|^3 + \sum_{k=q+1}^{2q} |\rho_{i,j,k}|^3 + \cdots,$$

i.e., we divide the sum into $\lceil d/q \rceil$ groups each containing no more than $q$ terms. By the definition of $\rho_{i,j,k}$, it is easy to see that the groups are independent. Also, we have shown that the sum in each group is at most $q \cdot \tilde{O}(q^{-3/2}) = \tilde{O}(q^{-1/2})$. Therefore, using another Hoeffding's inequality among the groups, and applying a union bound over all $i, j$, we know that with high probability for all $i, j$ $(i \neq j)$,

$$\frac{1}{d} \sum_{k=1}^{d} |\rho_{i,j,k}|^3 \leq \frac{1}{d} \tilde{O}(q^{-1/2}) \cdot \tilde{O}(\sqrt{d/q}) = \tilde{O}\left(\frac{1}{q\sqrt{d}}\right).$$

Therefore we have shown that with high probability, for all $i \neq j$,

$$\left| \left[ \boldsymbol{\Theta}_{\mathsf{CNN}}^{(2)} - \zeta^2 \boldsymbol{X} \boldsymbol{X}^{\top}/d \right]_{i,j} \right| = \tilde{O}\left(\frac{1}{q\sqrt{d}}\right).$$

This implies

$$\left\| \left( \boldsymbol{\Theta}_{\mathsf{CNN}}^{(2)} - \zeta^2 \boldsymbol{X} \boldsymbol{X}^{\top}/d \right)_{\mathrm{off}} \right\| \leq \left\| \left( \boldsymbol{\Theta}_{\mathsf{CNN}}^{(2)} - \zeta^2 \boldsymbol{X} \boldsymbol{X}^{\top}/d \right)_{\mathrm{off}} \right\|_F = \tilde{O}\left(\frac{n}{q\sqrt{d}}\right)$$

$$= \tilde{O}\left(\frac{n}{d^{\frac{1}{2}+2\alpha}\sqrt{d}}\right) = O\left(\frac{n}{d^{1+\alpha}}\right).$$

For the diagonal entries, we can easily see

$$\left\| \left( \boldsymbol{\Theta}_{\mathsf{CNN}}^{(2)} - \zeta^2 \boldsymbol{X} \boldsymbol{X}^{\top}/d \right)_{\mathrm{diag}} \right\| = O(1) = O\left(\frac{n}{d^{1+\alpha}}\right).$$

Combining the above two equations, we obtain

$$\left\| \boldsymbol{\Theta}_{\mathsf{CNN}}^{(2)} - \zeta^2 \boldsymbol{X} \boldsymbol{X}^{\top}/d \right\| = O\left(\frac{n}{d^{1+\alpha}}\right).$$

**Step 2: The first-layer NTK.** We calculate the derivative of the output of the CNN with respect to the first-layer weights as:

$$\nabla_{\boldsymbol{w}_r} f_{\mathsf{CNN}}(\boldsymbol{x}; \boldsymbol{W}, \boldsymbol{V}) = \frac{1}{\sqrt{md}} \sum_{k=1}^{d} [\boldsymbol{v}_r]_k \, \phi'\left(\left\langle \boldsymbol{w}_r, [\boldsymbol{x}]_{k:k+q} \right\rangle / \sqrt{q}\right) [\boldsymbol{x}]_{k:k+q} / \sqrt{q}.$$

Therefore, the entries in the first-layer NTK matrix are

$$\left[ \boldsymbol{\Theta}_{\mathsf{CNN}}^{(2)} \right]_{i,j} = \mathbb{E}_{\boldsymbol{W}, \boldsymbol{V}} \left[ \sum_{r=1}^{m} \langle \nabla_{\boldsymbol{w}_r} f_{\mathsf{CNN}}(\boldsymbol{x}_i; \boldsymbol{W}, \boldsymbol{V}), \nabla_{\boldsymbol{w}_r} f_{\mathsf{CNN}}(\boldsymbol{x}_j; \boldsymbol{W}, \boldsymbol{V}) \rangle \right]$$

$$= \mathbb{E}_{\boldsymbol{W}} \left[ \frac{1}{md} \sum_{r=1}^{m} \sum_{k=1}^{d} \phi'\left(\left\langle \boldsymbol{w}_r, [\boldsymbol{x}_i]_{k:k+q} \right\rangle / \sqrt{q}\right) \phi'\left(\left\langle \boldsymbol{w}_r, [\boldsymbol{x}_j]_{k:k+q} \right\rangle / \sqrt{q}\right) \rho_{i,j,k} \right]$$

$$= \mathbb{E}_{\boldsymbol{w} \sim \mathcal{N}(\boldsymbol{0}_q, \boldsymbol{I}_q)} \left[ \frac{1}{d} \sum_{k=1}^{d} \phi'\left(\left\langle \boldsymbol{w}, [\boldsymbol{x}_i]_{k:k+q} \right\rangle / \sqrt{q}\right) \phi'\left(\left\langle \boldsymbol{w}, [\boldsymbol{x}_j]_{k:k+q} \right\rangle / \sqrt{q}\right) \rho_{i,j,k} \right]$$

$$= \frac{1}{d} \sum_{k=1}^{d} Q(\rho_{i,j,k}) \cdot \rho_{i,j,k},$$

where

$$Q(\rho) := \mathbb{E}_{(z_1, z_2) \sim \mathcal{N}(\boldsymbol{0}, \boldsymbol{\Lambda})}[\phi'(z_1)\phi'(z_2)], \text{ where } \boldsymbol{\Lambda} = \begin{pmatrix} 1 & \rho \\ \rho & 1 \end{pmatrix}, \quad |\rho| \leq 1.$$

For $i \neq j$, we can do a Taylor expansion of $Q$ around $0$: $Q(\rho) = \zeta^2 \pm O(\rho^2)$. Here since $\phi' = \mathrm{erf}'$ is an even function, all the odd-order terms in the expansion vanish. Therefore we have

$$\left[\boldsymbol{\Theta}_{\mathsf{CNN}}^{(1)}\right]_{i,j} = \frac{1}{d}\sum_{k=1}^{d}(\zeta^2 \pm O(\rho_{i,j,k}^2))\rho_{i,j,k} = \frac{1}{d}\zeta^2 \boldsymbol{x}_i^\top \boldsymbol{x}_j \pm \frac{1}{d}\sum_{k=1}^{d} O(|\rho_{i,j,k}|^3).$$

Then, using the exact same analysis for the second-layer NTK, we know that with high probability,

$$\left\|\boldsymbol{\Theta}_{\mathsf{CNN}}^{(1)} - \zeta^2 \boldsymbol{X}\boldsymbol{X}^\top/d\right\| = O\left(\frac{n}{d^{1+\alpha}}\right).$$

Finally, combining the results for two layers, we conclude the proof of Proposition 4.1. $\qquad\square$