[Reviews · NeurIPS 2020]

Review 1

Summary and Contributions: This paper theoretically proves that gradient descent on a neural network can learn a NTK matrix which is close to a linear model in the early phase. First, several theoretical results are provided to prove that their proposal is valid on two-layer fully-connected neural networks. Then some analyses are given to show whether their proposal still holds on more complicated networks. This paper reveals that in the early learning of neural networks, it can be treated as a linear model, rather than as a black box that can be difficult to understand.

Strengths: This paper obtains their theoretical results by bounding the spectral norm of the difference between the Neural Tangent Kernel (NTK) and the kernel for a linear model. The proofs can withstand scrutiny. This paper proposes a direction to analyze the neural networks in early-time learning, which is relevant with the topics of NeurlPS.

Weaknesses: The theoretical results are not complete. Only the theoretical results and proofs of the two-layer fully-connected neural network are given, and the proof of extending the theoretical results to the deep neural network needs to be supplemented. The experimental results are not sufficient. This work only conducts experiments on single data set, and the experimental results could not fully verify the theoretical results.

Correctness: The theorems that have been proved in this paper are correct. However, the reasoning of deep neural networks is to be discussed. The theoretical results cannot be easily extended from two-layer fully-connected neural networks to deep neural networks. The analysis of the extended part in this paper needs further discussion.

Clarity: The logical relationship of the related work is not smooth enough. The writing of the proof by step in the theoretical proof section is clear. But the part of the theoretical results is not written well, since the experimental results appear among them.

Relation to Prior Work: The description of the related work is not sufficient, mainly because it does not explain the contribution of the work to the research of neural networks, but only shows the improvement compared with the related work.

Reproducibility: Yes

Additional Feedback: ------------------After rebuttal----------- Thank you for your response. I have carefully read the authors’ feedback and other reviews. Most of my initial concerns are erased. Thus I decide to improve my score.


Review 2

Summary and Contributions: This paper establishes a connection between the early-time learning dynamics of neural networks and the dynamics of a simple linear model on the input. The main contributions of this paper are as follows: -- It proves that the early-time dynamics of training neural networks and training a simple linear model of the input are coupled (close in function space). This is shown for training either or both layer in a wide two-layer neural network. -- When data is well conditioned, the paper shows in addition that the early-time interval is almost long enough for the dynamics to learn the *best* linear model that fits the training dataset. -- Experimental results verifying the agreement between neural network and the linear model in the early stage of training.

Strengths: -- The theoretical results presented in this paper are novel, and quite interesting to me in the following aspects: (1) The prior work of Nakkiran et al. 2019 showed empirically that the early time NN predictions are well explained by a linear model of the input, in an information theoretic sense. This paper establishes this connection theoretically. (2) The coupling result utilizes NTK theory, but unlike the NTK does not require the width to be a big polynomial of n. Indeed, the only requirement is m, n >= d^{1+\alpha}, and so the width can actually be smaller than n. (3) It is interesting to see training each of the two layers yields different linear models. In particular training W is coupled with an exact linear model of [x, 1] whereas training v is coupled with a linear model with additional features depending on ||x||^2. -- The proof builds on random matrix analyses of the NTK which could be of technical interest. -- The experiments are nice to see and support the theories quite well. In addition, though the paper is about the agreement between NN and the linear model, the experiments also demonstrated how NN deviates from the linear model after a certain time period. For example, in Figure 3(a) we see that the non-linear part of the error starts to decrease for both fully-connected and convolutional networks, and it seemed the convolutional network was reducing this error faster than the fully-connected network. These are complementary to the experiments in Nakkiran et al. and I can imagine more large-scale experiments of this type could be interesting.

Weaknesses: -- One minor concern I have about the theoretical result is the scaling regime, specifically why \alpha <= 1/4 is needed. What happens if \alpha > 1/4? (This may rather be a research question for the authors). Do you expect the NN to be coupled with a quadratic / higher-order kernel and thus learning a polynomial of the input? -- The extensions to the multi-layer and convolutional cases are only sketched out and not rigorously established.

Correctness: The proofs and empirical methodologies are correct upon my inspections.

Clarity: This paper is very well presented. I find it a quite enjoyable read.

Relation to Prior Work: The paper has sufficient discussions about prior work and the relationship between prior work and the present paper.

Reproducibility: Yes

Additional Feedback:


Review 3

Summary and Contributions: This paper proves that, if the input features are sampled from a well-behaved distribution, and if gradient descent is used for training, then on both the training set and test set, the output of a two-layer network is close to the output of a linear model (potentially with some norm-dependent features, as discussed in Section 3.4). Moreover, it is discussed how to extend the analysis to deep networks and convolutional networks, and some empirical verification is provided.

Strengths: I think the relationship between two-layer networks and linear models proved in this paper is interesting. Due to the NTK analysis, we already know that using gradient descent, the output of a wide network is close to the linear model with NTK features. However, this paper shows that for a well-behaved input distribution, the output of the network is close to the linear model with basically the original input features plus some norm-dependent features in the early phase of training.

Weaknesses: The main limitation in my opinion is the assumption on the input distribution, Assumption 3.1. Is it approximately satisfied by real-world data? Is there any way to transform a given dataset so that Assumption 3.1 holds, and does it improve the performance of the network?

Correctness: I do not see any correctness issue.

Clarity: Overall the paper is well-written. The empirical results described in Section 4.2 are interesting, and I have the following questions: 1. What exactly is the linear model? Did you calculate and include the constants xi, nu and the norm-dependent features? 2. How long does the early phase last if we use a large learning rate as in practice?

Relation to Prior Work: The discussion of prior work is thorough as far as I know.

Reproducibility: Yes

Additional Feedback: Reply to the feedback: Thanks for the response! I would like to increase my score, because I realize I missed an interesting point proved in this paper: as shown in Corollary 3.3, there exists a data set on which gradient descent first learns the optimal linear classifier, and then something more complicated (for example, the optimal solution in the NTK regime, when the width is large enough). Although people have had this intuition for a long time, I think it is nice to provide a concrete case and a rigorous proof. On the other hand, I think it would be nice to include more details of the experiments. I asked how long the early phase lasts with a large learning rate as in practice, but it seems that the authors did not answer this question. In Figure 3, did you train the networks until convergence? I feel that the convolutional network should be able to drive the loss lower than what is shown in Figure 3. I think it would be nice to see how long the early phase lasts in a typical training. Relatedly, in Appendix A it is mentioned that the learning rate is 0.01/||NTK||; how large is ||NTK|| exactly?


Review 4

Summary and Contributions: This paper studies the learning dynamics of a two-layer neural network and shows that in its early training phase, the network mimics behavior of a simple linear model on input data, which is assumed to be somewhat similar to Gaussian inputs. The analysis is based on NTK but only requires a modest number of hidden neurons. The theory is supported with experiments on synthetic and real data.

Strengths: To my knowledge, this work presents a new theoretical and empirical insight on the early-time training dynamics of neural networks. The analysis is intuitive and insightful. The requirement on width is reasonable, mainly because of the well-behaved data. Therefore, I recommend it for acceptance.

Weaknesses: There is a restrictive assumption on the data distribution, which behaves very much like Gaussian data. However, the experiment on CIFA10 does show a similar phenomenon.

Correctness: The theoretical results are sound and empirical studies are convincing. However, the claim “we show that these common perceptions can be completely false in the early phase of learning” is not accurate, for the reason that the results hold for a limited class of data distributions and two-layer networks. The phrase “early phase of learning” is not entirely clear.

Clarity: Yes

Relation to Prior Work: Yes

Reproducibility: Yes

Additional Feedback: I have some general questions: 1. Can the authors explain further the statement “we show that these common perceptions can be completely false”? 2. Any intuition/reason why the networks eventually escape the linear behavior? 3. Would one expect the same behavior for deep networks? Could the analysis be extended to deep network settings and more general data? 4. In Section 3.4, what if RELU was used instead of ERF? ---------- After rebuttal ---------- I am happy with the author response and keep the same score.

[Author Response · NeurIPS 2020]

We thank all reviewers for their valuable feedback. Below we address each reviewer's questions and concerns.

—— **Response to Reviewer 1** ——

[The theoretical results are "not complete" because only two-layer fully-connected networks are considered.] First of all, we'd like to remind the reviewer that many interesting deep learning theory papers focus on two-layer networks. There are *dozens or even hundreds* of such papers published in top conferences and journals. Just for example, [15,16,17,20,25,34,35,36,43] in our paper's references are all theoretical studies of two-layer networks. Second, it's already challenging to formally establish the two-layer results in our paper; we believe that the conceptual and technical contributions of the paper are interesting and insightful enough to meet the standard of publication, as acknowledged by all other reviewers. Third, we didn't claim we have rigorous results for multi-layer nets, but only provided empirical results and partial theory to support them, and we explicitly mentioned formally proving them as a future work direction in Section 5. For these reasons, we find it incorrect and unfair to call our theoretical results "not complete" and to recommend weak reject mainly based on this. We hope the reviewer can reconsider their decision.

["The experimental results are not sufficient. This work only conducts experiments on single data set, and the experimental results could not fully verify the theoretical results."] We do not understand this comment. We provided experimental results on synthetic data, CIFAR-10, as well as MNIST in the supplementary. These are more than a "single data set." Also, our experimental results match theoretical predictions very well, so we do not understand why they "could not fully verify the theoretical results." Note that all other reviewers find our experiments convincing.

[Related work not sufficient.] We do not understand what's the reviewer's specific concern about related work. We have tried to provide a thorough discussion of related work, and all other reviewers think our discussion is sufficient. If the reviewer can point out exactly what they think is missing in the discussion, we are happy to incorporate it in the paper.

["The part of the theoretical results is not written well, since the experimental results appear among them."] Thank you for the feedback about writing. In fact, we separated the experiments in two sections *on purpose*: Section 3.4 is to verify the two-layer results in Section 3, and Section 4.2 is for multi-layer and convolutional nets. We thought this is the clearest way to present our results, and all other reviewers think our paper is well-written (in particular, R2 finds it "a quite enjoyable read"). Also, we did separate theory and experiments in different subsections so that they are not really muddled together. Regarding the reviewer's comment that "theoretical results are not written well", we do not understand the reasoning that our arrangement of experiments affects whether the theoretical results are written well.

—— **Response to Reviewer 2** ——

[What if $\alpha > \frac{1}{4}$?] If $\alpha > \frac{1}{4}$, the conditions are still satisfied with $\alpha = \frac{1}{4}$, so our result still applies, which means the network still behaves like a linear model early in training. It is indeed an interesting question to characterize what happens after the linear learning period, possibly related to some higher-order kernel as the reviewer points out.

—— **Response to Reviewer 3** ——

[Is Assumption 3.1 approximately satisfied by real-world data? Can we transform a given dataset so that Assumption 3.1 holds?] We think it's possible that Assumption 3.1 is approximately satisfied by certain datasets, but not all. Yes, it's possible to transform the data to move closer to this assumption. For example, for CIFAR-10 and MNIST we applied standard pre-processing to normalize each image to have zero mean and fixed norm (lines 468-469). This already enables the linear learning behavior to hold. We believe whitening the data can make the assumption better satisfied.

[About experiments: (1) How is the linear model calculated? (2) How long does the early phase last if we use a large learning rate?] (1) For two-layer NN experiments, the linear model is exactly calculated using Eqn. (11). For multi-layer NNs, since we use erf activation in the experiments, it's easy to show that the corresponding linear model is $x \mapsto cx$ for some constant $c$ (without bias or the norm-dependent feature). We estimate $c$ by $c^2 \approx \frac{\lambda_{\max}(\text{NTK})}{\lambda_{\max}(XX^\top/d)}$, since we expect $\text{NTK} \approx c^2 XX^\top/d$. In general we can use the method sketched in Section 4.1 to compute the linear model analytically. (2) If the learning rate is 5 times larger, the number of iterations of the agreement will be 5 times smaller. Note that the correct way to think about this should be progress of learning rather than specific number of iterations. That is, the shapes of the learning curves will be the same regardless of learning rate (as long as training doesn't diverge), i.e., the agreement will last until the linear function finishes learning.

—— **Response to Reviewer 4** ——

[What does "we show that these common perceptions can be completely false" mean?] We simply meant that the network mimics a linear model and doesn't use its nonlinear capacity early in training, which is a rephrasing of our main result. We will try to modify the sentence to make it clearer. Thanks for pointing out the confusion.

[Any intuition/reason why the networks eventually escape the linear behavior?] The network has the capacity to express complex nonlinear functions, so it should not be a surprise that it eventually becomes nonlinear. [Extension to deep networks?] We discussed extension to deep networks in Section 4. [In Section 3.4, what if ReLU was used instead of ERF?] We indeed provided experiment on ReLU in Figure 2. The corresponding linear model is given by Eqn. (11).

[Meta-Review · NeurIPS 2020]

This paper studies the early training phase of a two-layer neural network and shows that the dynamics remain close to that of a linear model provided that the input data is well behaved. Initially, the paper received mixed reviews (marginally below, top 50%, accept, accept). On the positive side, R2 finds the results novel and quite interesting and the experiments nice, R3 finds the paper interesting, well written, with good experiments, R4 finds the paper presents new theoretical and experimental insights, the analysis intuitive and insightful, and the requirement in width reasonable. On the negative side, R2 finds the results are not complete (limited to one hidden layer) and the experiments not sufficient, and R3 and R4 find the data distribution assumption somewhat restrictive. The rebuttal argues that many interesting deep learning theory papers focus on two-layer networks and that this case is already challenging. I also find that the rebuttal successfully responds to all critiques from R1. Post rebuttal, the negative review was upgraded to accept, so now all reviewers recommend acceptance.